# Nonparametric, Contextual Preference Estimation and Assortment Optimization

## Abstract

The growth of AI- and ML-based decision tools provides an array of decision-support agents that can be implemented into the user's decision-making process. However few tools exist for contextual evaluation for the alignment of decision agents with clinicians' workflows. Consequently, few methods exist to test properties of a contextually-optimal set of aligned agents to target for adoption. Contextual evaluation of decision models is particularly important in settings with few or no gold-standard decisions or preference alignment, such as clinical decision-making. Our work adopts the multinomial logit choice (MNL) model as a framework for evaluating agent-alignment and identifying an optimal agent-set. We assume the observation of selections among a set of agents according to a contextual MNL model, characterized by context-dependent preference parameters. We study a weighted, regularized local likelihood maximization estimator, providing a uniform convergence rate over a compact context space. Additionally, when agent-specific utility parameters or functions are known, we provide results for the identification of a utility-optimal assortment of agents. In this setting, we provide results to construct valid confidence bands on inferential objects of interest and the ability to perform asymptotically valid tests on the composition of this optimal assortment.

**Keywords:** optimal assortment, model evaluation, decision model, nonparametric statistics

## 1 Introduction

Healthcare providers increasingly rely on a range of decision support tools to guide diagnostics and treatment planning. A central challenge for healthcare organizations is determining which subset of these tools constitutes the most effective toolkit for a given clinical setting (Wasylewicz and Scheepers-Hoeks, 2019; Sutton et al., 2020). Identifying this optimal set constitutes evaluating tools that may non-trivially differ in interpretability, cost, and clinical utility. Even among tools with demonstrated efficacy, clinicians continue to express concerns of the relative utility for the adoption of such decision-aids (Angus et al., 2025). Our work attempts to tackle this concern with the following motivation: Given usage data and tool-specific cost or clinical utility information, can we identify if a new class of medican decision-aids (e.g. new generative AI-based aids) should be included in a clinician's toolbox?

We attempt to provide a methodology for this question by modeling clinician preferences as a discrete-choice model over competing decision aids. We and study the resulting optimal assortment problem. We begin by briefly describing the assortment optimization problem, while Section 2 further and more formally articulates the decision model and estimands of interest. We consider a set of $n$ candidate items (plus a "none" or "no item" alternative). Each item has an associated preference parameter and utility parameter. A subset of items is identified and then presented to some decision-agent or decision maker. We refer to this subset as an "offered assortment", or more simply an "assortment". The decision-agent selects one of the offered items, where their decision is dictated (probabilistically) by their preference for these items. The selected item yields a returned utility. In our setup, both these preferences and utility may differ based on the context under which the decision is made.

An assortment can be be described by its expected utility, which is determined by the preferences of all items within that assortment and the item-specific utilities. The preferences determine the probability that each item is selected, while the utilities quantify the reward obtained when that item is chosen. Classically, this problem serves as a model of consumer behavior: preferences are explicitly customer preferences and utilities are item-specific revenue. In the context of clinical decision-aid evaluation they may instead represent quantities such as cost savings or more abstract, user-defined measures of benefit or harm. An ideal item is therefore one that is both likely to be selected and returns a large reward when selected. The optimal assortment problem then consists of identifying a subset of the $n$ items to offer (alongside the ever-present "none" option) that maximizes this expected reward, as formally defined by the estimand in Equation 2.2.

There are two central facets of the assortment-optimization problem for clinical decision support tools. First, the utility of and users' preferences for a given item are context dependent. They may vary with the clinical scenario (Shortliffe and Sepúlveda, 2018; Hager et al., 2024). For generative-AI or language-model–based tools, performance and perceived value can differ by prompt and by downstream task (e.g., note completion, treatment recommendation, imaging analysis) (Kohane, 2024; Li et al., 2025). Second, clinical settings prioritize different objectives: rapid triage may favor speed and immediate interpretability, whereas outpatient and chronic-care clinics may often require extended reasoning periods over longitudinal clinical information and external references.

The clinical nuance requires contextual analysis of the utility and preference or alignment of a given decision tool. Motivated by these nuances, we propose a method for the contextual estimation of preference alignment and assortment optimization. We study a version of the optimal assortment problem using multiway comparison data from a multinomial logit choice model (McFadden, 1973). The multinomial logit choice model is a well-studied, and convenient discrete choice model for the assortment optimization problem (Talluri and van Ryzin, 2004; Rusmevichientong et al., 2010). Applications studying this model range across disciplines including but not limited to marketing (Keane, 2015; Wang et al., 2025a), inventory management and pricing (Aouad and Segev, 2021; Du et al., 2016; Gallego et al., 2020), and health economics (Clark et al., 2014; Hall et al., 2022; Ozdemir et al., 2022).

## 1.1 RECENT LITERATURE

Most directly, we follow the framework of Shen et al. (2023) in their study of the non-contextual, multinomial logit choice model. Our contributions upon this work are in the extension and theoretical guarantees derived under the contextual setting of this decision model. Other recent works studying this assortment optimization problem include reinforcement learning inspired methods and regret bound analyses (Dong et al., 2023). Recently, Han et al. (2025) proposed a rank-breaking algorithm which is minimax optimal with respect to a "sub-optimality gap" of the reward function, in the non-contextual setting.

In recent years, contextual variations of the assortment optimization problem have been analyzed. Preliminary works focused explicitly on linear utility functions (Oh and Iyengar, 2021; Perivier and Goyal, 2022). Recent works study the contextual assortment optimization problem (or "contextual multinomial logit choice bandit problem") under more general classes of utility functions (Zhang and Luo, 2025; Lee and Oh, 2025). These works provide algorithms with minimax optimality guarantees for a regret-bound with respect to the expected utility of the returned assortment.

Other extensions of the non-contextual, multinomial logit choice model (and optimal assortment model) include estimation under various constraints (Barre et al., 2025; Housni et al., 2024; 2025). While we study a dynamic assortment in the offline setting, a large body of literature exists studying this problem in the online setting analyzing a regret bound on the expected utility (or revenue) of an estimated, optimal assortment (Agrawal et al., 2019; Chen et al., 2020; Saha and Gaillard, 2024). Many works study related models of pairwise comparisons, often the Bradley-Terry Luce (BTL) model, and multiway comparison data in the absence of utility functions (Fan et al., 2023; 2024; Chen et al., 2019; 2022). A series of works identified minimax optimality of regularized MLE estimators in non-contextual, pairwise ranking models (Negahban et al., 2017; Chen et al., 2019; 2022). In this area, Wang et al. (2025b) recently studied the contextual statement of this problem with a similar estimator. Though the form of our preference estimator is similar to this work, our goals and target estimand fundamentally differ. We study the contextually optimal assortment and

derive simultaneously valid testing procedures on this object, a function of additional (also possibly contextual) utility functions.

Lastly, the form of our estimator is motivated by the optimality of the regularized maximum likelihood estimator in pairwise comparison analyses (Chen et al., 2019) as well as local-likelihood procedures in classic, nonparametric literature (Tibshirani and Hastie, 1987; Fan et al., 1998).

## 1.2 CONTRIBUTIONS

**Preference and utility gap estimation.** Our work begins with a series of estimation results for the estimation of contextual preference parameters. That is, we work with a parametric decision model that can vary quite generally across covariate or contextual data. We provide uniform convergence rates on the estimation of these preference parameters via a local-likelihood approximation to the decision model's fully parametric likelihood. We then derive similar, uniform results for downstream objects, most importantly a *contextual, marginal utility gap* object. This object determines membership of an item in the contextually optimal assortment and is central to our inferential results. Nonparametric discrete choice work has primarily focused on identification and consistency of very flexible utility or heterogeneity structures in static settings, including nonparametric subutility estimation and nonparametric mixing distributions (Briesch et al., 2007; Bauer et al., 2022). Additional results for general value-function classes arise in online-learning frameworks, studying online regret bounds, rather than offline uniform convergence rates for a context-indexed preference functions (Zhang and Luo, 2025). Our work adds to the existing contextual, multinomial logit choice by providing uniform bounds on preference parameter and utility gap estimation via a regularized, local likelihood-based procedure, in addition to inferential tools outlined below.

**Inference on the contextually optimal assortment.** In our work, we define the contextually optimal assortment (our final estimand of interest) as the selection of items maximizing the expected utility. This definition takes into account the contribution of an item to an assortment's global utility with respect to individuals' preferences for that item *and* the relative contribution of the item via its utility function. We define an object that defines membership of an item in this contextually optimal assortment. Most importantly, we provide confidence bands on this object that are valid uniformly over the parameter space (of the underlying decision model) and valid simultaneously across a compact context space. This construction builds on the theoretical contributions to multiplier bootstrap theory in Wang et al. (2025b). The validity of our inferential procedures requires new technical arguments to control the suprema of the resulting empirical processes and to justify a Gaussian multiplier bootstrap in this contextual, combinatorial setting. We provide a contextual testing procedure, that similarly controls Type I error uniformly over the decision-model's parameter space and our compact, contextual space. These guarantees require only mild smoothness conditions on the preference parameters as a nonparametric function of context. Prior nonparametric discrete choice work provides asymptotic results for utility or preference functions themselves but does not offer an inferential framework for combinatorial functionals (related to the optimal assortment) (Briesch et al., 2007; Bauer et al., 2022; Berry and Haile, 2024). In this sense, our testing procedure appears to be the first offline frequentist method designed specifically for context-dependent assortments and marginal utility gaps in the contextual MNL model.

## 1.3 ORGANIZATION

Section 2 discusses the data-generating model of the observed data. Importantly, this section includes the contextual nature of our setup and an outline of the inferential procedure. Specifically, Section 2.3 contains examples of possible tests of interest in addition to a brief outline of our proposed testing procedure. Section 3 provides uniform guarantees on the estimation of preferences, $\theta(\mathbf{x})$, that parameterize the data-generating decision model. This section also studies a debiasing procedure, which is necessary for downstream objects of interest with which we construct our testing procedure. Section 4 then outlines this testing procedure, constructs confidence bands that are valid uniformly over the space of $\theta$ and simultaneously over context $\mathbf{x}$, and articulates a testing procedure (with similar guarantees) that follows from these confidence bands.

## 2 PRELIMINARIES

We begin by describing the multinomial logit model, by which we parameterize the underlying data-generating process of the decision data. We extend the typical setting to allow underlying preference values to vary (somewhat) arbitrarily across some context space $\Omega$.

### 2.1 DECISION MODEL

Consider a set of items $[n]$, and the augmented set $[n]_+ = [n] \cup \{0\}$. Here 0 represents "no selection", as opposed to selecting from among the $n$ items. We let $\mathcal{S}$ represent the set of all $2^n$ possible assortments of $[n]$. Similarly we represent an instance of an assortment of these items as $\mathcal{S} \subseteq [n]$ and let $\mathcal{S}_+ = \mathcal{S} \cup \{0\}$. In practice, we expect to observe only a subset of all possible assortments, which we will represent by $\mathcal{E} \subseteq \mathcal{S}$. We assume that we observe a given assortment with probability $p$. Lastly, for a given assortment $\mathcal{S}$, let $\mathcal{E}_{\mathcal{S}}$ represent the indicator of observation, i.e. $\mathcal{E}_{\mathcal{S}} = \mathbb{1}(\mathcal{S} \in \mathcal{E})$.

For a given $\mathbf{x} \in \Omega$, we assume each item $i \in [n]_+$ has a true, underlying preference value $u_i^*(\mathbf{x}) > 0$ and corresponding log-transformed value $\theta_i^*(\mathbf{x}) := \log(u_i^*(\mathbf{x}))$. We also write the column vectors of these values as $\mathbf{u}^*(\mathbf{x}) = (u_0^*(\mathbf{x}), ..., u_n^*(\mathbf{x}))^T$ and similarly $\boldsymbol{\theta}^*(\mathbf{x}) = (\theta_0^*(\mathbf{x}), ..., \theta_n^*(\mathbf{x}))^T$. We characterize the separation of these preferences by the condition number $c_\theta := \sup_{\mathbf{x} \in \Omega} \max_{i,j \in [n]_+} \frac{e^{\theta_j^*(\mathbf{x})}}{e^{\theta_i^*(\mathbf{x})}}$. For a given assortment $\mathcal{S} \in \mathcal{E}$, we observe $L, iid$ decisions $\mathbf{y}_{\mathcal{S}}^\ell$. We also observe the additional contextual information $\mathbf{X}_{\mathcal{S}}^\ell$, where $\{\mathbf{X}_{\mathcal{S}}^\ell\} \in \Omega$ for any $\ell \in [L], \mathcal{S} \in \mathcal{E}$. We can characterize the $\ell$th decision for the assortment $\mathcal{S}_+$ as $iid$, categorical random variables, or

$$\left\{ y_{\mathcal{S}}^{(i,\ell)} \right\}_{i \in \mathcal{S}_+} \mid \mathbf{X}_{\mathcal{S}}^\ell \sim \text{Multinomial}(1, p_0, ..., p_{|\mathcal{S}|}) \text{ where } p_i = \frac{\exp\left\{\theta_i^*\left(\mathbf{X}_{\mathcal{S}}^\ell\right)\right\}}{\sum_{j \in \mathcal{S}_+} \exp\left\{\theta_j^*\left(\mathbf{X}_{\mathcal{S}}^\ell\right)\right\}} . \quad (2.1)$$

Note that this model is only identifiable up to constant shifts in $\boldsymbol{\theta}^*(\cdot)$, thus we assume the constraint $\mathbf{1}^T \boldsymbol{\theta}^*(\mathbf{x}) = 0$ for any $\mathbf{x} \in \Omega$. Samples across assortments are independent but not $iid$, due to differences in the offered item set.

For brevity, we will let $\mathbf{y}_{\mathcal{S}}^\ell = \left[ y_{\mathcal{S}}^{(0,\ell)}, \ldots, y_{\mathcal{S}}^{(|\mathcal{S}|,\ell)} \right]^T$ and let $\mathbf{y}$ represent the complete collection of observed decisions, i.e. $\mathbf{y} \equiv \{\mathbf{y}_{\mathcal{S}}^\ell\}_{\ell \in [L], \mathcal{S} \in \mathcal{E}}$. Similarly let $\mathcal{D} = (\mathcal{E}_{\mathcal{S}}, \mathbf{y}_{\mathcal{S}}^\ell, \mathbf{X}_{\mathcal{S}}^\ell)_{(\ell,\mathcal{S}) \in ([L], \mathcal{E})}$ represent the total collection of observed data. We will also let $\mathbf{y}$ and $\mathbf{X}$ represent all observed decision and context respectively.

### 2.2 UTILITY AND OPTIMAL ASSORTMENT

Each item $i \in [n]$ is also associated with a utility $r_i(\mathbf{x})$. Without loss of generality, we assign the no-decision option to have corresponding utility $r_0(\mathbf{x}) = 0, \forall \mathbf{x} \in \Omega$. We also assign a utility condition number describing separation of these values (excluding the no-decision utility), $c_r := \sup_{\mathbf{x} \in \Omega} \max_{i,j \in [n]} \frac{r_i(\mathbf{x})}{r_j(\mathbf{x})}$.

We define total expected utility for assortment $\mathcal{S}$ at $\mathbf{x} \in \Omega$ and, for a given $\mathbf{x}$, we define the optimal assortment at $\mathbf{x}$ as the smallest assortment among those which maximize the total expected utility:

$$R(\mathcal{S}; \mathbf{x}) := \sum_{j \in \mathcal{S}} r_j(\mathbf{x}) \frac{u_j(\mathbf{x})}{\sum_{i \in \mathcal{S}_+} u_i(\mathbf{x})}, \quad \mathcal{S}^*(\mathbf{x}) = \underset{\substack{\mathcal{S} \in \arg\max R(\mathcal{S}; \mathbf{x}) \\ \mathcal{S} \subseteq [n]}}{\arg\min} |\mathcal{S}| . \quad (2.2)$$

Existing work demonstrates that this optimal assortment can be recovered given the underlying preference values $\mathbf{u}(\mathbf{x})$ and utilities $\mathbf{r}(\mathbf{x})$. For a fixed $\mathbf{x}$, we represent the ordered utilities as $r_{(1)}(\mathbf{x}) = \max_{i \in [n]} r_i(\mathbf{x})$ and $r_{(n)}(\mathbf{x}) = \min_{i \in [n]} r_i(\mathbf{x})$.

For a given set of contextual preference and utility values, we consider the following algorithm from Talluri and van Ryzin (2004) to identify an assortment $\mathcal{S}(\mathbf{x})$:

If the true underlying preference values $\mathbf{u}^*(\mathbf{x})$ and utility values $\mathbf{r}^*(\mathbf{x})$ are supplied, then the resulting $\mathcal{S}(\mathbf{x})$ recovers this optimal assortment $\mathcal{S}(\mathbf{x})^*$ as defined in Equation 2.2 (Talluri and van Ryzin, 2004).

---

**Algorithm 1** Smallest Optimal Assortment Calculation

---

**Require: Input**: Preference values $\mathbf{u}(\mathbf{x})$; utility values $\mathbf{r}(\mathbf{x})$
1: Sort utility values $r_{(1)}(\mathbf{x}) > ... > r_{(n)}(\mathbf{x})$
2: Index preference values such that $u_{(j)}$ corresponds to the item with $j$th largest utility
3: Initialize $j = 0, \Delta_j(\mathbf{x}) = -1$
4: **while** $j \leq n$ and $\Delta_j < 0$ **do**
5:     $j \leftarrow j + 1$
6:     $\Delta_j(\mathbf{x}) \leftarrow \sum_{i=1}^{j} r_{(i)}(\mathbf{x}) u_{(i)}(\mathbf{x}) - \left( \sum_{k=0}^{j} u_{(k)}(\mathbf{x}) \right) r_{(j)}(\mathbf{x})$
7: **end while**
8: **Output**: $[j] \to \mathcal{S}(\mathbf{x})$

---

## 2.3 INFERENCE ON OPTIMAL ASSORTMENT

From above, we can recover the smallest, contextually-optimal assortment, defined as the assortment which maximizes the expected revenue at a given point of context. Now let us describe the formal procedure of testing properties of $\mathcal{S}^*(\mathbf{x})$ and present two motivating examples. These procedures and examples mirror those of Shen et al. (2023), adapted for the contextual nature of our setup. Without loss of generality, assume that the items are indexed such that $r_1 \geq r_2 \geq \cdots \geq r_n \geq r_0 = 0$. As a result, the contextually optimal assortment at $\mathbf{x}$ takes the form $[\kappa^*(\mathbf{x})] \subseteq [n]$. Here, $\kappa^*(\mathbf{x})$ indicates the final item (or largest index) included in $\mathcal{S}^*(\mathbf{x})$ given the point $\mathbf{x} \in \Omega$.

Testing various properties of $\mathcal{S}^*(\mathbf{x})$ follows a general framework. Firstly, define hypotheses $H_0, H_1$ that relates to the substantive research question of interest. These often relate to the composition of $\mathcal{S}^*(\mathbf{x})$, and we present two such examples below. Secondly, define a null-property set, $\mathcal{K}_0 \subseteq [n]$, such that the previously defined hypotheses are equivalent to

$$H_0 : \kappa^*(\mathbf{x}) \in \mathcal{K}_0, \quad H_1 : \kappa^*(\mathbf{x}) \notin \mathcal{K}_0 .$$

The specific form of any $\mathcal{K}_0$ obviously depends on the hypotheses of interest, but most generally, $\mathcal{K}_0$ contains the set of indices satisfying the property of $\mathcal{S}^*(\mathbf{x})$ that the analyst first articulates in $H_0$.

For example, consider interest in a clinic incorporating one new, AI-based reference tool (e.g. OpenEvidence) directly into their electronic healthcare record. The clinic may be interested to see if providers effectively "trust" the tool, or if they tend to spend additional time corroborating the genAI-based recommendations. Such a question distills to testing the membership of a given item $i$ in $\mathcal{S}^*(\mathbf{x})$, which can be stated with respect to $\mathcal{S}^*(\mathbf{x})$ as

$$H_0 : i \in \mathcal{S}^*(\mathbf{x}), \quad H_1 : i \notin \mathcal{S}^*(\mathbf{x}) .$$

The null-property set can then be defined by reformulating this hypothesis, as $H_0 : \kappa^*(\mathbf{x}) \in \mathcal{K}_0, \quad \mathcal{K}_0 := \{j \in [n] : j < i\}$. This estimation and testing procedure are evaluated at a fixed point of context $\mathbf{x} \in \Omega$ that is specified by the analyst.

As a more complicated example, suppose that a clinic is considering a suite of new tools from Company X. Interest lies in identifying if these new tools comprise a meaningful proportion of decision-aids of use in the clinic. A certain proportion of Company X's products would then justify swapping a portion of the clinic's current ecosystem to use tools from this Company X. We can formulate this as testing properties of a subset of tools, $A \subset [n]$. Specifically we would like to test whether items from $A$ comprise a certain proportion (say $q$) of items in the contextually optimal assortment, $\kappa^*(\mathbf{x})$. Here, the set $A$ may comprise a certain type of tool (e.g. a set of generative AI-based tools) which are of particular interest. The hypotheses with respect to $\mathcal{S}^*(\mathbf{x})$ can be stated as

$$H_0 : \frac{|A \cap \kappa^*(\mathbf{x})|}{|\kappa^*(\mathbf{x})|} > q, \quad H_1 : \frac{|A \cap \kappa^*(\mathbf{x})|}{|\kappa^*(\mathbf{x})|} \leq q$$

The null property set can then be defined as $\mathcal{K}_0 = \{j \in [n] : [j] \cap [n]/n > q\}$, with the null hypothesis again translating to $H_0 : \kappa^*(\mathbf{x}) \in \mathcal{S}^*(\mathbf{x})$. For further examples in the static preference setting, see Shen et al. (2023), each of which can be easily adapted to the contextual setting as above. These examples indicate pointwise testing. That is, the analyst is interested in estimating and

testing for a given $\mathbf{x} \in \Omega$. This setup translates immediately to simultaneous testing by stating null hypotheses (and constructing related null property sets) that hold for all $\mathbf{x} \in \Omega$.

From Algorithm 1 above, membership of an item $k$ in $\mathcal{S}^*(\mathbf{x})$ depends entirely on the sign of $\Delta_k(\mathbf{x})$. In Section 4, we discuss concretely how to construct confidence bands on $\Delta_k(\mathbf{x}), k \in [n]$ of the form $[\widehat{D}_L(\mathbf{x}), \widehat{D}_U(\mathbf{x})]$. These objects are defined in detail in 4, but momentarily take $\widehat{D}_L(\mathbf{x}) \leq \widehat{D}_U(\mathbf{x})$, each of which are in $[n]$. One proceeds with inference on $\mathcal{S}^*(\mathbf{x})$ by (1) defining a property or null hypothesis of interest $H_0$, (2) constructing the null property set $\mathcal{K}_0$ with the equivalence $\{\mathcal{S}^*(\mathbf{x}) \text{ satisfies } H_0\} \iff \{\kappa^*(\mathbf{x}) \in \mathcal{K}_0\}$, (3) calculating the confidence band $[\widehat{D}_L(\mathbf{x}), \widehat{D}_U(\mathbf{x})]$, and (4) rejecting $H_0$ iff $[\widehat{D}_L(\mathbf{x}), \widehat{D}_U(\mathbf{x})] \cap \mathcal{K}_0 = \emptyset$. The form of the testing procedure relies on the equivalence between the form of the null hypothesis, $H_0$ with respect to $\mathcal{S}^*(\mathbf{x})$ and the membership of $\kappa^*(\mathbf{x}) \in \mathcal{K}_0$ through the constructed null-property set.

### 2.4 Assumptions

**Assumption 1** (Comparison Probability). In Section 2, we articulate the decision model, which includes sampling assortments $\mathcal{S} \subset [n]$ randomly with probability $p$, and making $L$ repeated comparisons. In Theorem 3.1, we require $2^n p L \gtrsim n \log(n)$ and $n d \log(2^n p L) \lesssim 2^n p L$. These scaling assumptions assure sufficient connectedness of the comparison graph for estimation of $\boldsymbol{\theta}$. We assume $L$ is equal for all assortments only for niceness of presentation. Results extend immediately by allowing $L$ to vary across assortments as $L_{\mathcal{S}}$.

**Assumption 2** (Context Space). We assume the contextual information is supported on a compact space of fixed dimension $d$, taken as $\Omega = [0, 1]^d$ without loss of generality. We assume that $f$ lies in the Sobolev space $W^{2,\infty}(\Omega)$, which also implies $f(\mathbf{x}) \in [c_f, C_f], \mathbf{x} \in \Omega$ for bounded $c_f, C_f > 0$.

**Assumption 3** (Preference Values). We assume that $\boldsymbol{\theta} : \Omega \mapsto \mathbb{R}^{n+1}$ is twice-differentiable with bounded derivatives, i.e. $\boldsymbol{\theta} \in W^{2,\infty}(\Omega)$, with additional identification and regularity conditions relevant to the decision model. That is $\boldsymbol{\theta} \in \Theta$ with $\Theta$ being the subset of $W^{2,\infty}(\Omega)$

$$\Theta = \left\{ \boldsymbol{\theta} \in W^{2,\infty}(\Omega) : \mathbf{1}^T \boldsymbol{\theta}(\mathbf{x}) = 0, \forall \mathbf{x} \in \Omega ; \sup_{\mathbf{x} \in \Omega} \max_{i,j \in [n]} e^{\theta_i(\mathbf{x}) - \theta_j(\mathbf{x})} \leq M \right\} . \quad (2.3)$$

**Assumption 4.** We assume a multiplicative kernel, $K(\mathbf{x}) = \prod_{i=1}^d K'(x_i)$ where $K' : A \mapsto \mathbb{R}$ is unimodal, symmetric, bounded for some compact support $A \subset \mathbb{R}$. Additional technical assumptions on the class of kernel functions are included in Section H.1.

## 3 Estimation

### 3.1 Preference Parameter Estimation

Under the MNL model characterized in equation 2.1 and the set $\mathcal{E}$ of observed assortments, the negative log-likelihood function can be expressed as

$$\Lambda(\boldsymbol{\theta}; \mathcal{D}) = -\sum_{\mathcal{S} \in \mathcal{E}} \left[ \sum_{i \in \mathcal{S}_+} L^{-1} \sum_{\ell \in [L]} y_{\mathcal{S}}^{(i,\ell)} \theta_i(\mathbf{X}_{\mathcal{S}}^\ell) - \log \left( \sum_{j \in \mathcal{S}_+} e^{\theta_j(\mathbf{X}_{\mathcal{S}}^\ell)} \right) \right] . \quad (3.1)$$

To estimate $\boldsymbol{\theta}(\mathbf{x})$ nonparametrically, we study a kernel-weighted local (log-)likelihood, indexed at a given context $\mathbf{x}$,

$$\mathcal{L}(\boldsymbol{\beta}; \mathbf{x}, \mathcal{D}) = -\sum_{\mathcal{S} \in \mathcal{E}} \sum_{\ell \in [L]} \left[ L^{-1} K_{\mathbf{H}} \left( \mathbf{X}_{\mathcal{S}}^\ell - \mathbf{x} \right) \left( \sum_{i \in \mathcal{S}_+} y_{\mathcal{S}}^{(i,\ell)} \beta_i - \log \left( \sum_{j \in \mathcal{S}_+} e^{\beta_j} \right) \right) \right] \quad (3.2)$$

for parameter vector $\boldsymbol{\beta} \in \mathbb{R}^{n+1}$. Our proposed estimator is the regularized minimizer of this local, negative log-likelihood:

$$\widehat{\boldsymbol{\theta}}(\mathbf{x}) = \underset{\boldsymbol{\beta} \in \mathbb{R}^{n+1}, \mathbf{1}^T \boldsymbol{\beta} = 0}{\arg\min} \mathcal{L}(\boldsymbol{\beta}; \mathbf{x}, \mathcal{D}) + \frac{\lambda}{2} ||\boldsymbol{\beta}||_2^2 . \quad (3.3)$$

The constraint $\mathbf{1}^T\boldsymbol{\beta} = 0$ enforces the aforementioned identification condition of our multinomial logit choice model.

From the form of Algorithm 1, one can observe that estimation of $\boldsymbol{\theta}(\mathbf{x})$ and knowledge of $\mathbf{r}(\mathbf{x})$ permits identification of the optimal assortment. Thus we begin by characterizing the estimation rate of our procedure above for $\widehat{\boldsymbol{\theta}}(\mathbf{x})$. The form of this result mirrors classic results in non-parametric regression, with an additional log-term resulting from uniform control over $\mathbf{x} \in \Omega$. We first state a result as a function of the tunable bandwidth parameter, $h$.

**Theorem 3.1.** Suppose there exists $c, C > 0$ such that $2^n p \geq Cn \log n$ and $nd \log(2^n pL) \leq c(2^n pL)^{4/4+d}$. Then, under the assumptions of Section 2.4 and the selection of $\lambda \asymp 2^n p \sqrt{\frac{d \log(2^n pLh^{-d})}{2^n pLh^d}} + 2^n pdh^2$, the regularized MLE $\widehat{\boldsymbol{\theta}}(\mathbf{x})$ defined in Equation 3.3 satisfies

$$\sup_{\mathbf{x}\in\Omega} ||\widehat{\boldsymbol{\theta}}(\mathbf{x}) - \boldsymbol{\theta}(\mathbf{x})||_2 \lesssim \sqrt{n} \left( \sqrt{\frac{d \log(2^n pLh^{-d})}{2^n pLh^d}} + dh^2 \right)$$

with probability at least $1 - O(n^{-10})$.

The proof of Theorem 3.1 in Section B involves studying the optimization procedure of Equation 3.3. The proof strategy mirrors that of previous studies of pairwise comparison and discrete choice models, adapted for our local likelihood in Equation 3.2 (Chen et al., 2019; Shen et al., 2023). The rate is stated in terms of a tunable bandwidth parameter, $h$. One can select $h$ in order to balance the bias and variance in the rate above, which again mirrors classical nonparametric regression results.

**Remark 3.2** ($h$ selection). Choosing $h \asymp [\log(2^n pL)/(2^n pL)]^{1/d+4}$ results in the simplification of Theorem 3.1

$$\sup_{\mathbf{x}\in\Omega} ||\widehat{\boldsymbol{\theta}}(\mathbf{x}) - \boldsymbol{\theta}(\mathbf{x})||_2 \lesssim d\sqrt{n} \left( \frac{\log(2^n pL)}{(2^n pL)} \right)^{2/4+d},$$

again with probability at least $1 - O(n^{-10})$ under the same conditions as Theorem 3.1.

## 3.2 DEBIASING

In order to conduct inference on our assortment, we must debias our preference parameter estimate $\widehat{\boldsymbol{\theta}}(\mathbf{x})$ studied in Theorem 3.1 due to the regularization in our estimation procedure. The form of this debiasing procedure is motivated by work in the pairwise comparisons model which studies a the regularized MLE of pairwise comparisons to infer top-$K$ rankings (Liu et al., 2023). It is useful here to identify the forms of the gradient and Hessian of our local-likelihood loss function with respect to $\boldsymbol{\beta}$,

$$\nabla\mathcal{L}(\boldsymbol{\beta}; \mathbf{x}, \mathcal{D}) = -\sum_{\mathcal{S}\in\mathcal{E}} \sum_{\ell\in[L]} L^{-1} K_{\mathbf{H}}\left(\mathbf{X}_{\mathcal{S}}^\ell - \mathbf{x}\right) \sum_{i\in\mathcal{S}_+} \left[ y_{\mathcal{S}}^{(i,\ell)} - \frac{e^{\beta_i}}{\sum_{j\in\mathcal{S}_+} e^{\beta_j}} \right] \mathbf{e}_i \tag{3.4}$$

$$\nabla^2\mathcal{L}(\boldsymbol{\beta}; \mathbf{x}, \mathcal{D}) = \sum_{\mathcal{S}\in\mathcal{E}} \sum_{\ell\in[L]} L^{-1} K_{\mathbf{H}}\left(\mathbf{X}_{\mathcal{S}}^\ell - \mathbf{x}\right) \sum_{i,k\in\mathcal{S}_+:k>i} \left[ \frac{e^{\beta_i} e^{\beta_k}}{\left(\sum_{j\in\mathcal{S}_+} e^{\beta_j}\right)^2} (\mathbf{e}_i - \mathbf{e}_k)(\mathbf{e}_i - \mathbf{e}_k)^T \right]. \tag{3.5}$$

We can now define the debiased version of our estimator in Equation 3.3,

$$\widehat{\boldsymbol{\theta}}^d(\mathbf{x}) := \widehat{\boldsymbol{\theta}}(\mathbf{x}) - \nabla^2\mathcal{L}(\widehat{\boldsymbol{\theta}}; \mathbf{x}, \mathcal{D})^\dagger \nabla\mathcal{L}(\widehat{\boldsymbol{\theta}}; \mathbf{x}, \mathcal{D}). \tag{3.6}$$

Here we understand $\nabla\mathcal{L}(\widehat{\boldsymbol{\theta}}; \mathbf{x}, \mathcal{D})$ to represent the kernel-weighted, local likelihood evaluated at $\beta = \widehat{\boldsymbol{\theta}}(\mathbf{x})$, or more comprehensively $\nabla\mathcal{L}(\widehat{\boldsymbol{\theta}}(\mathbf{x}); \mathbf{x}, \mathcal{D})$. We allow for similar notation for $\boldsymbol{\theta}^*$. Note that $\mathbf{1}^T\nabla\mathcal{L}(\theta; \mathbf{x}, \mathcal{D}) = 0$ for any $\theta, \mathbf{x}$. Thus the debiased estimator also satisfies $\mathbf{1}^T\widehat{\boldsymbol{\theta}}^d(\mathbf{x}) = 0$.

**Lemma 3.3** (Debiasing of $\boldsymbol{\theta}$). Suppose that Theorem 3.1 holds with selection of $h$ as in Remark 3.2. Then the debiased estimator $\widehat{\boldsymbol{\theta}}^d(\mathbf{x})$ defined in Equation 3.6 satisfies

$$\widehat{\boldsymbol{\theta}}^d(\mathbf{x}) - \boldsymbol{\theta}^*(\mathbf{x}) = -\nabla^2 \mathcal{L}(\boldsymbol{\theta}^*; \mathbf{x}, \mathcal{D})^\dagger \nabla \mathcal{L}(\boldsymbol{\theta}^*; \mathbf{x}, \mathcal{D}) + \mathbf{R}_\theta(\mathbf{x})$$

where, with probability at least $1 - O(n^{-10})$,

$$\sup_{\mathbf{x} \in \Omega} ||\mathbf{R}_\theta(\mathbf{x})||_2 \lesssim n^{5/2} d \left( \frac{\log(2^n pL)}{2^n pL} \right)^{4/4+d}$$

holds. See Section C.2 for a proof of this result.

**Corollary 3.4.** Under the conditions of Lemma 3.3, we also have that

$$\sup_{\mathbf{x} \in \Omega} ||\widehat{\boldsymbol{\theta}}^d(\mathbf{x}) - \boldsymbol{\theta}^*(\mathbf{x})||_2 \lesssim n^3 \sqrt{d} \left( \frac{\log(2^n pL)}{2^n pL} \right)^{2/4+d}$$

with probability at least $1 - O(n^{-10})$. The result follows from Lemmas 3.3 and C.2 for large enough $n, L$ as $[d \log(2^n pL)/(2^n pL)]^{2/4+d} = o(1)$.

# 4 INFERENCE

## 4.1 SETUP AND MOTIVATION

With the ability to consistently estimate $\boldsymbol{\theta}^*(\mathbf{x})$, we can motivate and provide a framework for contextual testing related to our true estimand of interest, the utility-maximizing assortment as defined in Equation 2.2. Through the remainder of this section, we assume without loss of generality the items are indexed such that $r_1(\mathbf{x}) > r_2(\mathbf{x}) \cdots > r_n(\mathbf{x})$.

We focus on the following marginal utility object to test properties of this optimal assortment, $\Delta_j(\boldsymbol{\beta}, \mathbf{x}) := \sum_{i=1}^{j} r_i(\mathbf{x}) e^{\beta_i} - \left( \sum_{k=0}^{j} e^{\beta_k} \right) r_j(\mathbf{x})$. By Algorithm 1, membership of an item is determined by the signs of $\{\Delta_k(\boldsymbol{\theta}^*, \mathbf{x})\}_{k \in [n]}$. More specifically, the optimal assortment is determined by the largest $k$ such that $\Delta_k(\boldsymbol{\theta}^*, \mathbf{x}) < 0$. This motivates the following series of results, which construct and demonstrate the validity of a testing procedure. Armed with this procedure, we are able to conduct inference on the contextually optimal assortment.

## 4.2 INFERENTIAL THEORY

Based on the de-biasing construction of Section 3.2, we must define a series of additional objects. Firstly, let us define central object with which we can conduct inference on the optimal assortment: $\widehat{\Delta}_k(\widehat{\boldsymbol{\theta}}^d, \mathbf{x})$.

$$\Delta_k(\widehat{\boldsymbol{\theta}}^d, \mathbf{x}) = \sum_{i=1}^{k} r_i(\mathbf{x}) \exp\left( \widehat{\theta}_i^d(\mathbf{x}) \right) - \left[ \sum_{j=0}^{k} \exp\left( \widehat{\theta}_j^d(\mathbf{x}) \right) \right] r_k(\mathbf{x}) .$$

We then construct the inferential object of interest $\widehat{\Delta}_k(\widehat{\boldsymbol{\theta}}^d, \mathbf{x})$. We will make use of the short-hand notation $\widehat{\Delta}_k(\mathbf{x}) \equiv \Delta_k(\widehat{\boldsymbol{\theta}}^d, \mathbf{x})$ and $\Delta_k(\mathbf{x}) = \Delta_k(\boldsymbol{\theta}^*, \mathbf{x})$. We define a related object

$$\mathbf{v}_k(\boldsymbol{\theta}, \mathbf{x}) = \left[ (0 - r_k(\mathbf{x})) e^{\theta_0(\mathbf{x})}, \ (r_1(\mathbf{x}) - r_k(\mathbf{x})) e^{\theta_1(\mathbf{x})}, \ \ldots, \ (r_{k-1}(\mathbf{x}) - r_k(\mathbf{x})) e^{\theta_{k-1}(\mathbf{x})}, \ 0, \ \ldots, \ 0 \right]^T .$$

Treating $\Delta_k(\boldsymbol{\beta}, \mathbf{x})$ as a function, $\mathbf{v}_k(\boldsymbol{\beta}, \mathbf{x})$ is simply the gradient with respect to $\boldsymbol{\beta}$. As with $\Delta_k(\mathbf{x})$, we let $\widehat{\mathbf{v}}_k(\mathbf{x}) \equiv \mathbf{v}_k(\widehat{\boldsymbol{\theta}}^d, \mathbf{x})$ and $\mathbf{v}_k(\mathbf{x}) \equiv \mathbf{v}_k(\boldsymbol{\theta}^*, \mathbf{x})$. Lastly, to construct the Gaussian multiplier bootstrap procedure, we then define a perturbed version of the negative log likelihood's and local log likelihood's gradients as

$$\nabla \widetilde{\Lambda}(\boldsymbol{\theta}; \mathbf{x}, \mathcal{D}) := -\sum_{\mathcal{S} \in \boldsymbol{\mathcal{E}}} \sum_{\ell \in [L]} Z_\mathcal{S}^\ell L^{-1} K_\mathbf{H}\left( \mathbf{X}_\mathcal{S}^\ell - \mathbf{x} \right) \sum_{i \in \mathcal{S}_+} \left[ y_{\mathcal{S}_+}^{(i,\ell)} - \frac{e^{\theta_i(\mathbf{X}_\mathcal{S}^\ell)}}{\sum_{j \in \mathcal{S}_+} e^{\theta_j(\mathbf{X}_\mathcal{S}^\ell)}} \right] \mathbf{e}_i$$

$$\nabla \widetilde{\mathcal{L}}(\boldsymbol{\theta}; \mathbf{x}, \mathcal{D}) := -\sum_{\mathcal{S} \in \boldsymbol{\mathcal{E}}} \sum_{\ell \in [L]} Z_\mathcal{S}^\ell L^{-1} K_\mathbf{H}\left( \mathbf{X}_\mathcal{S}^\ell - \mathbf{x} \right) \sum_{i \in \mathcal{S}_+} \left[ y_{\mathcal{S}_+}^{(i,\ell)} - \frac{e^{\theta_i(\mathbf{x})}}{\sum_{j \in \mathcal{S}_+} e^{\theta_j(\mathbf{x})}} \right] \mathbf{e}_i$$

for iid standard Gaussian random variables, $Z_{\mathcal{S}}^{\ell}$. The remaining procedure is motivated by the recent extensions of Gaussian multiplier bootstrap theory developed in Wang et al. (2025b).

Now, we define the following statistic

$$T = \sup_{\mathbf{x} \in \Omega} \max_{k \in [n]} \sqrt{Lh^d} \left( \frac{\widehat{\Delta}_k(\mathbf{x}) - \Delta_k(\mathbf{x})}{\sqrt{\widehat{\mathbf{v}}_k(\mathbf{x})^T \nabla^2 \mathcal{L}(\widehat{\boldsymbol{\theta}})^\dagger \widehat{\mathbf{v}}_k(\mathbf{x})}} \right) . \tag{4.1}$$

We will approximate the quantiles of $T$ by

$$W = \sup_{\mathbf{x} \in \Omega} \max_{k \in [n]} \sqrt{Lh^d} \left( \frac{-\widehat{\mathbf{v}}_k(\mathbf{x})^T \nabla^2 \mathcal{L}(\widehat{\boldsymbol{\theta}})^\dagger}{\sqrt{\widehat{\mathbf{v}}_k(\mathbf{x})^T \nabla^2 \mathcal{L}(\widehat{\boldsymbol{\theta}})^\dagger \widehat{\mathbf{v}}_k(\mathbf{x})}} \nabla \widetilde{\mathcal{L}}(\widehat{\boldsymbol{\theta}}) \right) . \tag{4.2}$$

Note that $\mathcal{L}, \Lambda$, and related objects remain functions of $(\mathbf{x}, \mathcal{D})$, although we suppress explicit notation of this dependency. Recall $\Lambda$ is the likelihood function as defined in Equation 3.1 and $\nabla \widetilde{\Lambda}(\widehat{\theta}) \equiv \nabla \widetilde{\Lambda}(\boldsymbol{\theta}; \mathbf{x}, \mathcal{D})$ is as defined above.

Now define the following quantile of $W$, conditional on the observed data, $\mathcal{D}$, $\widehat{c}_W(\alpha; \mathcal{D}) = \inf\{t \in \mathbb{R} : \mathbb{P}(W \geq t \mid \mathcal{D}) \leq \alpha\}$. To conduct inference on the optimal assortment $\mathcal{S}^*(\mathbf{x})$, the statistic $T$ arises in the construction of confidence bands on $\Delta_k(\mathbf{x})$ for $k \in [n]$. Conditional on data, $W$ becomes a Gaussian process, the quantiles of which we use to approximate the quantiles of $T$. Given these confidence bands, we can follow the testing procedure as discussed in Section 2.3. We outline the implementation in further detail in Algorithms 2 and 3 of Section 4.3. We thus begin by showing that $\widehat{c}_W(\alpha; \mathcal{D})$ is a valid approximation of the quantiles of our statistic $T$, in the following theorem.

**Theorem 4.1.** Suppose the conditions are met such that Theorem 3.1 holds. If we also have that $n^3 \log(2^n pL) \lesssim (2^n pL)^\varepsilon$ for some $\varepsilon < 2/(4+d)$, then $\sup_{\alpha \in (0,1)} |\mathbb{P}(T \geq \widehat{c}_W(\alpha; \mathcal{D})) - \alpha| = o(1)$. That is, the quantiles of our statistic $T$ asymptotically approach the conditional quantiles of $W$, which (conditional on $\mathcal{D}$) is the supremum of a centered Gaussian process. See Section D.1 for a proof of this result, which results on recent extensions of the Gaussian multiplier bootstrap (Chernozhukov et al., 2013; 2014; Wang et al., 2025b).

Now, we can define a procedure to construct $1 - \alpha$ confidence bands for $\Delta_k(\mathbf{x})$, using the quantile approximation above. First, we define the objects

$$\widehat{D}_L(\mathbf{x}) := \max \left\{ k \in [n] : \widehat{\Delta}_k(\mathbf{x}) < -\widehat{c}_W(\alpha/2; \mathcal{D}) \sqrt{\frac{\widehat{\mathbf{v}}_k(\mathbf{x})^T \nabla^2 \mathcal{L}(\widehat{\boldsymbol{\theta}}; \mathbf{x}, \mathcal{D}) \widehat{\mathbf{v}}_k(\mathbf{x})}{Lh^d}} \right\} \tag{4.3}$$

$$\widehat{D}_U(\mathbf{x}) := \max \left\{ k \in [n] : \widehat{\Delta}_k(\mathbf{x}) \leq \widehat{c}_W(\alpha/2; \mathcal{D}) \sqrt{\frac{\widehat{\mathbf{v}}_k(\mathbf{x})^T \nabla^2 \mathcal{L}(\widehat{\boldsymbol{\theta}}; \mathbf{x}, \mathcal{D}) \widehat{\mathbf{v}}_k(\mathbf{x})}{Lh^d}} \right\} . \tag{4.4}$$

With our quantile approximation result above, we can demonstrate that $\widehat{D}_L(\mathbf{x}), \widehat{D}_U(\mathbf{x})$ are valid, simultaneous confidence bands for $\kappa^*(\mathbf{x})$. An immediate consequence of these bands' validity is the validity of the testing procedure of Section 2.3.

**Theorem 4.2.** Given the conditions and result of Theorem 4.1, we have

$$\liminf_{n, L \to \infty} \inf_{\boldsymbol{\theta} \in \Theta} \mathbb{P}_{\boldsymbol{\theta}}(\kappa^*(\mathbf{x}) \in [\widehat{D}_L(\mathbf{x}), \widehat{D}_U(\mathbf{x})], \forall \mathbf{x} \in \Omega) \geq 1 - \alpha$$

for any $\alpha \in (0, 1)$. That is, $[\widehat{D}_L(\mathbf{x}), \widehat{D}_U(\mathbf{x})]$ construct simultaneously-valid confidence bands on $\kappa^*(\mathbf{x})$. An immediate consequence is then $\limsup_{n, L \to \infty} \sup_{\boldsymbol{\theta} \in \Theta: H_0} \mathbb{P}_{\boldsymbol{\theta}}(\text{Reject } H_0, \forall \mathbf{x} \in \Omega) \leq \alpha$. See Section D.2 for a proof of these results.

### 4.3 Testing Procedure

Algorithm 2 demonstrates to construct simultaneous confidence bands on $\Delta_k(\mathbf{x})$, while Algorithm 3 outlines the hypothesis testing procedure based on a constructed null property set, $\mathcal{K}_0$. The procedure involves estimation of quantities as outlined in previous sections and also includes an explicit

discretization of $\Omega$, to approximate suprema over this space. For pointwise tests, the procedures above in a similar but simplified fashion, forgoing the suprema approximations, with validity at $\mathbf{x}'$. As presented, one can take the grids in Algorithms 2 and 3 to contain only the point of interest, $\mathbf{x}'$, and perform as written (where suprema become trivial evaluation at a single point).

---

**Algorithm 2** Estimation and Uniformly Valid Confidence Band Construction Procedure

---

**Require:** Observed data $\mathcal{D} = (\mathbf{X}, \boldsymbol{\mathcal{E}}, \mathbf{y})$, fixed point $\mathbf{x} \in \Omega$, revenue parameters $\mathbf{r}(\mathbf{x})$, level $\alpha \in (0, 1)$, regularization $\lambda \in \mathbb{R}$, bandwidth $h \in \mathbb{R}$, discretized grid $\mathcal{G} \subset \Omega$, $B \in \mathbb{N}$ resampling iterations for $W$

1: Calculate $\widehat{\boldsymbol{\theta}}(\mathbf{x}) = \operatorname{argmin}_{\boldsymbol{\beta} \in \mathbb{R}^{n+1}, \mathbf{1}^T \boldsymbol{\beta} = 0} \mathcal{L}(\boldsymbol{\beta}; \mathbf{x}, \mathcal{D}) + \frac{\lambda}{2} ||\boldsymbol{\beta}||_2^2$.

2: Debias $\widehat{\boldsymbol{\theta}}^d(\mathbf{x}) \leftarrow \widehat{\boldsymbol{\theta}}(\mathbf{x}) - \nabla^2 \mathcal{L}(\widehat{\boldsymbol{\theta}}; \mathbf{x}, \mathcal{D})^\dagger \nabla \mathcal{L}(\widehat{\boldsymbol{\theta}}; \mathbf{x}, \mathcal{D})$

3: **for** $k \in [n]$ **do**

4: $\quad$ Calculate $\widehat{\Delta}_k(\widehat{\boldsymbol{\theta}}^d; \mathbf{x}) \leftarrow \sum_{i=1}^{k} r_i(\mathbf{x}) \exp\left(\widehat{\theta}_i^d(\mathbf{x})\right) - \left[\sum_{j=0}^{k} \exp\left(\widehat{\theta}_j^d(\mathbf{x})\right)\right] r_k(\mathbf{x})$

5: **end for**

6: **for** $b \in B$ **do**

7: $\quad$ Generate standard Normal random variables $Z_{\mathcal{S}}^{\ell} \sim N(0, 1)$ for $\mathcal{S} \in \boldsymbol{\mathcal{E}}, \ell \in [L]$

8: $\quad$ **for** $k \in [n]$ **do**

9: $\quad\quad$ **for** $\mathbf{x}_g \in \mathcal{G}$ **do**

10: $\quad\quad\quad$ Calculate $w_k(\mathbf{x}_g) \leftarrow \sqrt{Lh^d}\left(\frac{-\widehat{\mathbf{v}}_k(\mathbf{x}_g)^T \nabla^2 \mathcal{L}(\widehat{\boldsymbol{\theta}}; \mathbf{x}_g)^\dagger}{\sqrt{\widehat{\mathbf{v}}_k(\mathbf{x}_g)^T \nabla^2 \mathcal{L}(\widehat{\boldsymbol{\theta}}; \mathbf{x}_g)^\dagger \widehat{\mathbf{v}}_k(\mathbf{x}_g)}} \nabla \widetilde{\mathcal{L}}(\widehat{\boldsymbol{\theta}}; \mathbf{x}_g)\right)$

11: $\quad\quad$ **end for**

12: $\quad$ **end for**

13: $\quad$ Assign $W_b \leftarrow \max_{\mathbf{x}_g \in \mathcal{G}} \max_{k \in [n]} w_k(\mathbf{x}_g)$

14: **end for**

15: Compute $\widehat{c}_W(\alpha/2; \mathcal{D}) \leftarrow$ the $1 - \alpha/2$ quantile of $W_{1:B}$

16: Calculate $\left(\widehat{D}_L(\mathbf{x}), \widehat{D}_U(\mathbf{x})\right)$ by Equations (4.3) and (4.4)

**Output:** Confidence band for $\Delta_k(\mathbf{x})$: $\left(\widehat{D}_L(\mathbf{x}), \widehat{D}_U(\mathbf{x})\right)$

---

**Algorithm 3** Testing Procedure

---

**Require:** Null hypothesis $H_0$ on $\mathcal{S}^*(\mathbf{x}')$ and corresponding $\mathcal{K}_0$

**Require:** Discretized grid $\mathcal{G} \subset \Omega$, utilities $\mathbf{r}$, remaining inputs for Algorithm 2

1: **for** $\mathbf{x}_g \in \mathcal{G}$ **do**

2: $\quad$ Re-index items at $\mathbf{x}_g$ so that $r_1(\mathbf{x}_g) \geq r_2(\mathbf{x}_g) \cdots \geq r_n(\mathbf{x}_g)$.

3: $\quad$ Run Algorithm 1 to obtain $\mathcal{S}_g^*(\mathbf{x}_g)$ and $\kappa_g^*(\mathbf{x}_g)$.

4: $\quad$ Run Algorithm 2 for fixed point $\mathbf{x}_g$. Retain $\left(\widehat{D}_L(\mathbf{x}_g), \widehat{D}_U(\mathbf{x}_g)\right)$

5: **end for**

6: **Output:** Reject $H_0$ at level if $\exists \mathbf{x}_g \in \mathcal{G}$ such that $[\widehat{D}_L(\mathbf{x}_g), \widehat{D}_U(\mathbf{x}_g)] \cap \mathcal{K}_0 = \emptyset$.

---

## 5 CONCLUSION

Our work provides a method for the estimation of and tools to perform inference on a contextually optimal assortment of items. We build these methods upon multiway comparison data from an assumed multinomial logit discrete choice model. We target the estimation of and inferential tools for a contextually optimal assortment of items. This assortment maximizes the total expected utility, for any analyst-defined, contextual utility function. Our results hold uniformly over the parameter space for the assumed discrete choice, data-generating model and simultaneously across a compact space of contextual information. Future work can consider contextual estimation for the multinomial logit choice model under constraints, mirroring the non-contextual literature, or under more general classes of discrete choice models. Importantly, the multinomial logit choice model implicitly makes the independence of irrelevant alternatives assumption. This assumption may not be palatable in certain circumstances where contextual estimation is necessary. Lastly, our work assumes the utility parameters $\mathbf{r}(\mathbf{x})$ are known constants or functions in $\mathbf{x}$.

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

# Supplementary Materials

## A SIMULATION STUDIES

In the theoretical analysis of Theorem 3.1, we assume the initialization of the gradient descent procedure, $\boldsymbol{\theta}^0(\mathbf{x})$ is $\boldsymbol{\theta}^*(\mathbf{x})$. This is a technical convenience but obviously infeasible in practice. Practically, one must only ensure that the identification condition $\mathbf{1}^T\boldsymbol{\theta}^0(\mathbf{x}) = 0$ is met for the initialization $\boldsymbol{\theta}^0(\mathbf{x})$. Empirically, we have found the estimation procedure to be insensitive to initialization, and often set $\boldsymbol{\theta}^0(\mathbf{x}) = \mathbf{0}_{n+1}$ to trivially satisfy the identification condition. Below we also take $p$, the probability with which we randomly observe our assortments, as known. In instances where this term is unknown, it is estimable from the observed assortment data.

### A.1 SIMULATION PARAMETERS

To generate the multinomial data, we fix the parameters at $n = 10$ and $p = n\log(n)/2^n$ while we vary $L$. We generate our preference parameters $\boldsymbol{\theta}$ by $\widetilde{\theta}_j(\mathbf{x}) = g(\mathbf{x}_j, \beta_j)$ and assign $\boldsymbol{\theta}^*(\mathbf{x}) = \widetilde{\boldsymbol{\theta}}(\mathbf{x}) - \mathbf{1}^T\widetilde{\boldsymbol{\theta}}(\mathbf{x})/(n+1)$ to enforce the identification constraint. As an instance of such a function, we take $g(\cdot)$ to be linear, $\theta_j(\mathbf{x}) = \beta_j^T\mathbf{x}$ and sample $\beta_j \sim \mathrm{Uniform}(0, 2)$. We fixed a target context $\mathbf{x} \sim \mathrm{Uniform}(0, 1)$. We sampled revenue parameters $r_j \sim \mathrm{Uniform}(0.2, 3), j = 1, \ldots, n$ and assigned indices to $r, \boldsymbol{\theta}$ such that $r_1(\mathbf{x}) \geq r_2(\mathbf{x}) \geq r_3(\mathbf{x})$. We assigned $r_0(\mathbf{x}) = 0$ to the "no-decision" 0th option for any assortment.

Lastly for the hyperparameters of the method in Algorithm 2 set $\lambda = c(\sqrt{\log(2^n pLh^{-d})/2^n pLh^d} + dh^2)$ for tuning constant $c > 0$. We use the plug-in estimator $h = c\big(\log(2^n pL)/2^n pL\big)^{1/4+d}$ for $c = 0.2$ with good performance, although this constant is also tunable. Results below are shown from 100 Monte Carlo simulations.

### A.2 SIMULATION RESULTS

We assessed performance of our estimation procedure for both $\widehat{\boldsymbol{\theta}}(\mathbf{x})$ by a normalized, squared error: $||\widehat{\boldsymbol{\theta}}(\mathbf{x}) - \boldsymbol{\theta}^*(\mathbf{x})||_2/||\boldsymbol{\theta}^*(\mathbf{x})||_2$. We also include results for $\widehat{\Delta}(\mathbf{x}) = [\widehat{\Delta}_1(\mathbf{x}) \quad \ldots \quad \widehat{\Delta}_n(\mathbf{x})]^T$ by an analogous, normalized squared error. By the form of Algorithm 1, we also display the proportion of $\widehat{\Delta}_k(\mathbf{x})$ sharing the same sign as $\Delta_k(\mathbf{x})$, averaged across indices: $\sum_{\mathcal{S} \in \mathcal{E}} \sum_{k \in \mathcal{S}} \mathbb{1}(\widehat{\Delta}_k\Delta_k > 0)$, which was then averaged across 100 Monte Carlo simulations. Mirroring our theoretical results, we observe improved performance across increasing repeated samples per-assortment ($L$ on the x-axes) for both our preference parameters, $\widehat{\boldsymbol{\theta}}$, and marginal utility gap, $\widehat{\Delta}_k$, objects. Sign-matching of $\widehat{\Delta}_k, \Delta_k$ is contained in the right-hand graph of Figure 1, identifying that universally our estimated utility gaps match the sign of the true utility gaps for our simulated data. This corresponds to the inclusion rule and appropriate optimal assortment selection, as outlined by the procedure in Algorithm 1.

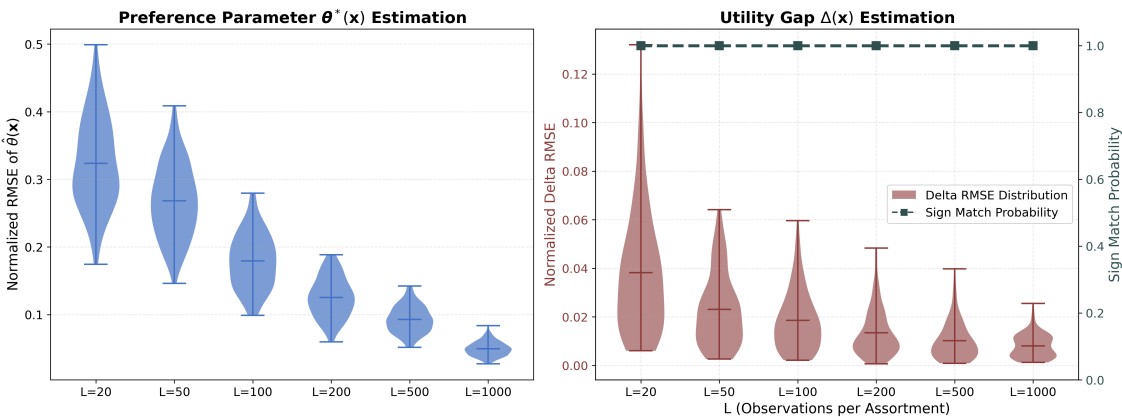

Figure 1: Simulation Results for Estimation of $\boldsymbol{\theta}^*(\mathbf{x})$, $\{\Delta_k(\mathbf{x})\}_{k\in[n]}$.

## B   PROOF OF THEOREM 3.1

Our proof is an adaptation of Shen et al. (2023) and Chen et al. (2019) Briefly, we propose a gradient descent optimization procedure to estimate $\widehat{\boldsymbol{\theta}}(\mathbf{x})$ at a fixed $\mathbf{x} \in \Omega$, which can be shown to converge by the smoothness and strong-convexity of the loss function.

Recall that $\mathcal{S}$ represents the set of all $2^n$ possible assortments, with all assortments including the "no-decision" option. We represent the subset of observed sets from $\mathcal{S}$ as $\mathcal{E}$. We also make use of the following notation

$$\lambda_{\min,\perp}(\mathbf{M}) = \min\{\lambda : \mathbf{v}^T\mathbf{M}\mathbf{v} \geq \lambda, \forall \mathbf{v} \text{ such that } \|\mathbf{v}\|_2 = 1, \mathbf{v}^T\mathbf{1} = 0\} .$$

Now recall our estimator

$$\widehat{\boldsymbol{\theta}}(\mathbf{x}) = \underset{\boldsymbol{\beta}\in R^{n+1},\mathbf{1}^T\boldsymbol{\beta}=0}{\operatorname{argmin}} \mathcal{L}(\boldsymbol{\beta}; \mathbf{x}, \mathcal{D}) + \frac{\lambda}{2}||\boldsymbol{\beta}||_2^2 \tag{B.1}$$

where $\mathcal{L}(\boldsymbol{\beta}; \mathbf{x}, \mathcal{D})$ is the regularized, local approximation of the negative log-likelihood of the multinomial logit decision model.

### B.1   LEMMAS FOR PROOF OF THEOREM 3.1

Before evaluating $\widehat{\boldsymbol{\theta}}(\mathbf{x})$, we first develop a series of useful lemmas. See Section E for proofs of the following.

**Lemma B.1** (Laplacian Eigenvalue Bounds). Suppose there exists a large enough constant $C > 0$ such that $2^n p \geq Cn \log n$. Then for matrix

$$\mathbf{L}_{\boldsymbol{\mathcal{E}}}(\mathbf{x}) = \sum_{\mathcal{S}\in\mathcal{E}} \sum_{\ell\in[L]} \sum_{i\in\mathcal{S}_+} L^{-1} K_{\mathbf{H}}\left(\mathbf{X}_{\mathcal{S}}^\ell - \mathbf{x}\right) \left(\sum_{k\in\mathcal{S}_+;\, k>i} \frac{1}{|\mathcal{S}_+|^2}(\mathbf{e}_i - \mathbf{e}_k)(\mathbf{e}_i - \mathbf{e}_k)^T\right) , \tag{B.2}$$

the inequalities

$$\sup_{\mathbf{x}\in\Omega} \lambda_{\max}(\mathbf{L}_{\boldsymbol{\mathcal{E}}}) \lesssim 2^n p\sqrt{\frac{d\log(2^npLh^{-d})}{2^npLh^d}} + 2^npdh^2 + C_f 2^np$$

$$\inf_{\mathbf{x}\in\Omega} \lambda_{\min,\perp}(\mathbf{L}_{\boldsymbol{\mathcal{E}}}) \gtrsim \frac{1}{2}(c_f + cdh^2)2^np/n - 2^np\sqrt{\frac{d\log(2^npLh^{-d})}{2^npLh^d}}$$

hold jointly with probability at least $1 - O(n^{-10})$.

**Lemma B.2** (Hessian Eigenvalue Bounds). For any $\boldsymbol{\beta} \in \mathbb{R}^{n+1}$ such that $\sup_{\mathbf{x} \in \Omega} \|\boldsymbol{\beta} - \boldsymbol{\theta}^*(\mathbf{x})\|_\infty < C_1$ for large enough constant $C_1 > 0$, then both

$$\sup_{\mathbf{x} \in \Omega} \lambda_{\max} \left( \nabla^2 \mathcal{L}_\lambda(\boldsymbol{\beta}; \mathbf{x}, \mathcal{D}) \right) \lesssim \lambda + \left( c_\theta e^{2C_1} \right)^2 \sup_{\mathbf{x} \in \Omega} \lambda_{\max} \left( \mathbf{L}_{\boldsymbol{\mathcal{E}}} \right)$$

and

$$\inf_{\mathbf{x} \in \Omega} \lambda_{\min, \perp} \left( \nabla^2 \mathcal{L}_\lambda(\boldsymbol{\beta}; \mathbf{x}, \mathcal{D}) \right) \gtrsim \lambda + \frac{1}{2(c_\theta e^{2C_1})^2} \inf_{\mathbf{x} \in \Omega} \lambda_{\min} \left( \mathbf{L}_{\boldsymbol{\mathcal{E}}} \right)$$

hold under the event of Lemma B.1.

**Lemma B.3.** If $2^n p \geq Cn \log n$ for some sufficiently large $C > 0$ and selecting $\lambda \asymp 2^n p \sqrt{\frac{d \log(2^n p L h^{-d})}{2^n p L h^d}} + 2^n p d h^2$, then

$$\sup_{\mathbf{x} \in \Omega} \|\nabla \mathcal{L}(\boldsymbol{\theta}^*; \mathbf{x}, \mathcal{D})\|_2 \lesssim 2^n p \sqrt{n \frac{d \log(2^n p L h^{-d})}{2^n p L h^d}} + 2^n p C d h^2 \sqrt{n}$$

holds with probability at least $1 - O(n^{-10})$.

**Lemma B.4.** The event

$$\||\nabla^2 \mathcal{L}_\lambda(\boldsymbol{\beta}; \mathbf{x}, \mathcal{D})\||_2 \leq \lambda + C_f 2^n np + 2^n npdh^2 + 2^n np \sqrt{\frac{d \log(2^n p L h^{-d})}{2^n p L h^d}} .$$

holds with probability at least $1 - O(n^{-10})$ for any $\boldsymbol{\beta} \in \mathbb{R}^{n+1}$.

**Lemma B.5.** Under the event of Lemma B.3, for some constant $C > 0$ and selection of $\lambda \asymp 2^n p \sqrt{\frac{d \log(2^n p L h^{-d})}{2^n p L h^d}} + 2^n p d h^2$, then

$$\sup_{\mathbf{x} \in \Omega} \|\widehat{\boldsymbol{\theta}}(\mathbf{x}) - \boldsymbol{\theta}^*(\mathbf{x})\|_2 \lesssim \sqrt{n} .$$

**Remark B.6.** Note that as $\nabla^2 \mathcal{L}_\lambda(\boldsymbol{\theta}; \mathbf{x}, \mathcal{D})$ is positive semi-definite (for any $\boldsymbol{\theta}, \mathbf{x}$), the result in Lemma B.4 is equivalently a bound on $\lambda_{\max} \left( \nabla^2 \mathcal{L}_\lambda \right)$. The result differs from Lemma B.2 by forgoing the bounded $\ell^\infty$ norm condition.

## B.2 PROOF OF ESTIMATION RATE

Now we begin the proof of Theorem 3.1. As in Chen et al. (2019) and Shen et al. (2023), we identify the estimation rate by analyzing a gradient descent procedure. Assume that we update our estimate at a given context $\mathbf{x} \in \Omega$

$$\boldsymbol{\theta}^{t+1}(\mathbf{x}) = \boldsymbol{\theta}^t(\mathbf{x}) - \eta \nabla \mathcal{L}_\lambda(\boldsymbol{\theta}^t; \mathbf{x}, \mathcal{D}) \tag{B.3}$$

over iterations $t = 0, \cdots, T$. Assume initialization at $\boldsymbol{\theta}^0(\mathbf{x}) = \boldsymbol{\theta}^*(\mathbf{x})$. Note that this initialization satisfies our identification condition immediately, $\mathbf{1}^T \boldsymbol{\theta}^0(\mathbf{x}) = 0$. We also have $\mathbf{1}^T \lambda \mathcal{L}_\lambda(\boldsymbol{\theta}; \mathbf{x}, \mathcal{D}) = 0$ for any $(\boldsymbol{\theta}, \mathbf{x}, \mathcal{D})$. Thus the gradient descent procedure in Equation B.3 will result in $\mathbf{1}^T \boldsymbol{\theta}^t(\mathbf{x}) = 0$ for any $\mathbf{x} \in \Omega, t \geq 0$. Thus the necessary identification condition is also guaranteed for $\widehat{\boldsymbol{\theta}}(\mathbf{x})$ estimated via this procedure.

Now, set a learning rate at

$$\eta \gtrsim \left( 2^n pn + 2^n pndh^2 + 2^n pn\nu_h + \lambda \right)^{-1}$$

where we define $\nu_h := \sqrt{\frac{d \log(2^n p L h^{-d})}{2^n p L h^d}}$ for brevity. We now decompose the estimation error as

$$\sup_{\mathbf{x} \in \Omega} \|\widehat{\boldsymbol{\theta}}(\mathbf{x}) - \boldsymbol{\theta}^*(\mathbf{x})\|_2 \leq \sup_{\mathbf{x} \in \Omega} \|\widehat{\boldsymbol{\theta}}(\mathbf{x}) - \boldsymbol{\theta}^T(\mathbf{x})\|_2 + \sup_{\mathbf{x} \in \Omega} \|\boldsymbol{\theta}^T(\mathbf{x}) - \boldsymbol{\theta}^*(\mathbf{x})\|_2 .$$

### B.2.1 $||\boldsymbol{\theta}^T(\mathbf{x}) - \widehat{\boldsymbol{\theta}}(\mathbf{x})||_2$

We first construct an error bound on $\sup_{\mathbf{x}\in\Omega}||\boldsymbol{\theta}^T(\mathbf{x}) - \widehat{\boldsymbol{\theta}}(\mathbf{x})||_2$. Trivially $\mathcal{L}_\lambda(\boldsymbol{\theta};\mathbf{y};\mathbf{x})$ is $\lambda$-strongly convex and by Lemma B.4 is $\lambda + n2^n p\nu_h + c2^n npdh^2 + C_f 2^n np$-smooth with probability at least $1 - O(n^{-10})$, for any $\mathbf{x}\in\Omega$. Conditional on this event, we then use Theorem 3.10 from Bubeck (2015) to study the gradient descent procedure's convergence to the minimizer of Equation 3.3.

$$\sup_{\mathbf{x}\in\Omega}||\boldsymbol{\theta}^T(\mathbf{x}) - \widehat{\boldsymbol{\theta}}(\mathbf{x})||_2 \le \exp\left(-\frac{T\lambda}{\lambda + n2^n p(\nu_h + dh^2 + c)}\right)\sup_{\mathbf{x}\in\Omega}||\widehat{\boldsymbol{\theta}}(\mathbf{x}) - \boldsymbol{\theta}^*(\mathbf{x})||_2$$

$$\lesssim \frac{1}{T}\left(\frac{\lambda + n2^n p(\nu_h + dh^2 + c)}{\lambda}\right)\sqrt{n}$$

$$\lesssim \frac{1}{T}\left(1 + n\left(\frac{2^n p\nu_h + 2^n pdh^2}{2^n p\nu_h + 2^n pdh^2}\right) + \frac{2^n np}{2^n p\nu_h + 2^n pdh^2}\right)\sqrt{n}$$

$$\lesssim \frac{\sqrt{n^3}}{T} + \frac{\sqrt{n^3}}{T(\nu_h + pdh^2)}$$

$$\lesssim \frac{\nu_h + dh^2}{\sqrt{2^n Lp}} + \frac{1}{\sqrt{2^n Lp}}$$

The second and third inequalities apply Lemma B.5 and the selection of our regularization parameter $\lambda \asymp 2^n p\sqrt{\frac{d\log(2^n pLh^{-d})}{2^n pLh^d}} + 2^n pdh^2$. The final inequality then holds assuming sufficiently large $T$, specifically $T \gtrsim \frac{\sqrt{2^n pLn^3}}{\nu_h + dh^2}$.

### B.2.2 $||\boldsymbol{\theta}^T(\mathbf{x}) - \boldsymbol{\theta}^*(\mathbf{x})||_2$

We now study the closeness of the gradient descent procedure's output to $\boldsymbol{\theta}^*(\mathbf{x})$. TO this end, we will argue by induction that

$$\sup_{\mathbf{x}\in\Omega}||\boldsymbol{\theta}^T(\mathbf{x}) - \boldsymbol{\theta}^*(\mathbf{x})||_2 \lesssim \sqrt{n}\left(\frac{\nu_h + cdh^2}{\nu_h + cdh^2 + C}\right).$$

The base case holds trivially under $\boldsymbol{\theta}^0(\mathbf{x}) = \boldsymbol{\theta}^*(\mathbf{x})$. Assume then for iteration $t$,

$$\sup_{\mathbf{x}\in\Omega}||\boldsymbol{\theta}^t(\mathbf{x}) - \boldsymbol{\theta}^*(\mathbf{x})||_2 \lesssim \sqrt{n}\left(\frac{\nu_h + cdh^2}{\nu_h + cdh^2 + C}\right). \tag{B.4}$$

We wish to show the same inequality holds for $\boldsymbol{\theta}^{t+1}(\mathbf{x})$ to conclude the argument.

Now let $\boldsymbol{\theta}_\tau(\mathbf{x}) = \boldsymbol{\theta}^*(\mathbf{x}) + \tau[\boldsymbol{\theta}^t(\mathbf{x}) - \boldsymbol{\theta}^*(\mathbf{x})]$ for $\tau\in[0,1]$. The update rule at our given context $\mathbf{x}$ can be written as

$$\boldsymbol{\theta}^{t+1}(\mathbf{x}) - \boldsymbol{\theta}^*(\mathbf{x}) = \boldsymbol{\theta}^t(\mathbf{x}) - \eta\nabla\mathcal{L}_\lambda(\boldsymbol{\theta}^t;\mathbf{x},\mathcal{D}) - \boldsymbol{\theta}^*(\mathbf{x})$$

$$= \boldsymbol{\theta}^t(\mathbf{x}) - \eta\nabla\mathcal{L}_\lambda(\boldsymbol{\theta}^t;\mathbf{x},\mathcal{D}) - [\boldsymbol{\theta}^*(\mathbf{x}) - \eta\nabla\mathcal{L}_\lambda(\boldsymbol{\theta}^*;\mathbf{x},\mathcal{D})] - \eta\nabla\mathcal{L}_\lambda(\boldsymbol{\theta}^*;\mathbf{x},\mathcal{D})$$

$$= \left(\mathbf{I}_n - \eta\underbrace{\int_0^1 \nabla^2\mathcal{L}_\lambda(\boldsymbol{\theta}_\tau;\mathbf{x},\mathcal{D})\,\mathrm{d}\tau}_{:=\mathbf{A}}\right)[\boldsymbol{\theta}^t(\mathbf{x}) - \boldsymbol{\theta}^*(\mathbf{x})] - \eta\nabla\mathcal{L}_\lambda(\boldsymbol{\theta}^*;\mathbf{x},\mathcal{D}).$$

Then

$$\sup_{\mathbf{x}\in\Omega}||\boldsymbol{\theta}^{t+1}(\mathbf{x}) - \boldsymbol{\theta}^*(\mathbf{x})||_2 \le \sup_{\mathbf{x}\in\Omega}\left||\left(\mathbf{I}_n - \eta\mathbf{A}\right)\left(\boldsymbol{\theta}^t(\mathbf{x}) - \boldsymbol{\theta}^*(\mathbf{x})\right)\right||_2 + \eta\sup_{\mathbf{x}\in\Omega}||\nabla\mathcal{L}_\lambda\left(\boldsymbol{\theta}^*(\mathbf{x});\mathcal{D}\right)||_2$$

Note then that $\mathbf{1}^T(\boldsymbol{\theta}^t(\mathbf{x}) - \boldsymbol{\theta}^*(\mathbf{x})) = 0$ for any $t\in[T]_+$, since $\mathbf{1}^T\nabla\mathcal{L}(\boldsymbol{\theta},\mathcal{D}) = \mathbf{0}$ as previously noted by our update rule in Equation B.3. We can thus control the first term as

$$\left||\left(\mathbf{I}_n - \eta\mathbf{A}\right)\left(\boldsymbol{\theta}^t(\mathbf{x}) - \boldsymbol{\theta}^*(\mathbf{x})\right)\right||_2 \le \max\left\{|1 - \eta\lambda_{\min,\perp}(\mathbf{A})|, |1 - \eta\lambda_{\max}(\mathbf{A})|\right\}||\boldsymbol{\theta}^t(\mathbf{x}) - \boldsymbol{\theta}^*(\mathbf{x})||_2.$$

The condition of Lemma B.2 that $\sup_{\mathbf{x}\in\Omega}\|\boldsymbol{\theta}(\mathbf{x}) - \boldsymbol{\theta}^*(\mathbf{x})\|_\infty < c$ is satisfied by the induction assumption in Equation B.4. By Lemma B.2 and selection of $\lambda$, we then have

$$\inf_{\mathbf{x}\in\Omega}\lambda_{\min,\perp}(\nabla^2\mathcal{L}_\lambda) \ge c2^n p\nu_h + C2^n pdh^2 + \frac{1}{2(c_\theta e^{2c})^2}\left[\frac{1}{2}(c_f + cdh^2)2^n p/n - 2^n p\nu_h\right]$$

$$\gtrsim 2^n pdh^2 + \frac{1}{2}(c_f + cdh^2)2^n p/n$$

with probability at least $1 - O(n^{-10})$, for $\lambda$ selected with sufficiently large constant scaling $C > 1/2(c_\theta e^{2c})^2$. Then, by choice of our learning rate $\eta \asymp \left(c2^n np + 2^n npdh^2 + 2^n pn\nu_h + \lambda\right)^{-1}$ with appropriate constants, we have $0 \leq \eta\lambda_{\min,\perp}(\mathbf{A}) \leq \eta\lambda_{\max}(\mathbf{A}) \leq 1$. As a result,

$$\sup_{\mathbf{x}\in\Omega} ||\boldsymbol{\theta}(\mathbf{x})^{t+1} - \boldsymbol{\theta}^*(\mathbf{x})||_2 \leq \sup_{\mathbf{x}\in\Omega} ||\boldsymbol{\theta}(\mathbf{x})^t - \boldsymbol{\theta}^*(\mathbf{x})||_2 + \eta \sup_{\mathbf{x}\in\Omega} ||\nabla\mathcal{L}_\lambda(\boldsymbol{\theta}^*; \mathcal{D})||_2$$

$$\lesssim \sqrt{n}\left(\frac{\nu_h + cdh^2}{\nu_h + cdh^2 + C}\right)$$

holds with probability at least $1 - O(n^{-10})$ by Lemma B.3.

### B.2.3 $||\widehat{\boldsymbol{\theta}}(\mathbf{x}) - \boldsymbol{\theta}^*(\mathbf{x})||_2$

Returning to our original decomposition, we have that

$$\sup_{\mathbf{x}\in\Omega} ||\widehat{\boldsymbol{\theta}}(\mathbf{x}) - \boldsymbol{\theta}^*(\mathbf{x})||_2 \leq \sup_{\mathbf{x}\in\Omega} ||\widehat{\boldsymbol{\theta}}(\mathbf{x}) - \boldsymbol{\theta}^T(\mathbf{x})||_2 + \sup_{\mathbf{x}\in\Omega} ||\boldsymbol{\theta}^T(\mathbf{x}) - \boldsymbol{\theta}^*(\mathbf{x})||_2$$

$$\lesssim \frac{\nu_h + dh^2}{\sqrt{2^n Lp}} + \frac{1}{\sqrt{2^n Lp}} + \sqrt{n}(\nu_h + dh^2)$$

$$\lesssim \frac{1}{\sqrt{2^n Lp}} + \sqrt{n}(\nu_h + dh^2)$$

$$\lesssim \sqrt{n}(\nu_h + dh^2)$$

holds with probability at least $1 - O(n^{-10})$ when $2^n p \gtrsim n \log n$. The final inequality makes use of the form of $\nu_h$ and assumption $h = o(1)$, such that

$$\nu_h := \sqrt{\frac{d\log(2^n pLh^{-d})}{2^n pLh^d}} \gtrsim \sqrt{\frac{1}{2^n pL}}$$

Thus the $\sqrt{n}\nu_h$ term dominates $1/\sqrt{2^n Lp}$ in the final rate. The statement holds in high probability, with randomness coming from the events in the previously derived technical lemmas. These include the smoothness and strong-convexity of $\nabla^2\mathcal{L}_\lambda(\boldsymbol{\theta}; \mathbf{x}, \mathcal{D})$ and control of $||\nabla\mathcal{L}(\boldsymbol{\theta}; \mathbf{x}, \mathcal{D})||_2$.

## C    PROOFS FOR DEBIASING

### C.1    PRELIMINARY RESULTS

**Remark C.1.** We first note the simplification of previously stated results, based on the selection of $\lambda, h$. By selection of $\lambda \asymp 2^n p \sqrt{\frac{d \log(2^n pLh^{-d})}{2^n pLh^d}} + 2^n pdh^2$, the result of Lemma B.2 can be restated as

$$\sup_{\mathbf{x} \in \Omega} \lambda_{\max}(\nabla^2 \mathcal{L}) \lesssim 2^n p\nu_h + 2^n pdh^2 + C_f 2^n p$$

$$\inf_{\mathbf{x} \in \Omega} \lambda_{\min,\perp}(\nabla^2 \mathcal{L}) \geq \frac{1}{2}(c_f + cdh^2)2^n p/n - 2^n p\nu_h$$

Note that these results are stated for the Hessian of the likelihood without regularization, $\nabla^2 \mathcal{L}$. Furthermore for $h$ selected as in Remark 3.2, the lower bound further simplifies to $\inf_{\mathbf{x} \in \Omega} \lambda_{\min,\perp}(\nabla^2 \mathcal{L}) \gtrsim 2^n p/n$, given $\nu_h + dh^2 = o(1)$ for sufficiently large $n, L$.

For brevity, let $\varphi = d\left(\frac{\log(2^n pL)}{2^n pL}\right)^{2/4+d}$ represent the estimation rate in Remark 3.2. Previous results then simplify to

$$\nu_h + dh^2 \lesssim \varphi \quad \text{and} \quad \sup_{\mathbf{x} \in \Omega} ||\widehat{\boldsymbol{\theta}}(\mathbf{x}) - \boldsymbol{\theta}(\mathbf{x})||_2 \lesssim \sqrt{n}\, \varphi$$

with the latter statement still holding in high probability under the conditions stated in Theorem 3.1. Similarly, Lemma B.3 also simplifies as

$$||\nabla \mathcal{L}_\lambda(\boldsymbol{\theta}^*; \mathbf{x}, \mathcal{D})||_2 \lesssim 2^n p\sqrt{n}\sqrt{\frac{d \log(2^n pLh^{-d})}{2^n pLh^d}} + 2^n p\sqrt{n}Cdh^2 \lesssim 2^n p\, \varphi$$

given $\nu_h + dh^2 \lesssim \varphi$.

We now control a series of objects that appear in the de-biasing of $\widehat{\boldsymbol{\theta}}(\mathbf{x})$ and decomposition of $\widehat{\Delta}_k(\mathbf{x}) - \Delta(\mathbf{x})$. We intermittently use the shortened notation $\nabla \mathcal{L}(\boldsymbol{\theta}) \equiv \nabla \mathcal{L}(\boldsymbol{\theta}; \mathbf{x}, \mathcal{D})$ and similarly $\nabla^2 \mathcal{L}(\boldsymbol{\theta}) \equiv \nabla^2 \mathcal{L}(\boldsymbol{\theta}; \mathbf{x}, \mathcal{D})$.

**Lemma C.2.** Suppose it holds that $2^n p \geq Cn \log(n)$ and $\sup_{\mathbf{x} \in \Omega} ||\widehat{\boldsymbol{\theta}}(\mathbf{x}) - \boldsymbol{\theta}^*(\mathbf{x})||_\infty \leq c$ for some constants $c, C > 0$. Assume selection of $h$ as in Remark 3.2 and let $\varphi = d\left(\log(2^n pL)/2^n pL\right)^{2/4+d}$. Then the following series of inequalities holds jointly with probability at least $1 - O(n^{-10})$:

$$\mathbf{B}_1 := \sup_{\mathbf{x} \in \Omega} ||\nabla \mathcal{L}(\boldsymbol{\theta}^*)||_\infty \lesssim 2^n p\, \varphi \tag{C.1}$$

$$\mathbf{B}_2 := \sup_{\mathbf{x} \in \Omega} \left|\left|\nabla \mathcal{L}(\widehat{\boldsymbol{\theta}}) - \nabla \mathcal{L}(\boldsymbol{\theta}^*) - \nabla^2 \mathcal{L}(\boldsymbol{\theta}^*)(\widehat{\boldsymbol{\theta}}(\mathbf{x}) - \boldsymbol{\theta}^*(\mathbf{x}))\right|\right|_\infty \lesssim 2^n p\, n\, \varphi^2(1 + \varphi)$$

$$\mathbf{B}_3 := \sup_{\mathbf{x} \in \Omega} ||\nabla^2 \mathcal{L}(\widehat{\boldsymbol{\theta}}) - \nabla^2 \mathcal{L}(\boldsymbol{\theta}^*; \mathbf{x}, \mathcal{D})||_\infty \lesssim 2^n p\sqrt{n}\, \varphi(1 + \varphi)$$

$$\mathbf{B}_4 := \sup_{\mathbf{x} \in \Omega} \left|\left|\begin{pmatrix} \nabla^2 \mathcal{L}(\widehat{\boldsymbol{\theta}}) & \mathbf{1} \\ \mathbf{1}^T & 0 \end{pmatrix}^{-1} - \begin{pmatrix} \nabla^2 \mathcal{L}(\boldsymbol{\theta}^*) & \mathbf{1} \\ \mathbf{1}^T & 0 \end{pmatrix}^{-1}\right|\right|_2 \lesssim \frac{n^2}{2^n p}(1 + \varphi)\varphi$$

$$\mathbf{B}_5 := \sup_{\mathbf{x} \in \Omega} ||\nabla^2 \mathcal{L}(\boldsymbol{\theta})^\dagger||_2 \leq \frac{n}{2^n p}$$

Note that the condition $\sup_{\mathbf{x}} ||\widehat{\boldsymbol{\theta}}(\mathbf{x}) - \boldsymbol{\theta}^*(\mathbf{x})||_\infty \leq c$ distills to the assumption $n \log(2^n pL) \lesssim (2^n pL)^{4/4+d}$ of Theorem 3.1. The bound for $\mathbf{B}_1$ is argued within the proof of Lemma B.3 in Section E.3. The remaining inequalities are proven in Section F.

### C.2    PROOF OF LEMMA 3.3

*Proof.* By the proof of Lemma B.3 in Shen et al. (2023), under our stated conditions, we have the decomposition

$$\widehat{\boldsymbol{\theta}}^d(\mathbf{x}) - \boldsymbol{\theta}^*(\mathbf{x}) = -\nabla^2 \mathcal{L}(\boldsymbol{\theta}^*; \mathbf{x}, \mathcal{D})^\dagger \nabla \mathcal{L}(\boldsymbol{\theta}^*; \mathbf{x}, \mathcal{D}) + \mathbf{R}_\theta$$

for remainder $\mathbf{R}_\theta$ such that

$$\sup_{\mathbf{x}\in\Omega}||\mathbf{R}_\theta(\mathbf{x})||_2 \lesssim \sqrt{n}\cdot\mathbf{B}_4\Big(\mathbf{B}_2+\mathbf{B}_1\Big)+\mathbf{B}_5\mathbf{B}_2\ . \tag{C.2}$$

Let $\varphi = [d\log(2^npL)/(2^npL)]2/4+d$, noting $\varphi = o(1)$ under the stated conditions. We then analyze the two components of Equation C.2 by applying results from Lemma C.2:

$$\mathbf{B}_5\mathbf{B}_2 \lesssim \frac{n^2}{2^np}(2^np\varphi^2+2^np\varphi^3) \le n^2(\varphi^2+\varphi^3) \lesssim (n\varphi)^2$$

$$\begin{aligned}\mathbf{B}_4(\mathbf{B}_2+\mathbf{B}_1) &\lesssim \frac{n^{5/2}}{2^np}\varphi(1+\varphi)(2^np\varphi^2(1+\varphi)+2^np\varphi)\\ &\le n^{5/2}(\varphi+\varphi^2)(\varphi+\varphi^2+\varphi^3)\\ &\lesssim n^{5/2}\varphi^2\end{aligned}$$

with the latter two inequalities holding by our scaling assumptions, such that $\varphi = o(1)$. Substituting these upper-bounds into Equation C.2 completes the argument.

$\square$

## C.3 STATEMENT AND PROOF OF $\widehat{\Delta}_k$ DEBIASING

**Lemma C.3** (Debiasing of $\widehat{\Delta}_k$)**.** Suppose $\widehat{\Delta}_k$ is calculated using the de-biased $\widehat{\boldsymbol{\theta}}^d(\mathbf{x})$. Let $\widetilde{\boldsymbol{\theta}}(\mathbf{x})$ lie between $\widehat{\boldsymbol{\theta}}(\mathbf{x})$ and $\boldsymbol{\theta}^*(\mathbf{x})$. Then

$$\widehat{\Delta}_k(\mathbf{x})-\Delta_k(\mathbf{x}) = \mathbf{v}_k(\boldsymbol{\theta}^*,\mathbf{x})^T\Big(-\nabla^2\mathcal{L}(\boldsymbol{\theta}^*;\mathbf{x},\mathcal{D})^\dagger\nabla\mathcal{L}(\boldsymbol{\theta}^*;\mathbf{x},\mathcal{D})+\mathbf{R}_\theta(\mathbf{x})\Big)+R_\Delta(\mathbf{x})$$

$$\text{for } |R_\Delta(\mathbf{x})| \lesssim n^6d\left(\frac{\log(2^npL)}{2^npL}\right)^{4/4+d}||\mathbf{v}_k(\boldsymbol{\theta}^*,\mathbf{x})||_\infty$$

with probability at least $1-O(n^{-10})$ under the conditions of Lemma C.2. Note that it is not immediately obvious that $||\mathbf{v}_k(\boldsymbol{\theta}^*,\mathbf{x})||_\infty$ is a well-controlled term. However, we will encounter $\widehat{\Delta}_k(\mathbf{x})-\widehat{\Delta}(\mathbf{x})$ in ratios with normalizing terms, which will permit tractable upper bounds.

*Proof.* By the definition of $\Delta_k(\mathbf{x})$ and Taylor expansion, we have

$$\widehat{\Delta}_k(\mathbf{x})-\Delta_k(\mathbf{x}) = \mathbf{v}_k(\boldsymbol{\theta}^*;\mathbf{x})^T\Big(-\nabla^2\mathcal{L}(\boldsymbol{\theta}^*;\mathbf{x},\mathcal{D})^\dagger\nabla\mathcal{L}(\boldsymbol{\theta}^*;\mathbf{x},\mathcal{D})+\mathbf{R}_d(\mathbf{x})\Big)+R_\Delta,$$

$$R_\Delta := \frac{1}{2}\big(\widehat{\boldsymbol{\theta}}^d(\mathbf{x})-\boldsymbol{\theta}^*(\mathbf{x})\big)^T\operatorname{diag}\Big(\mathbf{v}(\boldsymbol{\theta}^*;\mathbf{x})\Big)\big(\widehat{\boldsymbol{\theta}}^d(\mathbf{x})-\boldsymbol{\theta}^*(\mathbf{x})\big)+o(||\widehat{\boldsymbol{\theta}}(\mathbf{x})-\boldsymbol{\theta}^*(\mathbf{x})||_2^2)$$

Applying Corollary 3.4 provides

$$\sup_{\mathbf{x}\in\Omega}|R_\Delta| \lesssim \sup_{\mathbf{x}\in\Omega}||\mathbf{v}(\widetilde{\boldsymbol{\theta}};\mathbf{x})||_\infty\sup_{\mathbf{x}\in\Omega}||\widehat{\boldsymbol{\theta}}^d(\mathbf{x})-\boldsymbol{\theta}^*(\mathbf{x})||_2^2 \le n^6d\left(\frac{\log(2^npL)}{2^npL}\right)^{4/4+d}\sup_{\mathbf{x}\in\Omega}||\mathbf{v}(\boldsymbol{\theta}^*;\mathbf{x})||_\infty$$

with probability at least $1-O(n^{-10})$.

$\square$

## D  PROOFS FOR INFERENTIAL PROCEDURE

**Lemma D.1.** Let $\varphi = d \left( \frac{\log(2^n pL)}{2^n pL} \right)^{2/4+d}$. The following inequalities hold jointly with probability at least $1 - O(n^{-10})$ for large enough $n, L$ and $\sqrt{n}\varphi = o(1)$:

$$\left( \mathbf{v}_k(\mathbf{x})^T \nabla^2 \mathcal{L}(\boldsymbol{\theta}^*)^\dagger \mathbf{v}_k(\mathbf{x}) \right) \geq \frac{||\mathbf{P}_{\mathbf{1}}^\perp \mathbf{v}_k(\mathbf{x})||_2^2}{\lambda_{\min,\perp}(\nabla^2 \mathcal{L}(\boldsymbol{\theta}^*))} \gtrsim \frac{n \inf_{\mathbf{x} \in \Omega} ||\mathbf{P}_{\mathbf{1}}^\perp \mathbf{v}_k(\mathbf{x})||_2^2}{2^n p} \tag{D.1}$$

$$\sup_{\mathbf{x} \in \Omega} \max_{k \in [n]} \frac{||\mathbf{v}_k(\mathbf{x})||_\infty}{||\mathbf{P}_{\mathbf{1}}^\perp \mathbf{v}_k(\mathbf{x})||_2} \leq c_r c_\theta \tag{D.2}$$

$$||\widehat{\mathbf{v}}_k(\mathbf{x}) - \mathbf{v}_k(\mathbf{x})||_2 \lesssim \varphi \sqrt{n} ||\mathbf{v}_k(\mathbf{x})||_\infty \tag{D.3}$$

$$||\widehat{\mathbf{v}}_k(\mathbf{x})^T \nabla^2 \mathcal{L}(\widehat{\boldsymbol{\theta}})^\dagger - \mathbf{v}_k(\mathbf{x})^T \nabla^2 \mathcal{L}(\boldsymbol{\theta}^*)^\dagger||_2 \lesssim \left( n^{3/2}\varphi + n^2 \varphi ||\mathbf{P}_{\mathbf{1}}^\perp \mathbf{v}_k(\mathbf{x})||_2 \right)/2^n p \tag{D.4}$$

$$\left| \widehat{\mathbf{v}}_k(\mathbf{x})^T \nabla^2 \mathcal{L}(\widehat{\boldsymbol{\theta}})^\dagger \widehat{\mathbf{v}}_k(\mathbf{x}) - \mathbf{v}_k(\mathbf{x})^T \nabla^2 \mathcal{L}(\boldsymbol{\theta})^\dagger \mathbf{v}_k(\mathbf{x}) \right| \tag{D.5}$$

$$\lesssim \frac{n^2}{2^n p}\varphi ||\mathbf{P}_{\mathbf{1}}^\perp \mathbf{v}_k(\mathbf{x})||_2^2 + \varphi \frac{n^{3/2}}{2^n p} ||\mathbf{P}_{\mathbf{1}}^\perp \mathbf{v}_k(\mathbf{x})||_2 ||\mathbf{v}_k(\mathbf{x})||_\infty$$

$$\left| \frac{1}{\sqrt{\widehat{\mathbf{v}}_k(\mathbf{x})^T \nabla^2 \mathcal{L}(\widehat{\boldsymbol{\theta}})^\dagger \widehat{\mathbf{v}}_k(\mathbf{x})}} - \frac{1}{\sqrt{\mathbf{v}_k(\mathbf{x})^T \nabla^2 \mathcal{L}(\boldsymbol{\theta}^*)^\dagger \mathbf{v}_k(\mathbf{x})}} \right| \tag{D.6}$$

$$\lesssim \frac{n^3 \sqrt{2^n p}}{||\mathbf{P}_{\mathbf{1}}^\perp \mathbf{v}_k(\mathbf{x})||_2} \varphi + n^{5/2} \varphi \frac{||\mathbf{v}_k(\mathbf{x})||_\infty}{||\mathbf{P}_{\mathbf{1}}^\perp \mathbf{v}_k(\mathbf{x})||_2^2}$$

See Section G.3 for proofs of the above results.

### D.1  PROOF OF THEOREM 4.1

*Proof.* We begin by restating a series of useful objects that are referenced throughout the proof. Firstly, recall the four, likelihood-related quantities: the negative log-likelihood of the multinomial logit choice model, $\Lambda(\boldsymbol{\theta}; \mathbf{x})$ ; the local approximation, $\mathcal{L}(\boldsymbol{\theta}; \mathbf{x})$; and perturbed versions of each quantity respectively for $Z_{\mathcal{S}}^\ell$ being iid, standard Gaussian random variables:

$$\Lambda(\boldsymbol{\theta}; \mathbf{x}) := \sum_{\mathcal{S} \in \mathcal{E}} \left[ \frac{1}{L} \sum_{\ell \in [L]} K_{\mathbf{H}}(\mathbf{X}_{\mathcal{S}}^\ell - \mathbf{x}) \sum_{i \in \mathcal{S}^+} \left( y_{\mathcal{S}}^{(i,\ell)} \theta_i(\mathbf{X}_{\mathcal{S}}^\ell) - \log \left( \sum_{j \in \mathcal{S}^+} e^{\theta_j(\mathbf{X}_{\mathcal{S}}^\ell)} \right) \right) \right]$$

$$\widetilde{\Lambda}(\boldsymbol{\theta}; \mathbf{x}) := \sum_{\mathcal{S} \in \mathcal{E}} \left[ \frac{1}{L} \sum_{\ell \in [L]} Z_{\mathcal{S}}^\ell K_{\mathbf{H}}(\mathbf{X}_{\mathcal{S}}^\ell - \mathbf{x}) \sum_{i \in \mathcal{S}^+} \left( y_{\mathcal{S}}^{(i,\ell)} \theta_i(\mathbf{X}_{\mathcal{S}}^\ell) - \log \left( \sum_{j \in \mathcal{S}^+} e^{\theta_j(\mathbf{X}_{\mathcal{S}}^\ell)} \right) \right) \right]$$

$$\mathcal{L}(\boldsymbol{\theta}; \mathbf{x}) := \sum_{\mathcal{S} \in \mathcal{E}} \left[ \frac{1}{L} \sum_{\ell \in [L]} K_{\mathbf{H}}(\mathbf{X}_{\mathcal{S}}^\ell - \mathbf{x}) \sum_{i \in \mathcal{S}^+} \left( y_{\mathcal{S}}^{(i,\ell)} \theta_i(\mathbf{x}) - \log \left( \sum_{j \in \mathcal{S}^+} e^{\theta_j(\mathbf{x})} \right) \right) \right]$$

$$\widetilde{\mathcal{L}}(\boldsymbol{\theta}; \mathbf{x}) := \sum_{\mathcal{S} \in \mathcal{E}} \left[ \frac{1}{L} \sum_{\ell \in [L]} Z_{\mathcal{S}}^\ell K_{\mathbf{H}}(\mathbf{X}_{\mathcal{S}}^\ell - \mathbf{x}) \sum_{i \in \mathcal{S}^+} \left( y_{\mathcal{S}}^{(i,\ell)} \theta_i(\mathbf{x}) - \log \left( \sum_{j \in \mathcal{S}^+} e^{\theta_j(\mathbf{x})} \right) \right) \right] .$$

We will often suppress the explicit dependence on the given context $\mathbf{x}$ and data $\mathcal{D}$, letting $\mathcal{L}(\boldsymbol{\theta}) \equiv \mathcal{L}(\boldsymbol{\theta}; \mathbf{x}, \mathcal{D})$ and similarly for $\Lambda(\boldsymbol{\theta}), \widetilde{\mathcal{L}}(\boldsymbol{\theta}), \widetilde{\Lambda}(\boldsymbol{\theta})$. Now we recall the definitions of our statistic $T$ and

Gaussian approximation $W$, and we define two related objects $T_0, W_0$.

$$T = \sup_{\mathbf{x} \in \Omega} \max_{k \in [n]} \sqrt{Lh^d} \left( \frac{\widehat{\Delta}_k(\mathbf{x}) - \Delta_k(\mathbf{x})}{\sqrt{\widehat{\mathbf{v}}_k(\mathbf{x})^T \nabla^2 \mathcal{L}(\widehat{\boldsymbol{\theta}}; \mathbf{x}) \widehat{\mathbf{v}}_k(\mathbf{x})}} \right)$$

$$T_0 = \sup_{\mathbf{x} \in \Omega} \max_{k \in [n]} \sqrt{Lh^d} \left( \frac{-\mathbf{v}_k(\mathbf{x})^T \nabla^2 \Lambda(\boldsymbol{\theta}^*)^\dagger}{\sqrt{\mathbf{v}_k(\mathbf{x})^T \nabla^2 \Lambda(\boldsymbol{\theta}^*) \mathbf{v}_k(\mathbf{x})}} \nabla \Lambda(\boldsymbol{\theta}^*) \right)$$

$$W = \sup_{\mathbf{x} \in \Omega} \max_{k \in [n]} \sqrt{Lh^d} \left( \frac{-\widehat{\mathbf{v}}_k(\mathbf{x})^T \nabla^2 \mathcal{L}(\widehat{\boldsymbol{\theta}})^\dagger}{\sqrt{\widehat{\mathbf{v}}_k(\mathbf{x})^T \nabla^2 \mathcal{L}(\widehat{\boldsymbol{\theta}}) \widehat{\mathbf{v}}_k(\mathbf{x})}} \nabla \widetilde{\mathcal{L}}(\widehat{\boldsymbol{\theta}}) \right)$$

$$W_0 = \sup_{\mathbf{x} \in \Omega} \max_{k \in [n]} \sqrt{Lh^d} \left( \frac{-\mathbf{v}_k(\mathbf{x})^T \nabla^2 \Lambda(\boldsymbol{\theta}^*)^\dagger}{\sqrt{\mathbf{v}_k(\mathbf{x})^T \nabla^2 \Lambda(\boldsymbol{\theta}^*) \mathbf{v}_k(\mathbf{x})}} \nabla \widetilde{\Lambda}(\boldsymbol{\theta}^*) \right)$$

We also define an auxiliary object

$$T_{0,k} := \left( \frac{-\mathbf{v}_k(\mathbf{x})^T \nabla^2 \Lambda(\boldsymbol{\theta}^*)^\dagger}{\sqrt{\mathbf{v}_k(\mathbf{x})^T \nabla^2 \Lambda(\boldsymbol{\theta}^*) \mathbf{v}_k(\mathbf{x})}} \nabla \Lambda(\boldsymbol{\theta}^*) \right) . \tag{D.7}$$

By Corollary 4.6 of Wang et al. (2025b) and our assumption of $\Omega = [0,1]^d$, the proof is complete if the following results hold, for some $\zeta_1, \zeta_2 = o(1)$:

(1.) $\mathbb{P}(|T - T_0| \geq \zeta_1) \leq \zeta_2$

(2.) $\mathbb{P}(\mathbb{P}(|W - W_0| \geq \zeta_1 \mid \mathcal{D}) \geq \zeta_2) \leq \zeta_2$

(3.) $V[T_{0,k}(\mathbf{x})] \in [c, C]$ for any $\mathbf{x} \in \Omega, k \in [n]$ and some $c, C > 0$

(4.) Assumptions 4.11-12 of Wang et al. (2025b)

(5.) $\zeta_1 \sqrt{\log(2^n pL)} + \zeta_2 = o(1)$

We state results of the first three items, relegating their proofs to Section G.

**Lemma D.2.** Under the conditions and result of Theorem 3.4,

$$\mathbb{P}\left( |T - T_0| \geq dn^3 \frac{\log(2^n pL)}{(2^n pL)^{2/4+d}} \right) \lesssim n^{-10}$$

See Section G.4 for a proof of this Lemma.

**Lemma D.3.** Under the conditions and result of Theorem 3.4,

$$\mathbb{P}\left( \mathbb{P}\left( |W - W_0| \geq dn^3 \frac{\log(2^n pL)}{(2^n pL)^{2/4+d}} \right) \gtrsim n^{-10} \right) \lesssim n^{-10}$$

See Section G.5 for a proof of this Lemma.

**Lemma D.4.** For $T_{0,k}(\mathbf{x})$ as defined in Equation D.7, we have $V[T_{0,k}(\mathbf{x})] = 1$.

See Section G.6 for a proof of this lemma.

Assumptions 4.11-12 of Wang et al. (2025b) hold by applying results from Lemma C.2 and assumptions in Section H.1, respectively. From the above results, we take $\zeta_1 = dn^3 \frac{\log(2^n pL)}{(2^n pL)^{2/4+d}}, \zeta_2 = cn^{-10}$. Under the assumption $d = O(1)$ and $n^3 \log(2^n pL) \lesssim (2^n pL)^\varepsilon$ for some $\varepsilon < 2/(4 + d)$, condition (5) is met. Thus the argument is complete.

$\square$

### D.2 Proof of Theorem 4.2

We begin by proving the validity of the proposed confidence band $[\widehat{D}_L(\mathbf{x}), \widehat{D}_U(\mathbf{x})]$ for $\kappa^*(\mathbf{x})$. Then based on our proposed test construction using these confidence bands, we demonstrate asymptotic Type I error rate control of our testing procedure. Both arguments follow similarly to those in Shen et al. (2023) with alterations for our contextual set-up. In both instances, we begin with conservative, pointwise results before extending to simultaneously valid confidence bands and Type I error control respectively.

#### D.2.1 Validity of Confidence Bands

From Theorem 4.1, we have

$$\lim_{n,L\to\infty} \sup_{\alpha\in(0,1)} \sup_{\boldsymbol{\theta}\in\Theta} \mathbb{P}_{\boldsymbol{\theta}}\left(\sup_{\mathbf{x}\in\Omega} \max_{k\in[n]} \sqrt{Lh^d}\frac{\widehat{\Delta}_k(\mathbf{x}) - \Delta_k(\mathbf{x})}{\sqrt{\widehat{\mathbf{v}}_k(\mathbf{x})^T\nabla^2\mathcal{L}(\widehat{\boldsymbol{\theta}})^\dagger\widehat{\mathbf{v}}_k(\mathbf{x})}} \geq c_W(\alpha/2, \mathcal{D})\right) = \alpha/2$$

An equivalent argument of this Gaussian approximation result, using symmetry of the Gaussian, yields a similarly useful statement of valid quantile approximation:

$$\lim_{n,L\to\infty} \sup_{\alpha\in(0,1)} \sup_{\boldsymbol{\theta}\in\Theta} \mathbb{P}_{\boldsymbol{\theta}}\left(\inf_{\mathbf{x}\in\Omega} \min_{k\in[n]} \sqrt{Lh^d}\frac{\widehat{\Delta}_k(\mathbf{x}) - \Delta_k(\mathbf{x})}{\sqrt{\widehat{\mathbf{v}}_k(\mathbf{x})^T\nabla^2\mathcal{L}(\widehat{\boldsymbol{\theta}})^\dagger\widehat{\mathbf{v}}_k(\mathbf{x})}} \leq -c_W(\alpha/2, \mathcal{D})\right) = \alpha/2$$

We now want to show that

$$\liminf_{n,L\to\infty} \inf_{\boldsymbol{\theta}\in\Theta} \inf_{\mathbf{x}\in\Omega} \mathbb{P}_{\boldsymbol{\theta}}(\kappa^*(\mathbf{x}) \in [\widehat{D}_L(\mathbf{x}), \widehat{D}_U(\mathbf{x})]) \geq 1 - \alpha$$

for any $\alpha \in (0,1)$. That is, $[\widehat{D}_L(\mathbf{x}), \widehat{D}_U(\mathbf{x})]$ construct valid confidence bands on $\kappa^*(\mathbf{x})$.

$$\mathbb{P}_{\boldsymbol{\theta}}\left(\widehat{D}_L(\mathbf{x}) \leq \kappa^*(\mathbf{x}) \leq \widehat{D}_U(\mathbf{x})\right)$$

$$\geq 1 - \mathbb{P}_{\boldsymbol{\theta}}\left(\widehat{D}_L(\mathbf{x}) > \kappa^*(\mathbf{x})\right) - \mathbb{P}_{\boldsymbol{\theta}}\left(\Delta_{\widehat{D}_U(\mathbf{x})}(\mathbf{x}) < \kappa^*(\mathbf{x})\right)$$

$$= 1 - \mathbb{P}_{\boldsymbol{\theta}}\left(\Delta_{\widehat{D}_L(\mathbf{x})}(\mathbf{x}) \geq 0\right) - \mathbb{P}\left(\left\{\widehat{D}_U(\mathbf{x}) + 1 < 0\right\} \cap \left\{\widehat{D}_U(\mathbf{x}) < n\right\}\right)$$

$$= 1 - \mathbb{P}_{\boldsymbol{\theta}}\left(\left\{\Delta_{\widehat{D}_L(\mathbf{x})}(\mathbf{x}) \geq 0\right\} \cap \left\{\widehat{\Delta}_{\widehat{D}_L(\mathbf{x})}(\mathbf{x}) < -c_W(\alpha/2, \mathcal{D})\sqrt{\widehat{\mathbf{v}}_{\widehat{D}_L(\mathbf{x})}^T(\mathbf{x})\nabla^2\mathcal{L}(\mathbf{x})^\dagger\widehat{\mathbf{v}}_{\widehat{D}_L(\mathbf{x})}(\mathbf{x}))/Lh^d}\right\}\right)$$

$$\quad - \mathbb{P}_{\boldsymbol{\theta}}\left(\left\{\Delta_{\widehat{D}(\mathbf{x})_U+1}(\mathbf{x}) < 0\right\} \cap \left\{\widehat{D}_U(\mathbf{x}) < n\right\}\right.$$

$$\left.\quad\quad\quad \cap \left\{\widehat{\Delta}_{\widehat{D}_U(\mathbf{x})+1}(\mathbf{x}) > c_W(\alpha/2, \mathcal{D})\sqrt{\mathbf{v}_{\widehat{D}_U(\mathbf{x})+1}^T(\mathbf{x})\nabla^2\mathcal{L}(\mathbf{x})^\dagger\mathbf{v}_{\widehat{D}_U(\mathbf{x})+1}(\mathbf{x})/Lh^d}\right\}\right)$$

$$\geq 1 - \mathbb{P}_{\boldsymbol{\theta}}\left(\sqrt{Lh^d}\min_{k>\kappa^*(\mathbf{x})}\frac{\widehat{\Delta}_k(\mathbf{x}) - \Delta_k(\mathbf{x})}{\sqrt{\widehat{\mathbf{v}}_k(\mathbf{x})^\top\nabla^2\mathcal{L}(\mathbf{x})^\dagger\widehat{\mathbf{v}}_k(\mathbf{x})}} \leq -c_W(\alpha/2, \mathcal{D})\right)$$

$$\quad - \mathbb{P}_{\boldsymbol{\theta}}\left(\sqrt{Lh^d}\max_{k\leq\kappa^*(\mathbf{x})}\frac{\widehat{\Delta}_k(\mathbf{x}) - \Delta_k(\mathbf{x})}{\sqrt{\widehat{\mathbf{v}}_k(\mathbf{x})^\top\nabla^2\mathcal{L}(\mathbf{x})^\dagger\widehat{\mathbf{v}}_k(\mathbf{x})}} \geq c_W(\alpha/2, \mathcal{D})\right)$$

$$\geq 1 - \mathbb{P}_{\boldsymbol{\theta}}\left(\sqrt{Lh^d}\min_{k\in[n]}\frac{\widehat{\Delta}_k(\mathbf{x}) - \Delta_k(\mathbf{x})}{\sqrt{\widehat{\mathbf{v}}_k(\mathbf{x})^\top\nabla^2\mathcal{L}(\mathbf{x})^\dagger\widehat{\mathbf{v}}_k(\mathbf{x})}} \geq c_W(\alpha/2, \mathcal{D})\right)$$

$$\quad - \mathbb{P}_{\boldsymbol{\theta}}\left(\sqrt{Lh^d}\max_{k\in[n]}\frac{(\widehat{\Delta}_k(\mathbf{x}) - \Delta_k(\mathbf{x}))}{\sqrt{\widehat{\mathbf{v}}_k(\mathbf{x})^\top\nabla^2\mathcal{L}(\mathbf{x})^\dagger\widehat{\mathbf{v}}_k(\mathbf{x})}} \leq -c_W(\alpha/2, \mathcal{D})\right)$$

The result holds then by the quantile approximations stated above, and taking the suprema over $\alpha \in (0,1), \theta \in \Theta, \mathbf{x} \in \Omega$ and limit as $n, L \to \infty$ for both sides.

#### D.2.2 Validity of Testing Procedures

Below we assume the supremum and infimum are taken over subsets of $\boldsymbol{\theta} \in \Theta$ throughout for $\Theta$ defined in Equation 2.3. We suppress this explicit notation, writing only $\boldsymbol{\theta}$ to mean any $\boldsymbol{\theta} \in \Theta$ or

$\boldsymbol{\theta} : H_0$ for $\boldsymbol{\theta} \in \Theta$ such that $H_0$ holds.

$$
\begin{aligned}
\sup_{\mathbf{x} \in \Omega} \sup_{\boldsymbol{\theta}:H_0} \mathbb{P}_{\boldsymbol{\theta}}(\text{Reject } H_0) &= \sup_{\mathbf{x} \in \Omega} \sup_{\boldsymbol{\theta}:\kappa^*(\mathbf{x}) \in \mathcal{K}_0} \mathbb{P}_{\boldsymbol{\theta}} \left( \{\kappa^*(\mathbf{x}) \in \mathcal{K}_0\} \cap \{[\widehat{D}_L(\mathbf{x}), \widehat{D}_U(\mathbf{x})] \cap \mathcal{K}_0 = \emptyset\} \right) \\
&\leq \sup_{\mathbf{x} \in \Omega} \sup_{\boldsymbol{\theta}:\kappa^*(\mathbf{x}) \in \mathcal{K}_0} \mathbb{P}_{\boldsymbol{\theta}} \left( \{\kappa^*(\mathbf{x}) \in \mathcal{K}_0\} \cap \{\kappa^*(\mathbf{x}) \notin [\widehat{D}_L(\mathbf{x}), \widehat{D}_U(\mathbf{x})]\} \right) \\
&\leq 1 - \inf_{\mathbf{x} \in \Omega} \inf_{\boldsymbol{\theta}:\kappa^*(\mathbf{x}) \in \mathcal{K}_0} \mathbb{P} \left( \kappa^*(\mathbf{x}) \in [\widehat{D}_L(\mathbf{x}), \widehat{D}_U(\mathbf{x})] \right) \\
&= 1 - \inf_{\mathbf{x} \in \Omega} \inf_{\boldsymbol{\theta}:H_0} \mathbb{P} \left( \kappa^*(\mathbf{x}) \in [\widehat{D}_L(\mathbf{x}), \widehat{D}_U(\mathbf{x})] \right)
\end{aligned}
$$

The equality holds by equivalence of $H_0$ holding and $\kappa^* \in \mathcal{K}_0$ for null property set $\mathcal{K}_0$ as discussed in Section 2.3. The pointwise result holds by taking the limit over $n, L \to \infty$ and applying the validity of the confidence band, argued above.

We can also construct a uniformly valid test by a similar procedure, given our results above for simultaneously valid confidence bands.

$$
\liminf_{n,L \to \infty} \inf_{\boldsymbol{\theta} \in \Theta} \mathbb{P}_{\boldsymbol{\theta}} \left( \kappa^*(\mathbf{x}) \in [\widehat{D}_L(\mathbf{x}), \widehat{D}_U(\mathbf{x})], \forall \mathbf{x} \in \Omega \right) \geq 1 - \alpha
$$

By a similar argument, we can construct tests for any contextual rejection of the hull hypothesis:

$$
\begin{aligned}
\sup_{\boldsymbol{\theta}:H_0} \mathbb{P}_{\boldsymbol{\theta}}(\text{Reject } H_0, \forall \mathbf{x} \in \Omega) &= \sup_{\boldsymbol{\theta}:\kappa^*(\mathbf{x}) \in \mathcal{K}_0} \mathbb{P}_{\boldsymbol{\theta}} \left( \{\kappa^*(\mathbf{x}) \in \mathcal{K}_0\} \cap \{[\widehat{D}_L(\mathbf{x}), \widehat{D}_U(\mathbf{x})] \cap \mathcal{K}_0 = \emptyset\}, \forall \mathbf{x} \in \Omega \right) \\
&\leq \sup_{\mathbf{x} \in \Omega} \sup_{\boldsymbol{\theta}:\kappa^*(\mathbf{x}) \in \mathcal{K}_0} \mathbb{P}_{\boldsymbol{\theta}} \left( \{\kappa^*(\mathbf{x}) \in \mathcal{K}_0\} \cap \{\kappa^*(\mathbf{x}) \notin [\widehat{D}_L(\mathbf{x}), \widehat{D}_U(\mathbf{x})]\}, \forall \mathbf{x} \in \Omega \right) \\
&\leq 1 - \inf_{\mathbf{x} \in \Omega} \inf_{\boldsymbol{\theta}:\kappa^*(\mathbf{x}) \in \mathcal{K}_0} \mathbb{P} \left( \exists \mathbf{x} \in \Omega : \kappa^*(\mathbf{x}) \in [\widehat{D}_L(\mathbf{x}), \widehat{D}_U(\mathbf{x})] \right) \\
&\leq 1 - \inf_{\mathbf{x} \in \Omega} \inf_{\boldsymbol{\theta}:H_0} \mathbb{P} \left( \kappa^*(\mathbf{x}) \in [\widehat{D}_L(\mathbf{x}), \widehat{D}_U(\mathbf{x})], \forall \mathbf{x} \in \Omega \right)
\end{aligned}
$$

Once again the result holds in limit over $n, L \to \infty$

# E  Proof of Lemmas for Theorem 3.1 in Section B

Throughout these proofs, we implicitly use a truncation argument to control terms in high probability. That is, for some event $E_1$ that we care to analyze, and some event $E_2$, we have

$$\mathbb{P}(E_1) = \mathbb{P}\Big(\{E_1 \cap E_2\} \cup \{E_1 \cap E_2^c\}\Big) = \mathbb{P}(E_1 \mid E_2)\mathbb{P}(E_2) + \mathbb{P}(E_1 \mid E_2^c)\mathbb{P}(E_2^c) \leq \mathbb{P}(E_1 \mid E_2) + \mathbb{P}(E_2^c) \,.$$

This bound is sufficiently tight when $E_2^c$ and $E_1 \mid E_2$ hold in high probability. In our arguments, event $E_2$ will commonly be controlling the effective number of observed assortments, e.g. $|\mathcal{E}| \equiv \sum_{\mathcal{S} \in \mathcal{S}} \mathcal{E} \asymp 2^n p$, and event $E_1$ the object of interest for a given proof, e.g. $\lambda_{\max}(\nabla^2 \mathcal{L})$.

## E.1  Proof of Lemma B.1

Define the positive semidefinite matrix

$$\mathbf{M}_{\mathcal{S}} := \sum_{(i<k)\in\mathcal{S}_+} \frac{1}{|\mathcal{S}_+|^2}(\mathbf{e}_i - \mathbf{e}_k)(\mathbf{e}_i - \mathbf{e}_k)^T$$

and the scalar

$$b_{\ell,\mathcal{S}}(\mathbf{x}) := \mathcal{E}_{\mathcal{S}} K_{\mathbf{H}}\big(\mathbf{X}_{\mathcal{S}}^\ell - \mathbf{x}\big) - \mathbb{E}\Big[\mathcal{E}_{\mathcal{S}} K_{\mathbf{H}}\big(\mathbf{X} - \mathbf{x}\big)\Big] \,.$$

With expectation taken over $\mathbf{X}, \mathcal{E}$. We can then decompose our object of interest as

$$\mathbf{L}_{\boldsymbol{\mathcal{E}}}(\mathbf{x}) := \sum_{\mathcal{S}\in\mathcal{E}}\sum_{\ell\in[L]} L^{-1} K_{\mathbf{H}}\big(\mathbf{X}_{\mathcal{S}}^\ell - \mathbf{x}\big)\mathbf{M}_{\mathcal{S}}$$

$$= \underbrace{\sum_{\mathcal{S}\in\mathcal{S}}\sum_{\ell\in[L]} L^{-1} b_{\ell,\mathcal{S}}(\mathbf{x})\mathbf{M}_{\mathcal{S}}}_{:=\mathbf{A}} + \underbrace{\sum_{\mathcal{S}\in\mathcal{S}}\sum_{\ell\in[L]} L^{-1}\mathbb{E}\Big[\mathcal{E}_{\mathcal{S}} K_{\mathbf{H}}\big(\mathbf{X}_{\mathcal{S}}^\ell - \mathbf{x}\big)\Big]\mathbf{M}_{\mathcal{S}}}_{:=\mathbf{B}}$$

We will bound the extreme eigenvalues of $\mathbf{A}, \mathbf{B}$ and then control $\lambda_{\max}(\mathbf{L}_{\boldsymbol{\mathcal{E}}})$, $\lambda_{\min,\perp}(\mathbf{L}_{\boldsymbol{\mathcal{E}}})$ by Weyl's inequality, all uniformly over $\Omega$.

### E.1.1  $\lambda_{\max}(\mathbf{A})$

Note that $\lambda_{\max}(\mathbf{M}_{\mathcal{S}}) \leq 1$. Then

$$\sup_{\mathbf{x}\in\Omega}\lambda_{\max}(\mathbf{A}) \leq \sup_{\mathbf{x}\in\Omega}\frac{1}{L}\Big|\sum_{\mathcal{S}\in\mathcal{E}}\sum_{\ell\in[L]} b_{\ell,\mathcal{S}}(\mathbf{x})\Big| \,. \tag{E.1}$$

Thus we control $\lambda_{\max}(\mathbf{A})$ by mean deviations in the kernel weights, i.e. $b_{\ell,\mathcal{S}}(\mathbf{x})$. To control this term, define the event $E = \{\mathcal{E} \asymp 2^n p\}$. By Lemma H.2,

$$\mathbb{P}\left(2^n p \sup_{\mathbf{x}\in\Omega}\frac{1}{2^n pL}\Big|\sum_{\mathcal{S}\in\mathcal{E}}\sum_{\ell\in[L]} b_{\ell,\mathcal{S}}(\mathbf{x})\Big| \gtrsim 2^n p\sqrt{\frac{d\log(2^n pLh^{-d})}{2^n pLh^d}} \mid |\mathcal{E}| \asymp 2^n p\right) \lesssim n^{-10} \tag{E.2}$$

Then by Equations E.1 and E.2, and noting $\mathbb{P}(E^c) \lesssim n^{-10}$ by a standard Chernoff bound, we have

$$\mathbb{P}\left(\sup_{\mathbf{x}\in\Omega}\lambda_{\max}(\mathbf{A}) \gtrsim 2^n p\sqrt{\frac{d\log(2^n pLh^{-d})}{2^n pLh^d}}\right)$$

$$\lesssim \mathbb{P}\left(\sup_{\mathbf{x}\in\Omega}\lambda_{\max}(\mathbf{A}) \gtrsim 2^n p\sqrt{\frac{d\log(2^n pLh^{-d})}{2^n pLh^d}} \mid |\mathcal{E}| \asymp 2^n p\right) + \mathbb{P}(|\mathcal{E}| \not\asymp 2^n p) \lesssim n^{-10}$$

assuming $2^n p \geq Cn\log n$ for $C > 0$. The same bound holds for $\sup_{\mathbf{x}\in\Omega}\lambda_{\max}(-\mathbf{A})$.

### E.1.2 $\lambda_{\max}(\mathbf{B})$

By Lemma H.1 and Lemma A.1 in Shen et al. (2023), along with our positivity by Assumption 2, we have

$$\sup_{\mathbf{x}\in\Omega} \lambda_{\max}(\mathbf{B}) \leq 2^n p(C_f + cdh^2) \quad \text{and} \quad \inf_{\mathbf{x}\in\Omega} \lambda_{\min,\perp}(\mathbf{B}) \geq \frac{1}{2}(c_f + cdh^2)2^n p/n . \quad \text{(E.3)}$$

The first inequality holds deterministically, while the second holds with probability at least $1 - O(n^{-10})$ under the assumption $2^n p \geq Cn \log n$ for large enough $C > 0$ (Shen et al., 2023).

### E.1.3 $\lambda_{\max}(\mathbf{L}_{\mathcal{E}})$

Now using Weyl's inequality and the triangle inequality,

$$\sup_{\mathbf{x}\in\Omega} \lambda_{\max}(\mathbf{L}_{\mathcal{E}}) \leq \sup_{\mathbf{x}\in\Omega} \lambda_{\max}(\mathbf{A}) + \sup_{\mathbf{x}\in\Omega} \lambda_{\max}(\mathbf{B})$$

$$\leq C2^n p \sqrt{\frac{d \log(2^n pLh^{-d})}{2^n pLh^d}} + c2^n pdh^2 + C_f 2^n p$$

$$\inf_{\mathbf{x}\in\Omega} \lambda_{\min,\perp}(\mathbf{L}_{\mathcal{E}}) \geq \inf_{\mathbf{x}\in\Omega} \lambda_{\min,\perp}(\mathbf{B}) - \sup_{\mathbf{x}\in\Omega} \lambda_{\max}(-\mathbf{A})$$

$$\geq \frac{1}{2}(c_f + cdh^2)2^n p/n - 2^n p\sqrt{\frac{d \log(2^n pLh^{-d})}{2^n pLh^d}}$$

hold jointly with probability at least $1 - O(n^{-10})$ for positive constants $c, C, c_f, C_f$.

### E.2 Proof of Lemma B.2

The below argument is largely algebraic and follows the proof of Lemma A.2 in Shen et al. (2023) nearly identically, differing only in the uniform control over $\Omega$ in final statement of the results. Recall the assumption that $\sup_{\mathbf{x}\in\Omega} ||\boldsymbol{\beta} - \boldsymbol{\theta}^*(\mathbf{x})||_2 \leq C$

$$\frac{e^{\beta_i}}{\sum_{j\in\mathcal{S}_+} e^{\beta_j}} = \frac{1}{\sum_{j\in\mathcal{S}_+} \exp\{(\beta_j - \theta_j^*(\mathbf{x})) + (\theta_j^*(\mathbf{x}) - \theta_i^*(\mathbf{x})) + (\theta_i^*(\mathbf{x}) - \beta_i)\}}$$

$$\leq \frac{1}{\sum_{j\in\mathcal{S}_+} \exp\{-\log c_\theta - 2C_1\}} = \frac{c_\theta e^{2C_1}}{|\mathcal{S}_+|}$$

with similar work providing an analogous lower-bound $\frac{e^{\beta_i}}{\sum_{j\in\mathcal{S}_+} e^{\beta_j}} \geq \frac{1}{|\mathcal{S}_+|c_\theta e^{2C_1}}$. Noting these (scalar) upper- and lower-bounds are independent of $\mathbf{x}$, they hold in $\sup_{\mathbf{x}\in\Omega}, \inf_{\mathbf{x}\in\Omega}$ respectively. Returning to the form of our Hessian, we have

$$\nabla^2 \mathcal{L} = \sum_{\mathcal{S}\in\mathcal{E}} \sum_{\ell\in[L]} \sum_{i\in\mathcal{S}_+} L^{-1} K_{\mathbf{H}}(\mathbf{X}_{\mathcal{S}}^\ell - \mathbf{x}) \left( \sum_{k\in\mathcal{S}_+; \, k>i} \frac{e^{\theta_i(\mathbf{x})} e^{\theta_k(\mathbf{x})}}{\left(\sum_{j\in\mathcal{S}^+} e^{\theta_j(\mathbf{x})}\right)^2} (\mathbf{e}_i - \mathbf{e}_k)(\mathbf{e}_i - \mathbf{e}_k)^T \right)$$

$$\preccurlyeq \frac{1}{c_\theta^2 e^{4C_1}} \sum_{\mathcal{S}\in\mathcal{E}} \sum_{\ell\in[L]} \sum_{i\in\mathcal{S}_+} L^{-1} K_{\mathbf{H}}(\mathbf{X}_{\mathcal{S}}^\ell - \mathbf{x}) \left( \sum_{k\in\mathcal{S}_+; \, k>i} \frac{1}{|\mathcal{S}_+|^2} (\mathbf{e}_i - \mathbf{e}_k)(\mathbf{e}_i - \mathbf{e}_k)^T \right)$$

which is exactly a scaled version of $\mathbf{L}_{\mathcal{E}}$, the eigenvalues of which we control in Lemma B.1. Then

$$\sup_{\mathbf{x}\in\Omega} \lambda_{\max}\left(\nabla^2 \mathcal{L}(\boldsymbol{\beta}; \mathbf{x}, \mathcal{D})\right) \leq \lambda + c_\theta^2 e^{4C_1} \sup_{\mathbf{x}\in\Omega} \lambda_{\max}(\mathbf{L}_{\mathcal{E}})$$

for $\mathbf{L}_{\mathcal{E}}$ as defined in the Proof Lemma B.1. A similar analysis in the opposite direction using the scalar lower-bound provides result

$$\inf_{\mathbf{x}\in\Omega} \lambda_{\min,\perp}\left(\nabla^2 \mathcal{L}(\boldsymbol{\beta}; \mathbf{x}, \mathcal{D})\right) \geq \lambda + \frac{1}{c_\theta^2 e^{4C_1}} \inf_{\mathbf{x}\in\Omega} \lambda_{\min,\perp}(\mathbf{L}_{\mathcal{E}}) .$$

### E.3  PROOF OF LEMMA B.3

We first work with $\nabla \mathcal{L}(\boldsymbol{\theta}^*; \mathbf{x}, \mathcal{D})$, decomposing the gradient as we did the Hessian in Proof E.2. For ease of presentation, we write

$$g_{i,\mathcal{S}}(\mathbf{x}; \boldsymbol{\theta}) = \frac{e^{\theta_i(\mathbf{x})}}{\sum_{j \in \mathcal{S}_+} e^{\theta_j(\mathbf{x})}} \quad \text{and} \quad g_{i,\mathcal{S}}(\mathbf{x}) \equiv g_{i,\mathcal{S}}(\mathbf{x}; \boldsymbol{\theta}^*) \,.$$

The (negative) gradient $-\nabla \mathcal{L}(\boldsymbol{\theta}^*; \mathbf{x}, \mathcal{D})$ can then be written and decomposed as

$$\underbrace{\sum_{\mathcal{S} \in \boldsymbol{\mathcal{S}}} \sum_{\ell \in [L]} L^{-1} \left\{ \mathcal{E}_{\mathcal{S}} K_{\mathbf{H}}\left(\mathbf{X}_{\mathcal{S}}^{\ell} - \mathbf{x}\right) \sum_{i \in \mathcal{S}_+} \left(y_{\mathcal{S}}^{(i,\ell)} - g_i(\mathbf{x})\right)\mathbf{e}_i - \mathbb{E}\left[\mathcal{E}_{\mathcal{S}} K_{\mathbf{H}}\left(\mathbf{X} - \mathbf{x}\right) \sum_{i \in \mathcal{S}_+} \left(y_{\mathcal{S}}^{(i,\ell)} - g_i(\mathbf{x})\right)\mathbf{e}_i\right]\right\}}_{:=\mathbf{a}}$$

$$\underbrace{+ \sum_{\mathcal{S} \in \boldsymbol{\mathcal{S}}} \sum_{\ell \in [L]} L^{-1} \mathbb{E}\left[\mathcal{E}_{\mathcal{S}} K_{\mathbf{H}}\left(\mathbf{X} - \mathbf{x}\right) \sum_{i \in \mathcal{S}_+} \left(y_{\mathcal{S}}^{(i,\ell)} - g_i(\mathbf{x})\right)\mathbf{e}_i\right]}_{:=\mathbf{b}}$$

with expectations taken over the data $\mathcal{D} = (\mathcal{E}_{\mathcal{S}}, \mathbf{X}, \mathbf{y})$. We will now bound $\sup_{\mathbf{x} \in \Omega} \|\nabla \mathcal{L}(\boldsymbol{\theta}^*; \mathbf{x}, \mathcal{D})\|_2 \leq \sup_{\mathbf{x} \in \Omega} \|\mathbf{a}\|_2 + \sup_{\mathbf{x} \in \Omega} \|\mathbf{b}\|_2$.

#### E.3.1  $\|\mathbf{a}\|_2$

We first control $\|\mathbf{a}\|_\infty$. Let $\boldsymbol{\mathcal{E}}_i = \{\mathcal{S} : i \in \mathcal{S}_+, \mathcal{E}_{\mathcal{S}} = 1\}$. The $i$th coordinate of $\mathbf{a}$ is then

$$\mathbf{a}_i = \sum_{\mathcal{S} \in \boldsymbol{\mathcal{E}}_i} \sum_{\ell \in [L]} L^{-1} \underbrace{\left\{ K_{\mathbf{H}}\left(\mathbf{X}_{\mathcal{S}}^{\ell} - \mathbf{x}\right)\left(y_{\mathcal{S}}^{(i,\ell)} - g_i(\mathbf{x})\right) - \mathbb{E}\left[K_{\mathbf{H}}\left(\mathbf{X} - \mathbf{x}\right)\left(y_{\mathcal{S}}^{(i,\ell)} - g_i(\mathbf{x})\right)\right]\right\}}_{:=b_{\mathcal{S}}^{(i,\ell)}(\mathbf{x})} \,.$$

We will argue that for a fixed $i$, $\mathbb{P}(\sup_{\mathbf{x} \in \Omega} |\mathbf{a}_i| \gtrsim 2^n p \sqrt{\frac{d \log(2^n p L h^{-d})}{2^n p L h^d}} \mid |\boldsymbol{\mathcal{E}}_i| \asymp 2^n p) \lesssim O(n^{-11})$. Momentarily assuming this result, we then have

$$\mathbb{P}\left(\sup_{\mathbf{x} \in \Omega} |\mathbf{a}_i| \gtrsim 2^n p \sqrt{\frac{d \log(2^n p L h^{-d})}{2^n p L h^d}}\right) \leq \mathbb{P}\left(\sup_{\mathbf{x} \in \Omega} |\mathbf{a}_i| \gtrsim 2^n p \sqrt{\frac{d \log(2^n p L h^{-d})}{2^n p L h^d}} \mid |\boldsymbol{\mathcal{E}}_i| \asymp 2^n p\right) + \mathbb{P}(|\boldsymbol{\mathcal{E}}_i| \not\asymp 2^n p)$$

$$\leq O(n^{-11}) \,,$$

as $\mathbb{P}(\boldsymbol{\mathcal{E}}_i \not\asymp 2^n p) \leq O(n^{-11})$ by a standard Chernoff bound argument given $2^n p \geq Cn \log(n)$ for large enough $C > 0$ and by independence $\boldsymbol{\mathcal{E}} \perp (\mathbf{X}, \mathbf{y})$. We then have

$$\mathbb{P}\left(\sup_{\mathbf{x} \in \Omega} \|\mathbf{a}\|_\infty \gtrsim 2^n p \sqrt{\frac{d \log(2^n p L h^{-d})}{2^n p L h^d}}\right) = \mathbb{P}\left(\bigcup_{i=1}^{n} \left\{\sup_{\mathbf{x} \in \Omega} |\mathbf{a}_i| \gtrsim 2^n p \sqrt{\frac{d \log(2^n p L h^{-d})}{2^n p L h^d}}\right\}\right) \leq n^{-10} \,.$$

$$(\text{E.4})$$

Conditional on $|\boldsymbol{\mathcal{E}}_i| \asymp 2^n p$, we have $\sup_{\mathbf{x} \in \Omega} |\mathbf{a}_i| \gtrsim 2^n p \sqrt{\frac{\log(2^n p L h^{-d})}{2^n p L h^d}}$ by Lemma H.5 with probability at most $O(n^{-11})$, again under the assumption $2^n p \geq Cn \log(n)$ for large enough $C > 0$.

#### E.3.2  $\|\mathbf{b}\|_2$ AND $\|\nabla^2 \mathcal{L}_\lambda\|_2$

Note by the tower rule, distribution of $y_{\mathcal{S}}^{(i,\ell)} \mid \mathbf{X}_{\mathcal{S}}^{\ell}$, and $\mathcal{E}_{\mathcal{S}} \perp \mathbf{X}$ for any $\mathbf{X} \in \Omega$, that

$$\mathbb{E}\left[\mathcal{E}_{\mathcal{S}} K_{\mathbf{H}}\left(\mathbf{X}_{\mathcal{S}}^{\ell} - \mathbf{x}\right)\left(y_{\mathcal{S}}^{(i,\ell)} - g_i(\mathbf{x})\right)\right] = p\mathbb{E}_{\mathbf{X}_{\mathcal{S}}^{\ell}}\left[K_{\mathbf{H}}\left(\mathbf{X}_{\mathcal{S}}^{\ell} - \mathbf{x}\right)\left(g_i(\mathbf{X}_{\mathcal{S}}^{\ell}) - g_i(\mathbf{x})\right)\right]$$

Component-wise, we then have $\mathbf{b}_i = c|\boldsymbol{\mathcal{E}}|dh^2$ by applying Lemma H.4. Again given $\boldsymbol{\mathcal{E}} \asymp 2^n p$ with probability at least $1 - O(n^{-11})$, then with similar probability $\mathbf{b}_i \asymp 2^n p dh^2$ and by union bound $\|\mathbf{b}\|_\infty \asymp 2^n p dh^2$ with probability at least $1 - O(n^{-10})$.

We then have

$$\sup_{\mathbf{x}\in\Omega} ||\nabla\mathcal{L}(\boldsymbol{\theta}^*;\mathbf{x},\mathcal{D})||_\infty \le \sqrt{n}\sup_{\mathbf{x}\in\Omega}||\mathbf{a}||_\infty + \sqrt{n}||\mathbf{b}||_\infty$$

$$\lesssim 2^n p\sqrt{\frac{d\log(2^n pLh^{-d})}{2^n pLh^{-d}}} + 2^n pdh^2$$

holds with probability at least $1 - O(n^{-10})$. Noting $\sup_{\mathbf{x}\in\Omega}||\boldsymbol{\theta}^*(\mathbf{x})||_2 \lesssim \sqrt{n}\log c_\theta$, then

$$\sup_{\mathbf{x}\in\Omega}||\nabla\mathcal{L}_\lambda(\boldsymbol{\theta}^*;\mathbf{x},\mathcal{D})||_2 \lesssim \lambda\sqrt{n} + 2^n p\sqrt{n\frac{\log(2^n pLh^{-d})}{2^n pLh^d}} + 2^n pCdh^2\sqrt{n}$$

### E.4 Proof of Lemma B.4

Define the following (symmetric, positive semi-definite) matrix

$$\mathbf{M}'_{\mathcal{S}^+} := \sum_{i>k\in\mathcal{S}^+} \frac{e^{\beta_i}e^{\beta_k}}{\left(\sum_{j\in\mathcal{S}^+} e^{\beta_j}\right)^2}(\mathbf{e}_i - \mathbf{e}_k)(\mathbf{e}_i - \mathbf{e}_k)^T$$

for arbitrary $\boldsymbol{\beta} \in \mathbb{R}^{n+1}$. Note that $\sup_{\mathbf{x}\in\Omega}|||\mathbf{M}'_{\mathcal{S}^+}|||_2 \lesssim n$ for any $\boldsymbol{\beta} \in \mathbb{R}^{n+1}$. We will also use $b_{\ell,\mathcal{S}}(\mathbf{x})$ as defined in Lemma B.1:

$$b_{\ell,\mathcal{S}(\mathbf{x})} := K_{\mathbf{H}}\left(\mathbf{X}_{\mathcal{S}}^\ell - \mathbf{x}\right) - \mathbb{E}\left[\mathcal{E}_{\mathcal{S}}K_{\mathbf{H}}\left(\mathbf{u} - \mathbf{x}\right)\right].$$

Then

$$\sup_{\mathbf{x}\in\Omega}|||\nabla^2\mathcal{L}(\boldsymbol{\beta};\mathbf{x},\mathcal{D})|||_2 \le \sup_{\mathbf{x}\in\Omega}\sum_{\mathcal{S}\in\boldsymbol{\mathcal{S}}}\sum_{\ell\in[L]}\frac{1}{L}\mathcal{E}_{\mathcal{S}}K_{\mathbf{H}}\left(\mathbf{X}_{\mathcal{S}}^\ell - \mathbf{x}\right)|||\mathbf{M}'_{\mathcal{S}^+}|||_2$$

$$\lesssim n\sup_{\mathbf{x}\in\Omega}\sum_{\mathcal{S}\in\boldsymbol{\mathcal{S}}}\sum_{\ell\in[L]}\frac{1}{L}\mathcal{E}_{\mathcal{S}}K_{\mathbf{H}}\left(\mathbf{X}_{\mathcal{S}}^\ell - \mathbf{x}\right)$$

$$\le n\sup_{\mathbf{x}\in\Omega}\sum_{\mathcal{S}\in\boldsymbol{\mathcal{S}}}\sum_{\ell\in[L]}\frac{1}{L}\mathcal{E}_{\mathcal{S}}b_{\ell,\mathcal{S}}(\mathbf{x}) + n\sup_{\mathbf{x}\in\Omega}\sum_{\mathcal{S}\in\boldsymbol{\mathcal{S}}}\sum_{\ell\in[L]}\frac{1}{L}\mathbb{E}\left[\mathcal{E}_{\mathcal{S}}K_{\mathbf{H}}\left(\mathbf{u} - \mathbf{x}\right)\right]$$

$$\le n\sup_{\mathbf{x}\in\Omega}\underbrace{\left|\frac{1}{L}\sum_{\mathcal{S}\in\mathcal{E}}\sum_{\ell\in[L]}b_{\ell,\mathcal{S}}(\mathbf{x})\right|}_{:=\mathbf{A}} + 2^n n\sup_{\mathbf{x}\in\Omega}\underbrace{\mathbb{E}\left[\mathcal{E}_{\mathcal{S}}K_{\mathbf{H}}\left(\mathbf{u} - \mathbf{x}\right)\right]}_{:=\mathbf{B}} \qquad \text{(E.5)}$$

By Lemma H.1 and boundedness of $f_{\mathbf{X}}(\mathbf{x})$, $\sup_{\mathbf{x}\in\Omega}\mathbf{B} \le p(Cdh^2 + C_f)$. By Lemma H.5 and $\mathcal{E} \perp (\mathbf{X}, y)$, we have

$$\mathbb{P}\left(\sup_{\mathbf{x}\in\Omega}|\mathbf{A}| \gtrsim 2^n p\sqrt{\frac{d\log(2^n pLh^{-d})}{2^n pLh^d}} \mid |\mathcal{E}| \asymp 2^n p\right) \le O(n^{-10}).$$

Combined with $\mathbb{P}(\mathcal{E} \not\asymp 2^n p) \le O(n^{-10})$, this provides

$$\mathbb{P}\left(\sup_{\mathbf{x}\in\Omega}|\mathbf{A}| \gtrsim 2^n p\sqrt{\frac{d\log(2^n pLh^{-d})}{2^n pLh^d}}\right) \le \mathbb{P}\left(\sup_{\mathbf{x}\in\Omega}|\mathbf{A}| \gtrsim 2^n p\sqrt{\frac{d\log(2^n pLh^{-d})}{2^n pLh^d}} \mid |\mathcal{E}| \asymp 2^n p\right) + \mathbb{P}(|\mathcal{E}| \not\asymp 2^n p) \lesssim n^{-10}$$

by Lemma H.2, with probability at least $1 - O(n^{-10})$ given $2^n p \ge Cn\log(n)$ for $C > 0$. Then by our decomposition in Equation E.5, we have

$$||\nabla^2\mathcal{L}_\lambda(\boldsymbol{\beta};\mathbf{x},\mathcal{D})||_2 \lesssim \lambda + C_f 2^n np + 2^n npdh^2 + 2^n pn\sqrt{\frac{d\log(2^n pLh^{-d})}{2^n pLh^d}}.$$

again with high probability under the same conditions.

### E.5 Proof of Lemma B.5

The proof follows by a Taylor Expansion of $\mathcal{L}_\lambda(\widehat{\boldsymbol{\theta}}; \mathbf{x}, \mathcal{D})$ about $\boldsymbol{\theta}^*$ for a fixed $\mathbf{x}$, taking the supremum, and applying Lemma B.3. That is,

$$
\begin{aligned}
\mathcal{L}_\lambda(\widehat{\boldsymbol{\theta}}; \mathbf{x}, \mathcal{D}) = \mathcal{L}_\lambda(\boldsymbol{\theta}^*; \mathbf{x}, \mathcal{D}) &+ (\widehat{\boldsymbol{\theta}}(\mathbf{x}) - \boldsymbol{\theta}^*(\mathbf{x}))^T \nabla \mathcal{L}_\lambda(\boldsymbol{\theta}^*; \mathbf{x}, \mathcal{D}) \\
&+ \frac{1}{2}(\widehat{\boldsymbol{\theta}}(\mathbf{x}) - \boldsymbol{\theta}^*(\mathbf{x}))^T \nabla^2 \mathcal{L}_\lambda(\widetilde{\boldsymbol{\theta}}; \mathbf{x}, \mathcal{D})(\widehat{\boldsymbol{\theta}}(\mathbf{x}) - \boldsymbol{\theta}^*(\mathbf{x})) \\
&\leq \mathcal{L}_\lambda(\boldsymbol{\theta}^*; \mathbf{x}, \mathcal{D}) \,.
\end{aligned}
$$

for some $\widetilde{\boldsymbol{\theta}}(\mathbf{x})$ between $\widehat{\boldsymbol{\theta}}(\mathbf{x}), \boldsymbol{\theta}^*(\mathbf{x})$. As a result,

$$
\begin{aligned}
\frac{1}{2}\lambda_{\min}\left(\nabla^2 \mathcal{L}_\lambda(\widetilde{\boldsymbol{\theta}}; \mathbf{x}, \mathcal{D})\right) \|\widehat{\boldsymbol{\theta}}(\mathbf{x}) - \boldsymbol{\theta}^*(\mathbf{x})\|_2^2 &\leq \left|(\widehat{\boldsymbol{\theta}}(\mathbf{x}) - \boldsymbol{\theta}^*(\mathbf{x}))^T \nabla \mathcal{L}_\lambda(\boldsymbol{\theta}^*; \mathbf{x}, \mathcal{D})\right| \\
&\leq \|\widehat{\boldsymbol{\theta}}(\mathbf{x}) - \boldsymbol{\theta}^*(\mathbf{x})\|_2 \|\nabla \mathcal{L}_\lambda(\boldsymbol{\theta}^*; \mathbf{x}, \mathcal{D})\|_2
\end{aligned}
$$

and

$$
\|\widehat{\boldsymbol{\theta}}(\mathbf{x}) - \boldsymbol{\theta}^*(\mathbf{x})\|_2 \leq \frac{2\|\nabla \mathcal{L}_\lambda(\boldsymbol{\theta}^*; \mathbf{x}, \mathcal{D})\|_2}{\lambda_{\min}\left(\nabla^2 \mathcal{L}_\lambda(\widetilde{\boldsymbol{\theta}}; \mathbf{x}, \mathcal{D})\right)} \,.
$$

We then take the supremum over $\Omega$, take $\inf_{\mathbf{x} \in \Omega} \lambda_{\min, \perp}\left(\nabla^2 \mathcal{L}_\lambda(\boldsymbol{\theta}; \mathbf{x}, \mathcal{D})\right) \geq \lambda$, and apply Lemma B.3. Then

$$
\begin{aligned}
\sup_{\mathbf{x} \in \Omega} \|\widehat{\boldsymbol{\theta}}(\mathbf{x}) - \boldsymbol{\theta}^*(\mathbf{x})\|_2 &\leq \frac{2\sup_{\mathbf{x} \in \Omega} \|\nabla \mathcal{L}_\lambda(\boldsymbol{\theta}^*; \mathbf{x}, \mathcal{D})\|_2}{\inf_{\mathbf{x} \in \Omega} \lambda_{\min}\left(\nabla^2 \mathcal{L}_\lambda(\widetilde{\boldsymbol{\theta}}; \mathbf{x}, \mathcal{D})\right)} \\
&\lesssim \frac{\lambda + 2^n p \sqrt{n}\sqrt{d\log(2^n pLh^{-d})/2^n pLh^d} + 2^n p \sqrt{n} Cdh^2}{\lambda} \\
&= 1 + \sqrt{n}\left(\frac{2^n p\sqrt{d\log(2^n Lh^{-d})/2^n pLh^d} + 2^n p Cdh^2}{\lambda}\right) \,.
\end{aligned}
$$

Selecting $\lambda \asymp 2^n p\sqrt{\frac{d\log(2^n pLh^{-d})}{2^n pLh^d}} + 2^n pdh^2$ completes the argument.

## F    PROOF OF LEMMA C.2

Throughout the following proofs, we will encounter bounds including the term $\sup_{\mathbf{x}\in\Omega}||\widehat{\boldsymbol{\theta}}(\mathbf{x}) - \boldsymbol{\theta}^*(\mathbf{x})||_\infty$. By Theorem 3.1 (in high probability, under the aforementioned conditions), we upperbound this as

$$\sup_{\mathbf{x}\in\Omega}||\widehat{\boldsymbol{\theta}}(\mathbf{x}) - \boldsymbol{\theta}^*(\mathbf{x})||_\infty \leq \sup_{\mathbf{x}\in\Omega}||\widehat{\boldsymbol{\theta}}(\mathbf{x}) - \boldsymbol{\theta}^*(\mathbf{x})||_2 \lesssim \sqrt{nd}\left(\frac{\log(2^n pL)}{2^n pL}\right)^{2/4+d} .$$

We will also commonly make use of Remark 3.2, specifically that selecting $h \asymp [2^n pL/\log(2^n pL)]^{-1/4+d}$ implies

$$\nu_h + dh^2 \lesssim \sqrt{d}\left(\frac{\log(2^n pL)}{2^n pL}\right)^{2/4+d} .$$

We can now begin the proof of Lemma C.2.

*Proof.* The result controlling $\mathbf{B}_1$ is contained in Section E.3, the proof of Lemma B.3. For control of $\mathbf{B}_2, \mathbf{B}_3$, and $\mathbf{B}_4$, we make use of decompositions previously derived in Shen et al. (2023), within which we direct the reader to Section F.6 for further details. These decompositions require $\sup_{\mathbf{x}\in\Omega}||\widehat{\boldsymbol{\theta}}(\mathbf{x}) - \boldsymbol{\theta}^*(\mathbf{x})||_\infty \leq c$ for $c > 0$, which follows from the scaling condition stated in Theorem 3.1. Lastly, for brevity we use the shorthand throughout $\varphi = \sqrt{d}\left(\log(2^n pL)/2^n pL\right)^{2/4+d}$.

### F.1    $\mathbf{B}_2$

$$\sup_{\mathbf{x}\in\Omega}\left|\left[\nabla\mathcal{L}(\widehat{\boldsymbol{\theta}};\mathbf{x},\mathcal{D}) - \nabla\mathcal{L}(\boldsymbol{\theta}^*;\mathbf{x},\mathcal{D}) - \nabla^2\mathcal{L}(\boldsymbol{\theta}^*;\mathbf{x},\mathcal{D})(\widehat{\boldsymbol{\theta}}(\mathbf{x}) - \boldsymbol{\theta}^*(\mathbf{x}))\right]_i\right|$$

$$\leq \sup_{\mathbf{x}\in\Omega}||\widehat{\boldsymbol{\theta}}(\mathbf{x}) - \boldsymbol{\theta}^*(\mathbf{x})||_\infty^2 \sup_{\mathbf{x}\in\Omega}\sum_{\mathcal{S}\in\boldsymbol{\mathcal{E}}_i}\frac{1}{L}\sum_{\ell\in[L]}K_{\mathbf{H}}(\mathbf{X}_\mathcal{S}^\ell - \mathbf{x})\sum_{i\in\mathcal{S}^+}\left(\frac{e^{\theta_i^*(\mathbf{x})}}{\sum_{j\in\mathcal{S}}e^{\theta_j^*(\mathbf{x})}}\right)$$

$$\leq \sup_{\mathbf{x}\in\Omega}||\widehat{\boldsymbol{\theta}}(\mathbf{x}) - \boldsymbol{\theta}^*(\mathbf{x})||_\infty^2 \cdot \sup_{\mathbf{x}\in\Omega}\sum_{\mathcal{S}\in\boldsymbol{\mathcal{E}}_i}\frac{1}{L}\sum_{\ell\in[L]}K_{\mathbf{H}}(\mathbf{X}_\mathcal{S}^\ell - \mathbf{x})$$

$$\leq \sup_{\mathbf{x}\in\Omega}||\widehat{\boldsymbol{\theta}}(\mathbf{x}) - \boldsymbol{\theta}^*(\mathbf{x})||_\infty^2\left(2^n pC_f + 2^n pcdh^2 + 2^n p\sqrt{\frac{d\log(2^n pL/h^d)}{2^n pLh^d}}\right)$$

with probability at least $1 - O(n^{-10})$ given $2^n p \gtrsim n\log n$ and Lemma H.3. Selecting $h \asymp (2^n pL)^{-1/d+4}$ and using the result in Remark 3.2, the above is further upper-bounded by

$$\sup_{\mathbf{x}\in\Omega}||\widehat{\boldsymbol{\theta}}(\mathbf{x}) - \boldsymbol{\theta}^*(\mathbf{x})||_2^2\left(2^n pC_f + 2^n pcdh^2 + 2^n p\sqrt{\frac{d\log(2^n pL/h^d)}{2^n pLh^d}}\right)$$

$$\lesssim 2^n pn\,\varphi^2\left(1 + \varphi\right) .$$

The above bound holds with probability at least $1 - O(n^{-10})$ jointly over events of Lemma H.3 and Remark 3.2.

### F.2    $\mathbf{B}_3$

We aim to control

$$\sup_{\mathbf{x}\in\Omega}||\nabla^2\mathcal{L}(\widehat{\boldsymbol{\theta}};\mathbf{x},\mathcal{D}) - \nabla^2\mathcal{L}(\boldsymbol{\theta}^*;\mathbf{x},\mathcal{D})||_\infty = \max_{i\in[n]_+}||[\nabla^2\mathcal{L}(\widehat{\boldsymbol{\theta}};\mathbf{x},\mathcal{D})]_i - [\nabla^2\mathcal{L}(\boldsymbol{\theta}^*;\mathbf{x},\mathcal{D})]_i||_1 .$$

Thus we study the generic $i$th coordinate, and conclude the upper-bound holds for all $i \in [n]_+$.

$$||[\nabla^2 \mathcal{L}(\widehat{\boldsymbol{\theta}}; \mathbf{x}, \mathcal{D})]_i - [\nabla^2 \mathcal{L}(\boldsymbol{\theta}^*; \mathbf{x}, \mathcal{D})]_i||_1$$

$$\lesssim \left\| \sum_{\mathcal{S} \in \boldsymbol{\mathcal{E}}_i} \frac{1}{L} \sum_{\ell \in [L]} K_{\mathbf{H}}(\mathbf{X}_\mathcal{S}^\ell - \mathbf{x}) \sum_{i \in \mathcal{S}} \left[ \frac{e^{\widehat{\theta}_i(\mathbf{x})} \sum_{j \in \mathcal{S}^{-i}} e^{\widehat{\theta}_j(\mathbf{x})}}{\left(\sum_{j \in \mathcal{S}} e^{\widehat{\theta}_j(\mathbf{x})}\right)^2} \mathbf{e}_i - \sum_{j \in \mathcal{S}^{-i}} \frac{e^{\widehat{\theta}_i(\mathbf{x})} e^{\widehat{\theta}_j(\mathbf{x})}}{\left(\sum_{j \in \mathcal{S}} e^{\widehat{\theta}_j(\mathbf{x})}\right)^2} \mathbf{e}_j \right] - [\nabla^2 \mathcal{L}(\boldsymbol{\theta}^*; \mathbf{x}, \mathcal{D})]_i \right\|_1$$

$$\lesssim \sup_{\mathbf{x} \in \Omega} ||\widehat{\boldsymbol{\theta}}(\mathbf{x}) - \boldsymbol{\theta}^*(\mathbf{x})||_\infty \sup_{\mathbf{x} \in \Omega} \sum_{\mathcal{S} \in \boldsymbol{\mathcal{E}}_i} \frac{1}{L} \sum_{\ell \in [L]} K_{\mathbf{H}}(\mathbf{X}_\mathcal{S}^\ell - \mathbf{x})$$

$$\lesssim \sup_{\mathbf{x} \in \Omega} ||\widehat{\boldsymbol{\theta}}(\mathbf{x}) - \boldsymbol{\theta}^*(\mathbf{x})||_\infty \left( 2^n pc + 2^n pdh^2 + 2^n p \frac{\sqrt{d \log(2^n pL)}}{(2^n pL)^{2/+d}} \right)$$

$$\lesssim 2^n p \sqrt{n} \, \varphi \, (1 + \varphi)$$

The first inequality holds by the analysis in Shen et al. (2023) Section F.6. The remaining work holds with high probability by $h$-selection via Remark 3.2 and Lemma H.3.

### F.3 $\mathbf{B}_4$

Assume there exists some $C_n > 0$, varying only in $n$, such that $C_n \sqrt{n} \gtrsim 2^n p/n$. Then,

$$\sup_{\mathbf{x} \in \Omega} \left\| \begin{pmatrix} \nabla^2 \mathcal{L}(\widehat{\boldsymbol{\theta}}) & \mathbf{1} \\ \mathbf{1}^T & 0 \end{pmatrix}^{-1} - \begin{pmatrix} \nabla^2 \mathcal{L}(\boldsymbol{\theta}^*) & \mathbf{1} \\ \mathbf{1}^T & 0 \end{pmatrix}^{-1} \right\|_2$$

$$\leq \sup_{\mathbf{x} \in \Omega} \left\| \begin{pmatrix} \nabla^2 \mathcal{L}(\widehat{\boldsymbol{\theta}}) & C_n \mathbf{1} \\ C_n \mathbf{1}^T & 0 \end{pmatrix}^{-1} \right\|_2 \left\| \begin{pmatrix} \nabla^2 \mathcal{L}(\boldsymbol{\theta}^*) & C_n \mathbf{1} \\ C_n \mathbf{1}^T & 0 \end{pmatrix}^{-1} \right\|_2 \left\| \begin{pmatrix} \nabla^2 \mathcal{L}(\widehat{\boldsymbol{\theta}}) & C_n \mathbf{1} \\ C_n \mathbf{1}^T & 0 \end{pmatrix} - \begin{pmatrix} \nabla^2 \mathcal{L}(\boldsymbol{\theta}^*) & C_n \mathbf{1} \\ C_n \mathbf{1}^T & 0 \end{pmatrix} \right\|_2$$

$$\leq \sup_{\mathbf{x} \in \Omega} \lambda_{\min,\perp} \left( \nabla^2 \mathcal{L}(\widehat{\boldsymbol{\theta}}; \mathbf{x}, \mathcal{D}) \right)^{-1} \sup_{\mathbf{x} \in \Omega} \lambda_{\min,\perp} \left( \nabla^2 \mathcal{L}(\boldsymbol{\theta}^*; \mathbf{x}, \mathcal{D}) \right)^{-1} \sup_{\mathbf{x} \in \Omega} ||\nabla^2 \mathcal{L}(\widehat{\boldsymbol{\theta}}) - \nabla^2 \mathcal{L}(\boldsymbol{\theta}^*)||_\infty$$

$$\lesssim \frac{n^2}{(2^n p)^2} \left( 2^n p + 2^n p \varphi \right) \varphi$$

$$\lesssim \frac{n^2}{2^n p} \left( 1 + \varphi \right) \varphi \, .$$

holds with probability at least $1 - O(n^{-10})$. The probabilistic nature holds from the penultimate line, over Lemma C.2's bounds on $\mathbf{B}_3, \mathbf{B}_5$.

### F.4 $\mathbf{B}_5$

As noted in the proof of $\mathbf{B}_4$, selection of $h \asymp (2^n pL)^{-1/4+d}$ and Lemma B.2 yields $\lambda_{\min,\perp} \left( \nabla^2 \mathcal{L}(\boldsymbol{\theta}) \right) \gtrsim 2^n p/n$, and thus, $\sup_{\mathbf{x} \in \Omega} \left\| \nabla^2 \mathcal{L}(\boldsymbol{\theta}; \mathbf{x}, \mathcal{D})^\dagger \right\|_2 \lesssim \frac{n}{2^n p}$ in high probability.

$\square$

# G   Proof of Lemmas from Section D

## G.1   Lemma G.1 and proof

First, it is useful to state results similar to lemmas C.2 and D.1 for the log-likelihood $\Lambda(\boldsymbol{\theta}^*)$ and bounds on differences with our local approximation $\mathcal{L}(\boldsymbol{\theta}^*)$.

**Lemma G.1.** Let $\varphi = (\log(2^n pL)/2^n pL)^{2/4+d}$. Under the same conditions of Lemma D.1, the following results hold:

$$\sup_{\mathbf{x}\in\Omega}\sup_{\boldsymbol{\theta}\in\Theta}||\nabla^2\Lambda(\boldsymbol{\theta};\mathbf{x},\mathcal{D})||_2 \lesssim \frac{n}{2^n p} \tag{G.1}$$

$$\sup_{\mathbf{x}\in\Omega}||\nabla\mathcal{L}(\boldsymbol{\theta}^*) - \nabla\Lambda(\boldsymbol{\theta}^*)||_2 \lesssim 2^n p\sqrt{n}\varphi \tag{G.2}$$

$$\sup_{\mathbf{x}\in\Omega}||\nabla^2\mathcal{L}(\boldsymbol{\theta}^*) - \nabla^2\Lambda(\boldsymbol{\theta}^*)||_2 \lesssim 2^n p\varphi \tag{G.3}$$

$$\sup_{\mathbf{x}\in\Omega}||\nabla^2\mathcal{L}(\boldsymbol{\theta}^*)^\dagger - \nabla^2\Lambda(\boldsymbol{\theta}^*)^\dagger||_2 \lesssim n^2\varphi/2^n p \tag{G.4}$$

$$\sup_{\mathbf{x}\in\Omega}||\nabla\Lambda(\boldsymbol{\theta}^*)||_2 \lesssim 2^n p\sqrt{n}\varphi \tag{G.5}$$

holds with probability at least $1 - O(n^{-10})$.

*Proof.* The results of Lemma G.1 mirror those between Lemmas C.2 and D.1. Rather than bounding differences in terms of $\boldsymbol{\theta}^*, \widehat{\boldsymbol{\theta}}$, we now bound differences based on our local approximation to the log likelihood, i.e. $\mathcal{L}(\boldsymbol{\theta}^*), \Lambda(\boldsymbol{\theta}^*)$. Arguments for these bounds are equivalent to those in the proofs of these earlier lemmas, with minor alterations, described below.

Firstly, note that by an analogous (but somewhat simpler) analysis as in Section E.2. That is, $\nabla^2\Lambda(\boldsymbol{\theta};\mathbf{x},\mathcal{D}) \preccurlyeq \mathbf{L}_{\boldsymbol{\mathcal{E}}}(\mathbf{x}), \forall\boldsymbol{\theta}\in\Theta$ as defined in Equation B.2. Equation G.1 then holds by identical analysis as in the proof of Lemma D.1. Equation G.2 follows and Lemma B.3 Equation G.5 above. Equation G.3 holds by a similar argument as in Section F.2 , with an expansion of $\boldsymbol{\theta}^*$ by our smoothness conditions on $\boldsymbol{\theta}\in\Theta$ and assumed compactness of $\Omega$. Equation G.4 holds by a similar argument as Section F.3 proof (of a similar result in Lemma C.2). Lastly, Equation G.5 holds by a similarly yet simpler analysis as in Section E.3, forgoing the mean-deviation bound as $\mathbb{E}[\Lambda(\boldsymbol{\theta}^*) = 0], \forall\mathbf{x}\in\Omega$ and requiring the kernel density results H.3.

$\square$

## G.2   Lemma G.2 and Proof

**Lemma G.2** (Conditional Variance of Weighed Gaussian)**.** First, define the useful short-hand

$$\widehat{\alpha}_k^{\ell,\mathcal{S}}(\mathbf{x},\mathcal{D}) = \frac{1}{L}K_{\mathbf{H}}(\mathbf{X}_{\mathcal{S}}^\ell - \mathbf{x})\sum_{i\in\mathcal{S}^+}\frac{\left[y_{\mathcal{S}}^{(i,\ell)} - \frac{e^{\widehat{\theta}_i(\mathbf{x})}}{\sum_{j\in\mathcal{S}_+}e^{\widehat{\theta}_j(\mathbf{x})}}\right]\left[\nabla^2\mathcal{L}(\widehat{\boldsymbol{\theta}})^\dagger\widehat{\mathbf{v}}_k(\mathbf{x})\right]_i}{\sqrt{\widehat{\mathbf{v}}_k(\mathbf{x})^T\nabla^2\mathcal{L}(\widehat{\boldsymbol{\theta}})^\dagger\widehat{\mathbf{v}}_k(\mathbf{x})}}$$

$$\alpha_{0,k}^{\ell,\mathcal{S}}(\mathbf{x},\mathcal{D}) = \frac{1}{L}K_{\mathbf{H}}(\mathbf{X}_{\mathcal{S}}^\ell - \mathbf{x})\sum_{i\in\mathcal{S}^+}\frac{\left[y_{\mathcal{S}}^{(i,\ell)} - \frac{e^{\theta_i^*(\mathbf{x})}}{\sum_{j\in\mathcal{S}_+}e^{\theta_j(\mathbf{x})}}\right]\left[\nabla^2\Lambda(\boldsymbol{\theta}^*)^\dagger\widehat{\mathbf{v}}_k(\mathbf{x})\right]_i}{\sqrt{\mathbf{v}_k(\mathbf{x})^T\nabla^2\Lambda(\boldsymbol{\theta}^*)^\dagger\mathbf{v}_k(\mathbf{x})}}$$

We then collect these values into vectors $\widehat{\alpha}_k(\mathbf{x},\mathcal{D}), \alpha_{0,k}(\mathbf{x},\mathcal{D})$ over $(\ell,\mathcal{S})\in[L]\times\boldsymbol{\mathcal{E}}$, and let $Z$ be a vector of independent, standard Gaussians. Now, suppose the conditions of Theorem 3.1 hold and $n^3\log(2^n pL) \lesssim (2^n pL)^\varepsilon$ for some $\varepsilon < 2/(4+d)$. Then conditional on the events of Lemmas D.1 and G.1,

$$\sigma^2(\mathcal{D}) := \sup_{\mathbf{x}\in\Omega}\max_{k\in[n]}V\left(\sqrt{Lh^d}[\widehat{\alpha}_k(\mathbf{x},\mathcal{D}) - \alpha_{0,k}(\mathbf{x},\mathcal{D})]^T Z \mid \mathcal{D}\right) \le n^{3/2}\varphi \ .$$

Here we again take $\varphi = \left(\frac{\log(2^n pL)}{2^n pL}\right)^{2/4+d}$.

*Proof.* We begin by noting that by conditioning on $\mathcal{D}$, the only randomness in the expression is in $Z$.

Now note that

$$\widehat{\alpha}_k^{\ell,\mathcal{S}}(\mathbf{x},\mathcal{D}) - \alpha_{0,k}^{\ell,\mathcal{S}}(\mathbf{x},\mathcal{D}) \lesssim \frac{1}{L}K_{\mathbf{H}}(\mathbf{X}_{\mathcal{S}}^\ell - \mathbf{x}) \underbrace{\left( \frac{||\nabla^2\mathcal{L}(\widehat{\boldsymbol{\theta}})^\dagger||_2 ||\widehat{\mathbf{v}}_k(\mathbf{x})||_2}{\sqrt{\widehat{\mathbf{v}}_k(\mathbf{x})^T \nabla^2\mathcal{L}(\widehat{\boldsymbol{\theta}})^\dagger \widehat{\mathbf{v}}_k(\mathbf{x})}} - \frac{||\nabla^2\Lambda(\widehat{\boldsymbol{\theta}})^\dagger||_2 ||\mathbf{v}_k(\mathbf{x})||_2}{\sqrt{\mathbf{v}_k(\mathbf{x})^T \nabla^2\nabla^2\mathcal{L}(\widehat{\boldsymbol{\theta}})^\dagger \mathbf{v}_k(\mathbf{x})}} \right)}_{:=\psi_k(\mathbf{x})}$$

By applying Lemmas C.2 and G.1 we have $\sup_{\mathbf{x}\in\Omega} \max_{k\in[n]} \psi_k(\mathbf{x}) \lesssim n^{3/2}\varphi/\sqrt{2^n p}$.

$$\sup_{\mathbf{x}\in\Omega} \max_{k\in[n]} \sigma^2(\mathcal{D}) := \sup_{\mathbf{x}\in\Omega} \max_{k\in[n]} V\left( \sqrt{Lh^d}[\widehat{\alpha}_k(\mathbf{x},\mathcal{D}) - \alpha_{0,k}(\mathbf{x},\mathcal{D})]^T Z \mid \mathcal{D} \right)$$

$$= \sup_{\mathbf{x}\in\Omega} \max_{k\in[n]} \frac{h^d}{L}[\psi_k(\mathbf{x})]^2 \sum_{\mathcal{S}\in\mathcal{E}} \sum_{\ell\in[L]} K_{\mathbf{H}}(\mathbf{X}_{\mathcal{S}}^\ell - \mathbf{x})^2$$

$$\leq \frac{n^3\varphi^2}{2^n pL} \sup_{\mathbf{x}\in\Omega} \sum_{\mathcal{S}\in\mathcal{E}} \sum_{\ell\in[L]} h^{-d}K^2(\mathbf{X}_{\mathcal{S}}^\ell - \mathbf{x})/h^d$$

$$\lesssim n^3\varphi^2(C_f + h^2)$$

$$\lesssim n^3\varphi^2$$

The penultimate inequality holds by applying Lemma H.3 for new kernel $K^2(\cdot)$ and boundedness of the density $f_{\mathbf{X}}$. This event holds with probability at least $1 - O(n^{-10})$ on the events of Lemmas D.1 and G.1 and results in Section H.

$\square$

### G.3 PROOF OF LEMMA D.1

*Proof.* Again, we make use of the shorthand $\varphi = (\log(2^n pL)/2^n pL)^{2/4+d}$ Equation D.1 holds given Lemma B.2. Equation D.2 holds by the same argument as Shen et al. (2023), with the additional uniform control over $\Omega$ by our definitions of $c_r, c_\theta$. Equation D.3 holds via a Taylor series expansion and remainder control given $\varphi = o(1)$ for large enough $n, L$:

$$\sup_{\mathbf{x}\in\Omega} ||\widehat{\mathbf{v}}_k(\mathbf{x}) - \mathbf{v}_k(\mathbf{x})||_2 \leq \sup_{\mathbf{x}\in\Omega} \left|\left| \text{diag}\left( \mathbf{v}_k(\mathbf{x}) \right)\left[ \widehat{\boldsymbol{\theta}}(\mathbf{x}) - \boldsymbol{\theta}^*(\mathbf{x}) \right] \right|\right|_2 + o(\varphi^2).$$

Equation D.4 holds given Equation D.3 and results from Lemma C.2:

$$||\widehat{\mathbf{v}}_k(\mathbf{x})^T \nabla^2\mathcal{L}(\widehat{\boldsymbol{\theta}})^\dagger - \mathbf{v}_k(\mathbf{x})^T \nabla^2\mathcal{L}(\boldsymbol{\theta}^*)^\dagger||_2$$

$$\leq ||(\widehat{\mathbf{v}}_k(\mathbf{x}) - \mathbf{v}_k(\mathbf{x}))||_2 ||\nabla^2\mathcal{L}(\widehat{\boldsymbol{\theta}})^\dagger||_2 + ||\mathbf{P}_{\mathbf{1}}^\perp \mathbf{v}_k(\mathbf{x})||_2 ||\nabla^2\mathcal{L}(\boldsymbol{\theta}^*)^\dagger - \nabla^2\mathcal{L}(\widehat{\boldsymbol{\theta}})||_2$$

$$\lesssim \frac{n^{3/2}\varphi}{2^n p}||\mathbf{v}_k(\mathbf{x})||_\infty + \frac{n^2\varphi}{2^n p}||\mathbf{P}_{\mathbf{1}}^\perp \mathbf{v}_k(\mathbf{x})||_2$$

Equation D.5 holds by

$$\left| \widehat{\mathbf{v}}_k(\mathbf{x})^T \nabla^2\mathcal{L}(\widehat{\boldsymbol{\theta}})^\dagger \widehat{\mathbf{v}}_k(\mathbf{x}) - \mathbf{v}_k(\mathbf{x})^T \nabla^2\mathcal{L}(\boldsymbol{\theta}^*)^\dagger \mathbf{v}_k(\mathbf{x}) \right|$$

$$\leq |\widehat{\mathbf{v}}_k(\mathbf{x})^T (\nabla^2\mathcal{L}(\widehat{\boldsymbol{\theta}}(\mathbf{x}))^\dagger - \nabla^2\mathcal{L}(\boldsymbol{\theta}^*(\mathbf{x}))^\dagger)\widehat{\mathbf{v}}_k(\mathbf{x})| + |(\widehat{\mathbf{v}}_k(\mathbf{x}) - \mathbf{v}_k(\mathbf{x}))^T \nabla^2\mathcal{L}(\boldsymbol{\theta}^*(\mathbf{x}))^\dagger \mathbf{v}_k(\mathbf{x})|$$

$$+ |(\widehat{\mathbf{v}}_k(\mathbf{x}) - \mathbf{v}_k(\mathbf{x}))^T \nabla^2\mathcal{L}(\boldsymbol{\theta}^*(\mathbf{x}))^\dagger \widehat{\mathbf{v}}_k(\mathbf{x})|$$

$$\lesssim ||\nabla^2\mathcal{L}(\widehat{\boldsymbol{\theta}})^\dagger - \nabla^2\mathcal{L}(\boldsymbol{\theta}^*)^\dagger||_2 ||\mathbf{P}_{\mathbf{1}}^\perp \mathbf{v}_k(\mathbf{x})||_2^2 + ||\widehat{\mathbf{v}}_k(\mathbf{x}) - \mathbf{v}_k(\mathbf{x})||_2 ||\nabla^2\mathcal{L}(\boldsymbol{\theta}^*)^\dagger||_2 ||\mathbf{P}_{\mathbf{1}}^\perp \mathbf{v}_k(\mathbf{x})||_2$$

$$\lesssim \frac{n^2}{2^n p}(\varphi + \varphi^2)||\mathbf{P}_{\mathbf{1}}^\perp \mathbf{v}_k(\mathbf{x})||_2^2 + ||\mathbf{P}_{\mathbf{1}}^\perp \mathbf{v}_k(\mathbf{x})||_2 \frac{n}{2^n p}\varphi\sqrt{n}||\mathbf{v}_k(\mathbf{x})||_\infty$$

with the final line holding in high probability by C.2 and $\sqrt{n}\varphi = o(1)$. Lastly, Equation D.6 holds, by

$$\left| \frac{1}{\sqrt{\widehat{\mathbf{v}}_k(\mathbf{x})^T \nabla^2 \mathcal{L}(\widehat{\boldsymbol{\theta}})^\dagger \widehat{\mathbf{v}}_k(\mathbf{x})}} - \frac{1}{\sqrt{\mathbf{v}_k(\mathbf{x})^T \nabla^2 \mathcal{L}(\boldsymbol{\theta}^*)^\dagger \mathbf{v}_k(\mathbf{x})}} \right|$$

$$\lesssim (\mathbf{v}_k(\mathbf{x}) \nabla^2 \mathcal{L}(\boldsymbol{\theta}^*)^\dagger \mathbf{v}_k(\mathbf{x}))^{-1/2} \left| \frac{\widehat{\mathbf{v}}_k(\mathbf{x})^T \nabla^2 \mathcal{L}(\widehat{\boldsymbol{\theta}})^\dagger \widehat{\mathbf{v}}_k(\mathbf{x}) - \mathbf{v}_k(\mathbf{x})^T \nabla^2 \mathcal{L}(\boldsymbol{\theta}^*)^\dagger \mathbf{v}_k(\mathbf{x})}{\mathbf{v}_k(\mathbf{x})^T \nabla^2 \mathcal{L}(\boldsymbol{\theta}^*)^\dagger \mathbf{v}_k(\mathbf{x})} \right|$$

$$\lesssim (\mathbf{v}_k(\mathbf{x}) \nabla^2 \mathcal{L}(\boldsymbol{\theta}^*)^\dagger \mathbf{v}_k(\mathbf{x}))^{-1/2} \left[ n^2(\varphi + \varphi^2) + n^{3/2}\varphi \frac{||\mathbf{v}_k(\mathbf{x})||_\infty}{||\mathbf{P}_{\mathbf{1}}^\perp \mathbf{v}_k(\mathbf{x})||_2} \right]$$

given Equations D.1 and D.5. The stated result then holds given $\varphi = o(1)$. $\qquad\square$

## G.4  Proof of Lemma D.2 $\mathbb{P}(|T - T_0| \geq \zeta_1) \leq \zeta_2$

*Proof.* Define

$$T_{\mathbf{x}} = \sup_{\mathbf{x} \in \Omega} \max_{k \in [n]} \sqrt{Lh^d} \left( \frac{-\mathbf{v}_k(\mathbf{x})^T \nabla^2 \mathcal{L}(\boldsymbol{\theta}^*)^\dagger}{\sqrt{\mathbf{v}_k(\mathbf{x})^T \nabla^2 \mathcal{L}(\boldsymbol{\theta}^*)^\dagger \mathbf{v}_k(\mathbf{x})}} \nabla \mathcal{L}(\boldsymbol{\theta}^*) \right)$$

$$W_{\mathbf{x}} = \sup_{\mathbf{x} \in \Omega} \max_{k \in [n]} \sqrt{Lh^d} \left( \frac{-\mathbf{v}_k(\mathbf{x})^T \nabla^2 \mathcal{L}(\boldsymbol{\theta}^*)^\dagger}{\sqrt{\mathbf{v}_k(\mathbf{x})^T \nabla^2 \mathcal{L}(\boldsymbol{\theta}^*) \mathbf{v}_k(\mathbf{x})}} \nabla \widetilde{\mathcal{L}}(\boldsymbol{\theta}^*) \right) .$$

We again consider a truncation-style argument on the result of Lemma 3.1 (and $h$-selection in Remark 3.2), so that

$$\mathbb{P}(|T - T_0| \geq \zeta_1) \leq \mathbb{P}\left( |T - T_0| \geq \zeta_1 \mid ||\widehat{\boldsymbol{\theta}}(\mathbf{x}) - \boldsymbol{\theta}^*(\mathbf{x})||_2 \lesssim \sqrt{nd} \left( \frac{\log(2^n pL)}{2^n pL} \right)^{2/4+d} \right) + O(n^{-10})$$

We further decompose $|T - T_0| \leq |T - T_{\mathbf{x}}| + |T_{\mathbf{x}} - T_0|$. We will show each of these terms is well controlled by some $\zeta_1$ with probability at least $! - O(n^{-10})$. First analyzing $|T - T_{\mathbf{x}}|$.

$$|T - T_{\mathbf{x}}| \leq \sqrt{Lh^d} \sup_{\mathbf{x} \in \Omega} \max_{k \in [n]} \left| \frac{\widehat{\Delta}_k(\mathbf{x}) - \Delta_k(\mathbf{x})}{\sqrt{\widehat{\mathbf{v}}_k(\mathbf{x})^T \nabla^2 \mathcal{L}(\widehat{\boldsymbol{\theta}}) \widehat{\mathbf{v}}_k(\mathbf{x})}} - \frac{-\mathbf{v}_k(\mathbf{x})^T \nabla^2 \mathcal{L}(\boldsymbol{\theta}^*)}{\sqrt{\mathbf{v}_k(\mathbf{x})^T \nabla^2 \mathcal{L}(\boldsymbol{\theta}^*) \mathbf{v}_k(\mathbf{x})}} \nabla \mathcal{L}(\boldsymbol{\theta}^*) \right|$$

$$\leq \sqrt{Lh^d} \sup_{\mathbf{x} \in \Omega} \max_{k \in [n]} ||\mathbf{P}_{\mathbf{1}}^\perp \mathbf{v}_k(\mathbf{x})||_2 ||\widehat{\boldsymbol{\theta}}^d(\mathbf{x}) - \boldsymbol{\theta}^*(\mathbf{x})||_2 \left| \frac{1}{\sqrt{\widehat{\mathbf{v}}_k(\mathbf{x})^T \nabla^2 \mathcal{L}(\widehat{\boldsymbol{\theta}}) \widehat{\mathbf{v}}_k(\mathbf{x})}} - \frac{1}{\sqrt{\mathbf{v}_k(\mathbf{x})^T \nabla^2 \mathcal{L}(\boldsymbol{\theta}^*) \mathbf{v}_k(\mathbf{x})}} \right|$$

$$+ \sqrt{Lh^d} \sup_{\mathbf{x} \in \Omega} \max_{k \in [n]} \frac{||\mathbf{P}_{\mathbf{1}}^\perp \mathbf{v}_k(\mathbf{x})||_2 ||\mathbf{R}_\theta(\mathbf{x})||_2 + ||\mathbf{v}_k(\mathbf{x})||_\infty ||\widehat{\boldsymbol{\theta}}^d(\mathbf{x}) - \boldsymbol{\theta}^*(\mathbf{x})||_2^2}{\sqrt{\mathbf{v}_k(\mathbf{x})^T \nabla^2 \mathcal{L}(\boldsymbol{\theta}^*)^\dagger \mathbf{v}_k(\mathbf{x})}}$$

$$\leq \sqrt{Lh^d} \sup_{\mathbf{x} \in \Omega} \max_{k \in [n]} \left\{ \underbrace{\frac{||\mathbf{P}_{\mathbf{1}}^\perp \mathbf{v}_k(\mathbf{x})||_2 ||\widehat{\boldsymbol{\theta}}^d(\mathbf{x}) - \boldsymbol{\theta}^*(\mathbf{x})||_2}{\sqrt{\mathbf{v}_k(\mathbf{x})^T \nabla^2 \mathcal{L}(\boldsymbol{\theta}^*)^\dagger \mathbf{v}_k(\mathbf{x})}}}_{:=A_1} \underbrace{\left| \frac{\widehat{\mathbf{v}}_k(\mathbf{x})^T \nabla^2 \mathcal{L}(\widehat{\boldsymbol{\theta}})^\dagger \widehat{\mathbf{v}}_k(\mathbf{x}) - \mathbf{v}_k(\mathbf{x})^T \nabla^2 \mathcal{L}(\boldsymbol{\theta}^*)^\dagger \mathbf{v}_k(\mathbf{x})}{\mathbf{v}_k(\mathbf{x})^T \nabla^2 \mathcal{L}(\boldsymbol{\theta}^*)^\dagger \mathbf{v}_k(\mathbf{x})} \right|}_{:=A_2} \right.$$

$$\left. + \sqrt{Lh^d} \sup_{\mathbf{x} \in \Omega} \max_{k \in [n]} \underbrace{\frac{||\mathbf{P}_{\mathbf{1}}^\perp \mathbf{v}_k(\mathbf{x})||_2 ||\mathbf{R}_\theta(\mathbf{x})||_2 + ||\mathbf{v}_k(\mathbf{x})||_\infty ||\widehat{\boldsymbol{\theta}}^d(\mathbf{x}) - \boldsymbol{\theta}^*(\mathbf{x})||_2^2}{\sqrt{\mathbf{v}_k(\mathbf{x})^T \nabla^2 \mathcal{L}(\boldsymbol{\theta}^*)^\dagger \mathbf{v}_k(\mathbf{x})}}}_{:=A_3} \right\}$$

$$\leq \sqrt{Lh^d} \sup_{\mathbf{x} \in \Omega} \max_{k \in [n]} A_1 \sup_{\mathbf{x} \in \Omega} \max_{k \in [n]} A_2 + \sqrt{Lh^d} \sup_{\mathbf{x} \in \Omega} \max_{k \in [n]} A_3$$

By Lemma D.1, we have

$$\sup_{\mathbf{x} \in \Omega} \max_{k \in [n]} A_1 \lesssim n^2 \sqrt{d2^n p} \left( \frac{\log(2^n pL)}{2^n pL} \right)^{2/4+d}, \quad \sup_{\mathbf{x} \in \Omega} \max_{k \in [n]} A_2 \lesssim n\sqrt{d} \left( \frac{\log(2^n pL)}{2^n pL} \right)^{2/4+d},$$

$$\sup_{\mathbf{x} \in \Omega} \max_{k \in [n]} A_3 \lesssim n^{5/2} d \sqrt{2^n p} \left( \frac{\log(2^n pL)}{2^n pL} \right)^{4/4+d}$$

jointly with probability at least $1 - O(n^{-10})$. Combining the above, we have

$$\sup_{\mathbf{x} \in \Omega} \max_{k \in [n]} |T - T_0| \lesssim n^3 \sqrt{d} \left( \frac{\log(2^n pL)}{(2^n pL)} \right)^{4/4+d} \sqrt{2^n pLh^d} \lesssim n^3 \frac{d \log(2^n pL)}{(2^n pL)^{2/4+d}}$$

given $h \asymp (\log(2^n pL)/2^n pL)^{-1/4+d}$ with probability at least $1 - O(n^{-10})$ over events applied from Lemma D.1.

Now controlling $|T_{\mathbf{x}} - T_0|$:

$$|T_{\mathbf{x}} - T_0| = \left| \sup_{\mathbf{x} \in \Omega} \max_{k \in [n]} \sqrt{Lh^d} \frac{-\mathbf{v}_k(\mathbf{x})^T \nabla^2 \Lambda(\boldsymbol{\theta}^*)^\dagger}{\sqrt{\mathbf{v}_k(\mathbf{x})^T \nabla^2 \Lambda(\boldsymbol{\theta}^*) \mathbf{v}_k(\mathbf{x})}} \nabla \Lambda(\boldsymbol{\theta}^*) - \sup_{\mathbf{x} \in \Omega} \max_{k \in [n]} \sqrt{Lh^d} \frac{-\mathbf{v}_k(\mathbf{x})^T \nabla^2 \mathcal{L}(\boldsymbol{\theta}^*)^\dagger}{\sqrt{\mathbf{v}_k(\mathbf{x})^T \nabla^2 \mathcal{L}(\boldsymbol{\theta}^*)^\dagger \mathbf{v}_k(\mathbf{x})}} \nabla \mathcal{L}(\boldsymbol{\theta}^*) \right|$$

$$\leq \sup_{\mathbf{x} \in \Omega} \max_{k \in [n]} \left| \sqrt{Lh^d} \frac{-\mathbf{v}_k(\mathbf{x})^T \nabla^2 \Lambda(\boldsymbol{\theta}^*)^\dagger}{\sqrt{\mathbf{v}_k(\mathbf{x})^T \nabla^2 \Lambda(\boldsymbol{\theta}^*) \mathbf{v}_k(\mathbf{x})}} \nabla \Lambda(\boldsymbol{\theta}^*) - \sqrt{Lh^d} \frac{-\mathbf{v}_k(\mathbf{x})^T \nabla^2 \mathcal{L}(\boldsymbol{\theta}^*)^\dagger}{\sqrt{\mathbf{v}_k(\mathbf{x})^T \nabla^2 \mathcal{L}(\boldsymbol{\theta}^*)^\dagger \mathbf{v}_k(\mathbf{x})}} \nabla \mathcal{L}(\boldsymbol{\theta}^*) \right|$$

$$\lesssim \sqrt{Lh^d} \sup_{\mathbf{x} \in \Omega} \max_{k \in [n]} \frac{\left| \mathbf{v}_k(\mathbf{x})^T \nabla^2 \mathcal{L}(\boldsymbol{\theta}^*)^\dagger \nabla \mathcal{L}(\boldsymbol{\theta}^*) - \mathbf{v}_k(\mathbf{x})^T \nabla^2 \Lambda(\boldsymbol{\theta}^*)^\dagger \nabla \Lambda(\boldsymbol{\theta}^*) \right|}{\sqrt{\mathbf{v}_k^T(\mathbf{x}) \nabla^2 \Lambda(\boldsymbol{\theta}^*)^\dagger \mathbf{v}_k(\mathbf{x})}}$$

$$+ \sqrt{Lh^d} \sup_{\mathbf{x} \in \Omega} \max_{k \in [n]} \left| \mathbf{v}_k(\mathbf{x})^T \nabla^2 \mathcal{L}(\boldsymbol{\theta}^*)^\dagger \nabla \mathcal{L}(\boldsymbol{\theta}^*) \right| \left| (\mathbf{v}_k^T(\mathbf{x}) \nabla^2 \mathcal{L}(\boldsymbol{\theta}^*)^\dagger \mathbf{v}_k(\mathbf{x}))^{-1/2} - (\mathbf{v}_k^T(\mathbf{x}) \nabla^2 \Lambda(\boldsymbol{\theta}^*)^\dagger \mathbf{v}_k(\mathbf{x}))^{-1/2} \right|$$

Controlling the first term, as in Lemma D.1 and with Lemma G.1 we have $(\mathbf{v}_k(\mathbf{x})^T \nabla^2 \Lambda(\boldsymbol{\theta}^*)^\dagger \mathbf{v}_k(\mathbf{x}))^{-1} \gtrsim \sqrt{n} ||\mathbf{P}_{\mathbf{1}}^\perp \mathbf{v}_k(\mathbf{x})||_2 / 2^n p$, controlling the denominator. The numerator we control as

$$\left| \mathbf{v}_k(\mathbf{x})^T \nabla^2 \mathcal{L}(\boldsymbol{\theta}^*)^\dagger \nabla \mathcal{L}(\boldsymbol{\theta}^*) - \mathbf{v}_k(\mathbf{x})^T \nabla^2 \Lambda(\boldsymbol{\theta}^*)^\dagger \nabla \Lambda(\boldsymbol{\theta}^*) \right|$$

$$\leq \left| \mathbf{v}_k(\mathbf{x})^T [\nabla^2 \mathcal{L}(\boldsymbol{\theta}^*)^\dagger - \nabla^2 \Lambda(\boldsymbol{\theta}^*)^\dagger] \nabla \Lambda(\boldsymbol{\theta}^*) \right| + \left| \mathbf{v}_k^T(\mathbf{x}) \nabla^2 \mathcal{L}(\boldsymbol{\theta}^*)^\dagger [\nabla \mathcal{L}(\boldsymbol{\theta}_*) - \nabla \Lambda(\boldsymbol{\theta}^*)] \right|$$

$$\leq ||\mathbf{P}_{\mathbf{1}}^\perp \mathbf{v}_k(\mathbf{x})||_2 ||\nabla^2 \mathcal{L}(\boldsymbol{\theta}^*)^\dagger - \nabla^2 \Lambda(\boldsymbol{\theta}^*)^\dagger||_2 ||\nabla \Lambda(\boldsymbol{\theta}^*)||_2 + ||\mathbf{P}_{\mathbf{1}}^\perp \mathbf{v}_k(\mathbf{x})||_2 ||\nabla^2 \mathcal{L}(\boldsymbol{\theta}^*)^\dagger||_2 ||\nabla \mathcal{L}(\boldsymbol{\theta}_*) - \nabla \Lambda(\boldsymbol{\theta}^*)||_2$$

We then combine these bounds and apply results from Lemma G.1:

$$\sqrt{Lh^d} \sup_{\mathbf{x} \in \Omega} \max_{k \in [n]} \frac{\left| \mathbf{v}_k(\mathbf{x})^T \nabla^2 \mathcal{L}(\boldsymbol{\theta}^*)^\dagger \nabla \mathcal{L}(\boldsymbol{\theta}^*) - \mathbf{v}_k(\mathbf{x})^T \nabla^2 \Lambda(\boldsymbol{\theta}^*)^\dagger \nabla \Lambda(\boldsymbol{\theta}^*) \right|}{\sqrt{\mathbf{v}_k^T(\mathbf{x}) \nabla^2 \Lambda(\boldsymbol{\theta}^*)^\dagger \mathbf{v}_k(\mathbf{x})}}$$

$$\leq \sqrt{Lh^d} \sup_{\mathbf{x} \in \Omega} \sqrt{\frac{2^n p}{n}} \left( ||\nabla^2 \mathcal{L}(\boldsymbol{\theta}^*)^\dagger - \nabla^2 \Lambda(\boldsymbol{\theta}^*)^\dagger||_2 ||\nabla \Lambda(\boldsymbol{\theta}^*)||_2 + ||\nabla^2 \mathcal{L}(\boldsymbol{\theta}^*)^\dagger||_2 ||\nabla \mathcal{L}(\boldsymbol{\theta}_*) - \nabla \Lambda(\boldsymbol{\theta}^*)||_2 \right)$$

$$\leq \sqrt{Lh^d} \sup_{\mathbf{x} \in \Omega} \sqrt{\frac{2^n p}{n}} \left( 2^n p \varphi \frac{n^2 \varphi}{2^n p} + \frac{n}{2^n p} 2^n p \varphi \right) \lesssim \sqrt{2^n pLh^d} \sqrt{n} \varphi$$

Controlling the second term, we control the first expression as

$$\left| \mathbf{v}_k(\mathbf{x})^T \nabla^2 \mathcal{L}(\boldsymbol{\theta}^*)^\dagger \nabla \mathcal{L}(\boldsymbol{\theta}^*) \right| \lesssim ||\mathbf{P}_{\mathbf{1}}^\perp \mathbf{v}_k(\mathbf{x})||_2 ||\nabla^2 \mathcal{L}(\boldsymbol{\theta}^*)^\dagger||_2 ||\nabla \mathcal{L}(\boldsymbol{\theta}^*)||_2$$

Now we control the second, difference expression, using the algebraic bounds $|a^{-1} - b^{-1}| \leq |(ab)^{-1} a - b| \leq |(ab)^{-1}(a^2 - b^2)/(a + b)|$. We will begin by analyzing the "$(a^2 - b^2)/(a + b)$" term,

$$\frac{\left| (\mathbf{v}_k^T(\mathbf{x}) \nabla^2 \mathcal{L}(\boldsymbol{\theta}^*)^\dagger \mathbf{v}_k(\mathbf{x}))^2 - (\mathbf{v}_k^T(\mathbf{x}) \nabla^2 \Lambda(\boldsymbol{\theta}^*)^\dagger \mathbf{v}_k(\mathbf{x}))^2 \right|}{\mathbf{v}_k^T(\mathbf{x}) \nabla^2 \mathcal{L}(\boldsymbol{\theta}^*)^\dagger \mathbf{v}_k(\mathbf{x}) + \mathbf{v}_k^T(\mathbf{x}) \nabla^2 \Lambda(\boldsymbol{\theta}^*)^\dagger \mathbf{v}_k(\mathbf{x})} \leq \frac{|\mathbf{v}_k(\mathbf{x})^T [\nabla^2 \mathcal{L}(\boldsymbol{\theta}^*)^\dagger - \nabla^2 \Lambda(\boldsymbol{\theta}^*)^\dagger] \mathbf{v}_k(\mathbf{x})|}{2\sqrt{n/2^n p}}$$

$$\leq \sqrt{\frac{2^n p}{n}} ||\nabla^2 \mathcal{L}(\boldsymbol{\theta}^*)^\dagger - \nabla^2 \Lambda(\boldsymbol{\theta}^*)^\dagger||_2 ||\mathbf{P}_{\mathbf{1}}^\perp \mathbf{v}_k(\mathbf{x})||_2$$

by results from Lemmas D.1 and G.1. We also control

$$\left( \mathbf{v}_k(\mathbf{x})^T \nabla^2 \mathcal{L}(\boldsymbol{\theta}^*)^\dagger \mathbf{v}_k(\mathbf{x}) \right)^{-1/2} \left( \mathbf{v}_k(\mathbf{x})^T \nabla^2 \Lambda(\boldsymbol{\theta}^*)^\dagger \mathbf{v}_k(\mathbf{x}) \right)^{-1/2} \leq \frac{2^n p}{n ||\mathbf{P}_{\mathbf{1}}^\perp||_2^2}$$

by Lemma D.1. A similar result as Equation D.1 holds in terms of $\nabla^2 \Lambda(\boldsymbol{\theta}^*)$ by an equivalent argument and results in Lemma G.1. Combining these results, and applying Lemmas D.1 and G.1, we construct the upper-bound

$$\sqrt{Lh^d} \sup_{\mathbf{x} \in \Omega} \max_{k \in [n]} \left| \mathbf{v}_k(\mathbf{x})^T \nabla^2 \mathcal{L}(\boldsymbol{\theta}^*)^\dagger \nabla \mathcal{L}(\boldsymbol{\theta}^*) \right| \left| (\mathbf{v}_k^T(\mathbf{x}) \nabla^2 \mathcal{L}(\boldsymbol{\theta}^*)^\dagger \mathbf{v}_k(\mathbf{x}))^{-1/2} - (\mathbf{v}_k^T(\mathbf{x}) \nabla^2 \Lambda(\boldsymbol{\theta}^*)^\dagger \mathbf{v}_k(\mathbf{x}))^{-1/2} \right|$$

$$\leq \sqrt{\frac{2^n p L h^d}{n}} \frac{2^n p}{n} \sup_{\mathbf{x} \in \Omega} ||\nabla^2 \mathcal{L}(\boldsymbol{\theta}^*)^\dagger||_2 ||\nabla \mathcal{L}(\boldsymbol{\theta}^*)||_2 ||\nabla^2 \mathcal{L}(\boldsymbol{\theta}^*)^\dagger - \nabla^2 \Lambda(\boldsymbol{\theta}^*)^\dagger||_2$$

$$\leq \sqrt{\frac{2^n p L h^d}{n}} \left( \frac{2^n p}{n} \right) \frac{n}{2^n p} (2^n p \varphi) \frac{n^2 \varphi}{2^n p} \lesssim \sqrt{2^n p L h^d} n^{3/2} \varphi^2$$

Combining the above results, with our $h$-selection as in the upper-bound on $|T - T_\mathbf{x}|$, we have with probability at least $1 - O(n^{-10})$ for any $\boldsymbol{\theta} \in \Theta$ that

$$|T - T_0| \leq |T - T_\mathbf{x}| + |T_\mathbf{x} - T_0| \lesssim n^3 d \frac{\log(2^n pL)}{(2^n pL^{4/4+d})},$$

completing the argument.

$\square$

## G.5 PROOF OF LEMMA D.3: $\mathbb{P}(\mathbb{P}(|W - W_0| \geq \zeta_1 \mid \mathcal{D}) \geq \zeta_2) \leq \zeta_2$

*Proof.* As in the statement and proof of Lemma G.2, we first define a series of useful, coefficients. We write

$$\widehat{\alpha}_k^{\ell,\mathcal{S}}(\mathbf{x}, \mathcal{D}) = \frac{1}{L} K_\mathbf{H}(\mathbf{X}_\mathcal{S}^\ell - \mathbf{x}) \sum_{i \in \mathcal{S}^+} \frac{\left[ y_\mathcal{S}^{(i,\ell)} - \frac{e^{\widehat{\theta}_i(\mathbf{x})}}{\sum_{j \in \mathcal{S}_+} e^{\widehat{\theta}_j(\mathbf{x})}} \right] \left[ \nabla^2 \mathcal{L}(\widehat{\boldsymbol{\theta}})^\dagger \widehat{\mathbf{v}}_k(\mathbf{x}) \right]_i}{\sqrt{\widehat{\mathbf{v}}_k(\mathbf{x})^T \nabla^2 \mathcal{L}(\widehat{\boldsymbol{\theta}})^\dagger \widehat{\mathbf{v}}_k(\mathbf{x})}}$$

This form conveniently allows us to write

$$W = \sqrt{Lh^d} \sup_{\mathbf{x} \in \Omega} \max_{k \in [n]} \sum_{\mathcal{S} \in \boldsymbol{\mathcal{E}}} \sum_{\ell \in [L]} \widehat{\alpha}_k^{\ell,\mathcal{S}}(\mathbf{x}, \mathcal{D}) Z_\mathcal{S}^\ell .$$

We can construct analogous coefficients $\alpha_{0,k}^{\ell,\mathcal{S}}$ such that

$$W_0 = \sqrt{Lh^d} \sup_{\mathbf{x} \in \Omega} \max_{k \in [n]} \sum_{\mathcal{S} \in \boldsymbol{\mathcal{E}}} \sum_{\ell \in [L]} \alpha_{0,k}^{\ell,\mathcal{S}}(\mathbf{x}, \mathcal{D}) Z_\mathcal{S}^\ell .$$

We let $\widehat{\alpha}_k(\mathbf{x}, \mathcal{D}), \alpha_{0,k}(\mathbf{x}, \mathcal{D})$ represent vectors collecting the scalar coefficients over $(\ell, \mathcal{S}) \in [L] \times \boldsymbol{\mathcal{E}}$. Note that these depend only on the data $\mathcal{D}$, item index $k$, and fixed context $\mathbf{x}$. We similarly collect the Gaussians $\{Z_\mathcal{S}^\ell\}(\ell, \mathcal{S}) \in [L] \times \boldsymbol{\mathcal{E}}$ into vector $\mathbf{Z}$. We then bound $|W - W_0|$

$$W - W_0 \leq \sqrt{Lh^d} \sup_{\mathbf{x} \in \Omega} \max_{k \in [n]} \left| \frac{-\widehat{\mathbf{v}}_k(\mathbf{x})^T \nabla^2 \mathcal{L}(\widehat{\boldsymbol{\theta}})^\dagger}{\widehat{\mathbf{v}}_k(\mathbf{x})^T \nabla^2 \mathcal{L}(\widehat{\boldsymbol{\theta}})^\dagger \widehat{\mathbf{v}}_k(\mathbf{x})} \nabla \widetilde{\mathcal{L}}(\widehat{\boldsymbol{\theta}}) - \frac{-\mathbf{v}_k(\mathbf{x})^T \nabla^2 \Lambda(\boldsymbol{\theta}^*)^\dagger}{\mathbf{v}_k(\mathbf{x})^T \nabla^2 \Lambda(\boldsymbol{\theta}^*)^\dagger \mathbf{v}_k(\mathbf{x})} \nabla \widetilde{\Lambda}(\boldsymbol{\theta}^*) \right|$$

$$\leq \sqrt{Lh^d} \sup_{\mathbf{x} \in \Omega} \max_{k \in [n]} \left| [\widehat{\alpha}_k(\mathbf{x}, \mathcal{D}) - \alpha_k(\mathbf{x}, \mathcal{D})]^T \mathbf{Z} \right|$$

Conditional on $\mathcal{D}$, $\sqrt{Lh^d} [\widehat{\alpha}_k(\mathbf{x}, \mathcal{D}) - \alpha_{0,k}(\mathbf{x}, \mathcal{D})]^T \mathbf{Z}$ is a centered Gaussian process with conditional variance $\sigma^2(\mathcal{D}) = Lh^d ||\widehat{\alpha}_k(\mathbf{x}, \mathcal{D}) - \alpha_{0,k}(\mathbf{x}, \mathcal{D})||_2^2$. We now study this term, after which we will apply maximal-inequality results to derive our final bound.

Define $G_k(\mathbf{x}) = \sqrt{Lh^d} [\widehat{\alpha}_k(\mathbf{x}, \mathcal{D}) - \alpha_{0,k}(\mathbf{x}, \mathcal{D})]^T Z$. Note this is also a function of $\mathcal{D}$, but we suppress the notation for brevity. We've thus shown $|W - W_0| \leq \sup_{\mathbf{x} \in \Omega} \max_{k \in [n]} |G_k(\mathbf{x})|$. It remains for us to control this term, which we will do using classical Gaussian maximal inequality results. Recall that we have controlled the conditional variance of $G_k(\mathbf{x})$ in Lemma G.2:

$$\sigma^2(\mathcal{D}) := \sup_{\mathbf{x} \in \Omega} \max_{k \in [n]} V \left( \sqrt{Lh^d} [\widehat{\alpha}_k(\mathbf{x}, \mathcal{D}) - \alpha_{0,k}(\mathbf{x}, \mathcal{D})]^T Z \mid \mathcal{D} \right) \leq n^3 \varphi^2 .$$

Now we will discretize our space $\Omega$, take a finite Gaussian maximal inequality, and apply a union bound, then using the results above on $\sigma(\mathcal{D})$. To do so, we use and bound the pseudometric

$$
\begin{aligned}
d((x,k),(x\,k)) &:= \left(\mathbb{E}\big([G_k(\mathbf{x}) - G_k(\mathbf{x}')]^2 \mid \mathcal{D}\big)\right)^{1/2} \\
&= \sqrt{Lh^d}\,\|\widehat{\alpha}_k(\mathbf{x},\mathcal{D}) - \alpha_{0,k}(\mathbf{x},\mathcal{D}) - (\widehat{\alpha}_k(\mathbf{x}',\mathcal{D}) - \alpha_{0,k}(\mathbf{x}',\mathcal{D}))\|_2 \\
&\lesssim \sigma(\mathcal{D})\frac{\|x - x'\|_2}{h}
\end{aligned}
$$

by the form of the $\alpha$ coefficients and Lipschitzness of the kernel assumed in Section H.1.

Let $C_{\log} := c_0 \log(2^n pL)$ for large enough $c_0 > 0$. Construct $G \subset \Omega$ such that $|G| \le Ch^{-d}C_{\log}^d$, and let $\pi(\mathbf{x})$ be the projection of $\mathbf{x}$ to the nearest $g \in G$. Then

$$
\sup_{\mathbf{x} \in \Omega}\max_{k \in [n]}|G_k(\mathbf{x})| \le \max_{g \in G, k \in [n]}|G_k(g)| + \sup_{\mathbf{x} \in \Omega}\max_{k \in [n]}|G_k(\mathbf{x}) - G_k(\pi(\mathbf{x}))| \lesssim \max_{g \in G, k \in [n]}|G_k(g)| + \frac{\sigma(\mathcal{D})}{C_{\log}},
$$

and now we need only control the finite maximum $\max_{g \in G, k \in [n]}|G_k(g)|$. Conditional on $\mathcal{D}$, each $G_k(g)$ is a centered Gaussian with variance bound $\sigma^2(\mathcal{D})$, such that

$$
\mathbb{P}\left(\max_{g \in G, k \in [n]}|G_k(g)| \ge t \mid \mathcal{D}\right) \le 2n|G|\exp\left(\frac{t^2}{2\sigma^2(\mathcal{D})}\right).
$$

Then fix $t = c_1\sigma(\mathcal{D})\sqrt{\log(n|G|) + 10\log n}$, and note that $t \lesssim \sigma(\mathcal{D})\sqrt{\log(nh^{-d})} \lesssim n^{5/2}\varphi\sqrt{\log(2^n pL)}$. From this, with $\sigma(\mathcal{D}) \lesssim n^{3/2}\varphi$ above, we then have the bound

$$
\mathbb{P}\left(|W - W_0| \gtrsim dn^3\frac{\log(2^n pL)}{(2^n pL)^{2/4+d}} \mid \mathcal{D}\right) \lesssim n^{-10}
$$

Lastly, define $\Gamma$ to be the event that Lemma D.1, G.1 and G.2 hold. By our union bound arguments, these hold jointly with probability at least $1 - O(n^{-10})$. These events imply the conditional-tail bound above. We then have

$$
\mathbb{P}\left(\mathbb{P}\left(|W - W_0| \ge dn^3\frac{\log(2^n pL)}{(2^n pL)^{2/4+d}}\right) \gtrsim n^{-10}\right) \le \mathbb{P}(\Gamma^c) \le n^{-10}
$$

completing the argument.

$\square$

### G.6 PROOF OF D.4: MOMENT RESULT

*Proof.* Recall the object of the lemma statement,

$$
T_{0,k}(\mathbf{x}) := \left(\frac{-\mathbf{v}_k(\mathbf{x})^T\nabla^2\Lambda(\boldsymbol{\theta}^*)^{\dagger}}{\sqrt{\mathbf{v}_k(\mathbf{x})^T\nabla^2\Lambda(\boldsymbol{\theta}^*)\mathbf{v}_k(\mathbf{x})}}\nabla\Lambda(\boldsymbol{\theta}^*)\right)
$$

such that as previously defined, $T_0 = \sqrt{Lh^d}\sup_{x \in \mathbf{X}}\max_{k \in [n]}T_{0,k}(\mathbf{x})$. We first demonstrate that $T_{0,k}(\mathbf{x})$ is centered for any $\mathbf{x} \in \Omega, k \in [n]$. Given $\boldsymbol{\mathcal{E}} \perp \mathbf{X}$, we have

$$
\mathbb{E}[T_{0,k}(\mathbf{x}) \mid \mathbf{X}] = \left(\frac{-\mathbf{v}_k(\mathbf{x})^T\nabla^2\Lambda(\boldsymbol{\theta}^*)^{\dagger}}{\sqrt{\mathbf{v}_k(\mathbf{x})^T\nabla^2\Lambda(\boldsymbol{\theta}^*)\mathbf{v}_k(\mathbf{x})}}\right)\mathbb{E}\left[\nabla\Lambda(\boldsymbol{\theta}^*) \mid \mathbf{X}\right].
$$

Now evaluating $\mathbb{E}\left[\nabla\Lambda(\boldsymbol{\theta}^*) \mid \mathbf{X}\right]$,

$$
\begin{aligned}
\mathbb{E}\left[\nabla\Lambda(\boldsymbol{\theta}^*) \mid \mathbf{X}\right] &= \mathbb{E}\left[\sum_{\mathcal{S} \in \boldsymbol{\mathcal{E}}}\sum_{\ell \in [L]}\frac{1}{L}K_{\mathbf{H}}(\mathbf{X}_{\mathcal{S}}^{\ell} - \mathbf{x})\sum_{i \in \mathcal{S}_+}\left(y_{\mathcal{S}}^{(i,\ell)} - \frac{e^{\theta_i(\mathbf{X}_{\mathcal{S}}^{\ell})}}{\sum_{j \in \mathcal{S}_+}e^{\theta_j(\mathbf{X}_{\mathcal{S}}^{\ell})}}\right) \mid \mathbf{X}\right] \\
&= \sum_{\mathcal{S} \in \boldsymbol{\mathcal{E}}}\sum_{\ell \in [L]}\frac{1}{L}K_{\mathbf{H}}(\mathbf{X}_{\mathcal{S}}^{\ell} - \mathbf{x})\sum_{i \in \mathcal{S}_+}\left(\mathbb{E}\left[y_{\mathcal{S}}^{(i,\ell)} \mid \mathbf{X}\right] - \frac{e^{\theta_i(\mathbf{X}_{\mathcal{S}}^{\ell})}}{\sum_{j \in \mathcal{S}_+}e^{\theta_j(\mathbf{X}_{\mathcal{S}}^{\ell})}}\right) = 0.
\end{aligned}
$$

Thus we need only $\mathbb{E}[T_{0,k}(\mathbf{x})^2]$. Note that $\mathbf{v}_k(\mathbf{x}), \nabla^2 \Lambda(\boldsymbol{\theta})$ do not depend on the observed decisions, $\mathbf{y}$. We immediately then have

$$\mathbb{E}\left[T_{0,k}^2(\mathbf{x}) \mid \mathbf{X}\right] = \frac{-\mathbf{v}_k(\mathbf{x})^T \nabla^2 \Lambda(\boldsymbol{\theta}^*)^\dagger \mathbb{E}\left[\nabla \Lambda(\boldsymbol{\theta}^*) \nabla \Lambda(\boldsymbol{\theta}^*)^T \mid \mathbf{X}\right] \nabla^2 \Lambda(\boldsymbol{\theta}^*)^\dagger \mathbf{v}_k(\mathbf{x})}{\mathbf{v}_k(\mathbf{x})^T \nabla^2 \Lambda(\boldsymbol{\theta}^*) \mathbf{v}_k(\mathbf{x})} = 1 \ .$$

$\square$

## H   NOTES ON KERNEL ESTIMATION

### H.1   KERNEL FUNCTION ASSUMPTIONS

For Kernel functions $K : \mathbb{R}^d \mapsto \mathbb{R}_{\geq 0}$, we use the shorthand notation $K_{\mathbf{H}}(\mathbf{x}) = |\mathbf{H}|^{-1/2} K\left(\mathbf{H}^{-1/2}\mathbf{x}\right)$. Here we assume $\mathbf{H}$ is a diagonal matrix of positive values bounded above by some value $h = o(1)$, using the convention that $(\mathbf{H})_{ij} = h_{ij}^2$. We assume multiplicative, multivariate kernels. That is, $K(\mathbf{x}) = \prod_{i \in [d]} K'(x_i)$. We assume $K$ is $B$-Lipschitz, and satisfies the integrability conditions below for $\alpha \in [4]$:

$$\int K(x)dx = 1 \quad \int xK(x)dx = 0 \quad \int K^{\alpha}(x)dx \leq C_{\alpha} < \infty \quad \int x^{\alpha}K(x)\,\mathrm{d} \leq c_{\alpha} < \infty$$

Lastly, we assume $K$ is bounded in total variation, such that $\int |\nabla K(x)|\,\mathrm{d}x \leq C_{TV} < \infty$. The above conditions are standard in kernel-based, nonparametric regression methodologies. Standard kernels such as the Epanechnikov, uniform, quadratic (and higher-order polynomial), among others satisfy the above conditions.

Below we outline useful properties of this kernel function, its moments, and kernel-based density and regression estimation rates. When brief, we include proofs for comprehensiveness. For cumbersome (but standard) proofs, we cite existing results. The density and regression estimation results in Lemmas H.2 to H.5 have been well-characterized in the nonparametric literature. A nice, holistic discussion of results similar to those presented below is presented in Hansen (2008). Earlier works include foundational analyses of these kernel-based density and egression estimators, including uniform rates of convergence analogous to those of this section (Ruppert and Wand, 1994; Häardie and Müller, 2000; Giné and Nickl, 2008).

### H.2   PROOF OF KERNEL DENSITY ESTIMATION RATES

Throughout this section, we assume $\mathbf{x}$ is some fixed value, and $\{\mathbf{X}_i\}_{i \in [m]}$ are independent (but not necessarily $iid$) observations with bounded density $f_{\mathbf{x}}$ satisfying the assumptions of Section 2.4.

**Lemma H.1** (Kernel Moment Bounds). Consider a diagonal bandwidth matrix with each element bounded above by some maximal value $h = o(1)$, that is $(\mathbf{H})_{ij} = \mathbb{1}(i = j)h_{ij}^2$ for $\max_{ij} h_{ij} < h$. Then

$$\mathbb{E}\left[K_{\mathbf{H}}\left(\mathbf{X}_i - \mathbf{x}\right)\right] = f_{\mathbf{x}}(\mathbf{x}) + O(dh^2) \tag{H.1}$$

$$\mathbb{E}\left[K_{\mathbf{H}}\left(\mathbf{X}_i - \mathbf{x}\right) - \mathbb{E}\left[K_{\mathbf{H}}\left(\mathbf{X}_i - \mathbf{x}\right)\right]\right]^2 = O(h^{-d}) \tag{H.2}$$

$$\mathbb{E}\left[K_{\mathbf{H}}\left(\mathbf{X}_{\mathcal{S}}^{(i,\ell)} - \mathbf{x}\right) - f_{\mathbf{x}}(\mathbf{x})\right]^2 = O(h^{-d}) \tag{H.3}$$

*Proof.* The proof of (H.1) is a standard result in density estimation.

$$\mathbb{E}K_{\mathbf{H}}(\mathbf{X} - \mathbf{x}) = \int_{\mathcal{X}} |\mathbf{H}|^{-1/2} K(\mathbf{H}^{-1/2}(\mathbf{u} - \mathbf{x})) f(\mathbf{u})\,\mathrm{d}\mathbf{u}$$

$$= \int_{\mathcal{X}} K(\mathbf{s}) f(\mathbf{x} + \mathbf{H}^{1/2}\mathbf{s})\,\mathrm{d}\mathbf{s}$$

$$= \int_{\mathcal{X}} K(\mathbf{s})\left(f(\mathbf{x}) + \mathbf{s}^T\mathbf{H}^{1/2}\nabla f(\mathbf{x}) + \frac{1}{2}\mathbf{s}^T\mathbf{H}^{1/2}\nabla^2 f(\mathbf{x})\mathbf{H}^{1/2}\mathbf{s} + o\left(\mathbf{s}^T\mathbf{H}\mathbf{s}\right)\right)\,\mathrm{d}\mathbf{s}$$

$$= f(\mathbf{x}) + \frac{1}{2}\operatorname{tr}\left(\int_{\mathcal{X}} \nabla^2 f(\mathbf{x})\mathbf{H}\mathbf{s}\mathbf{s}^T K(\mathbf{s})\,\mathrm{d}\mathbf{s}\right) + \int_{\mathcal{X}} K(\mathbf{s})o\left(\mathbf{s}^T\mathbf{H}\mathbf{s}\right)\,\mathrm{d}\mathbf{s}$$

$$= f(\mathbf{x}) + \frac{1}{2}\operatorname{tr}\left(\mathbf{H}\nabla^2 f(x)\right) + o\left(\operatorname{tr}(\mathbf{H})\right)$$

$$= f(\mathbf{x}) + O(dh^2)$$

Equation (H.2) follows by a similar Taylor expansion and analysis on the second moment $\mathbb{E}[K_{\mathbf{H}}(\mathbf{X} - \mathbf{x})^2]$. Lastly, equation (H.3) follows by square-expansion with the rates from equations (H.1) and (H.2).

$\square$

**Lemma H.2** (Deviation of Kernel Function About its Mean). Define $b_j(\mathbf{x}) = K_{\mathbf{H}}(\mathbf{X}_j - \mathbf{x}) - \mathbb{E}[K_{\mathbf{H}}(\mathbf{X} - \mathbf{x})]$.

$$\mathbb{E}\left[\sup_{\mathbf{x} \in \Omega}\left|\frac{1}{m}\sum_{j=1}^{m} b_j(\mathbf{x})\right|\right] \leq \sqrt{\frac{d}{mh^d}\log\frac{1}{h^d}}\ . \tag{H.4}$$

Similarly,

$$\sup_{\mathbf{x} \in \Omega}\left|\frac{1}{m}\sum_{j=1}^{m} b_j(\mathbf{x})\right| - \mathbb{E}\sup_{\mathbf{x} \in \Omega}\left|\frac{1}{m}\sum_{j=1}^{m} b_j(\mathbf{x})\right| \lesssim \sqrt{\frac{d}{mh^d}\log\frac{m}{h^d}} \tag{H.5}$$

holds with probability at least $1 - O(m^{-10})$.

**Lemma H.3** (Rate of Kernel Density Estimation). For the kernel density estimator $\frac{1}{m}\sum_{i=1}^{m} K_{\mathbf{H}}(\mathbf{X}_i - \mathbf{x})$,

$$\sup_{\mathbf{x} \in \Omega}\left\{\frac{1}{m}\sum_{i=1}^{m} K_{\mathbf{H}}(\mathbf{X}_i - \mathbf{x}) - f(\mathbf{x})\right\} \lesssim cdh^2 + c\sqrt{\frac{d}{mh^d}\log\frac{m}{h^d}}$$

with probability at least $1 - O(m^{-c}), c > 0$.

### H.3 KERNEL REGRESSION RESULTS

**Lemma H.4** (Kernel Regression Moment). Let $g : \mathbb{R}^d \mapsto \mathbb{R}$ be a smooth function, and $K$ a kernel function inheriting the assumptions and notation from Section H.1.

$$\sup_{\mathbf{x} \in \Omega}\left|\mathbb{E}_{\mathbf{u}}[K_{\mathbf{H}}(\mathbf{x} - \mathbf{u})g(\mathbf{u})] - f_{\mathbf{X}}(\mathbf{x})g(\mathbf{x})\right| \lesssim cdh^2$$

**Lemma H.5** (Kernel Regression Maximal Inequalities). Let $\widehat{g}(\mathbf{x}) = m^{-1}\sum_{i=1}^{m} K_{\mathbf{H}}(\mathbf{X}_i - \mathbf{x})g(\mathbf{X}_i)$. Under the same conditions as Lemma H.4,

$$\mathbb{E}\sup_{\mathbf{x} \in \Omega}\left|\widehat{g}(\mathbf{x}) - \mathbb{E}[\widehat{g}(\mathbf{x})]\right| \lesssim \sqrt{\frac{\log(h^{-d})}{mh^d}}\ . \tag{H.6}$$

Also, with probability at least $1 - O(m^{-C})$,

$$\sup_{\mathbf{x} \in \Omega}\left|\widehat{g}(\mathbf{x}) - \mathbb{E}[\widehat{g}(\mathbf{x})]\right| - \mathbb{E}\sup_{\mathbf{x} \in \Omega}\left|\widehat{g}(\mathbf{x}) - \mathbb{E}[\widehat{g}(\mathbf{x})]\right| \lesssim C\sqrt{\frac{\log(mh^{-d})}{mh^d}}\ . \tag{H.7}$$

Combining these two results gives

$$\sup_{\mathbf{x} \in \Omega}\left|\widehat{f}(\mathbf{x}) - \mathbb{E}[\widehat{f}(\mathbf{x})]\right| \lesssim C_2\sqrt{\frac{\log(mh^{-d})}{mh^d}}$$

with probability at least $1 - O(m^{-C})$, for $C_2 > C > 0$.

