# OpenReview forum: "Nonparametric, Contextual Preference Estimation and  Assortment Optimization"
_ICLR.cc/2026/Conference — Submitted to ICLR 2026_

### Official Review · Reviewer_p5b6 · 2025-10-30

**Soundness:** 3
**Presentation:** 2
**Contribution:** 2
**Rating:** 2
**Confidence:** 3

**Summary:**

In this work, the authors propose a framework for contextual assortment optimization and inference. Under a multinomial logit model that can vary conditional on context x, the authors provide tools for statistical inference of the "optimal assortment" of a set of items at x, which can be viewed as the smallest assortment among those that maximize the total expected utilities. The utility model extends to consider both context-specific user preferences and context-specific utilities. This framework can be viewed as extending that of Shen et al. (2023) to the contextual setting. The authors provide rigorous theoretical treatment of their results with preliminary empirical validation.

**Strengths:**

The authors clearly state the proposed technical problem and provide a formal treatment. The related work section clearly identifies the gap in prior work that is addressed by this approach -- i.e., Shen et al. (2023). Both the model and the target estimands (i.e., contextual marginal utility gap) are natural in a discrete choice setting, and the dependence of both the utility functions and preference function on x is beneficial for flexibility of the approach. Clear statement of theoretical assumptions and their limitations (e.g., that the multinomial logit choice model implicitly makes the independence of irrelevant alternatives assumption) is also a clear strength of the work.

**Weaknesses:**

## Limited empirical validation

In its current form, the work has very limited empirical validation, consisting of a single simulation study in the appendix. Additionally, there are limited details of the setup, performance metrics, and results for this experiment. This makes the work incomplete in its current form.

This work is off to an excellent start; as the authors continue to refine it into its final form, I encourage the authors to substantially extend the scope of the empirical validation by adding further numeric experiments, as well as experiments with semi-synthetic or real-world data. This will provide important evidence for the empirical validity of the work. In experiments, I encourage the authors to illustrate how context-dependent estimation improves the empirical results. Ablations with context-dependent preference functions and utility functions would both be interesting, as would experiments that vary the assortment set size.


## Limited motivation

In its current form, there is also a strong disconnect between the motivation for the paper and its technical content. While the technical content of the work is sound, the authors repeatedly reference "decision agents" in the abstract, but drop this discussion in the introduction and the remainder of the main text.

Throughout the abstract and introduction, a key thing I was missing is *why is this optimal assortment problem is important?* Further, I was looking for a more concrete motivation for why a decision-support tool developer might want to select a model on the basis of discrete choice preferences. Typically tool adoption decisions are made on the basis of some external outcome (e.g., utility/accuracy) based measure.

While the proposed framework accounts for "tool-specific cost or utility information”, typically in clinical settings preference information is used to characterize some some component of the decision subject preferences or decision outcomes. E.g., the canonical decision curve analysis characterizes tradeoff between the cost of false positive and false negatives. Similarly, while the authors claim that users' perceived value can differ by prompt, this seems to motivate that the choice of decision-making tool would vary depending on the prompt, which is not practically feasible.

Overall, I am certain that this technical problem *is* important in some setting, and I invite the authors to either (1) provide a stronger support for why application to decision-support tool selection is a salient application area, or (2) switch to a different application area. I encourage the authors to pick a motivating application that is aligned with the domain of the semi-synthetic / real-world experiments.

## Dense technical presentation

While the current main text exposition provides good detail, it is prohibitively dense in its current form. I encourage the authors to provide more exposition surrounding key results, especially theorems 4.1 and 4.2. Further, while statement of assumptions in 2.1 is important, shifting these to elsewhere in the text would strengthen the flow leading into the problem formulation. Spending more time in Sections 2.1 and 2.2 describing the theoretical setup and target quantities would also help the reader understand the relevance and importance of this problem (see remarks above). Finally, in its current form, Section 4.3 reads as cursory and "tacked on". Given the importance of the hypothesis testing procedure, I invite the authors to spend more time outlining the estimation procedure, perhaps with pseudocode.

**Questions:**

- Can you elaborate on how this work differs from Wang et al. (2025b), given the similarities with the estimators reported in that work? Further, can you elaborate on the differences between this technical setup of the multinomial logit model and the standard BTL model?
- Could you clarify the intended application domain? If decision-support tools remain the target, how do you envision practitioners selecting tools based on choice preferences rather than outcome measures?
- Can you provide more details on the simulation study in the appendix (setup, metrics, baselines)? What performance improvements does the contextual approach provide over non-contextual baselines?

---

> ### Author Response · Authors · 2025-11-29
> **Reply to Reviewer p5b6**
>
> We thank the reviewer for their thoughtful feedback, which we believe has highlighted important items that can improve our manuscript. Below, we highlight edits to our submission that attempt to address their concerns. We attempt to further articulate our reasoning behind such edits in the comments below. We paraphrase the reviewer's comments below to specify which review our responses address.
>
> ### **Limited motivation**
>
> We attempt to further clarify the importance of our optimal assortment motivation in the context of decision-support tools in the Introduction. We specifically focus on two points in the reviewer’s comments regarding motivation: the importance of the assortment problem, beyond evaluation of tools for baseline efficacy and/or accuracy, and more concrete examples of our motivation.
>
> Our analysis does assume a baseline efficacy for any tool, such that they are safe, accurate, appropriate for clinical practice etc. Our motivation lies in the optimal adoption/identification of such tools. Having shown efficacy, adoption still requires proper cost-benefit analysis, motivating the optimal assortment problem. Competing products do not necessarily have comparable metrics, which in turn may differ by clinical context (https://jamanetwork.com/journals/jama/fullarticle/2840175). The discrete choice preferences take provider preferences into account within their decision-making process. We have added writing in our Introduction to make this point (including the referenced article above).
>
> Lastly, we do discuss the utility parameter abstractly to allow for general measures of clinical utility (e.g. measure(s) of prediction performance, measures of efficiency gains, etc.). We have further edited the Introduction to include writing, based on some of the discussion below towards this reviewer’s comment.
>
> Towards specific applications, we include text from a response to comment from Reviewer kZoM: We attempt to weave our motivating example throughout. In Section 2.3, we previously presented two examples of the construction of tests. We have added writing that connects these constructions to the motivation of the Introduction in further detail.
>
> ### **Dense technical presentation**
>
> **(Review): I encourage the authors to provide more exposition surrounding key results, especially theorems 4.1 and 4.2.**
>
> We agree with the reviewer and have added additional writing for all theorems, connecting results to the study's goals: with edits in Section 4.3 (discussed further, two comments below), which now contains walkthrough of implementation via pseudocode in Algorithms 2, 3.
>
> **(Review): ...Spending more time in Sections 2.1 and 2.2 describing the theoretical setup and target quantities would also help...**
>
> Mirroring response to a similar comment from Reviewer K2VV, we have attempted to more lucidly present our theoretical setup and target estimands, both with respect to our mathematical setup and motivating example. We have added additional writing in our Introduction to better introduce the problem statement in plain language and connect these to the objects in Section 2. We have reorganized Sections 1 and 2, relegating technical assumptions to the end of Section 2. In both the Introduction and Section 2, we attempt to more explicitly highlight the estimands of interest (with these equations remaining in Section 2 and plain language discussion in the Introduction, with explicit references connecting the two sections).
>
> **(Review): ... in its current form, Section 4.3 reads as cursory and "tacked on". Given the importance of the hypothesis testing procedure, I invite the authors to spend more time outlining the estimation procedure, perhaps with pseudocode.**
>
> We agree with the reviewer's suggestion, and have replaced much of our proce with pseudocode for our procedure in two algorithms for the construction of confidence bands and conducting resulting tests. We have also edited the prose of this section to connect our testing procedure with the motivating problem from Section 2.3. We hope that these revisions both clarify the testing procedure  and help to maintain a consistent thread of our motivation throughout.
>
>
> ### **Limited Empirical Validation & Q3**
>
> We have clarified the simulation setup in Appendix Section A by more clearly outlining the data-generating process, including new comments on the generation of observed covariate data. This section also contains writing that defines the metrics used to assess performance. Changes are included in blue text in Appendix Section A.
>
> ### **Questions:**
>
> **Q1:** We direct the reader to our **W1b** in our response to Reviewer kZoM, who raised a similar point. We again thank both reviewers for their comment, as these points were previously underdiscussed in our submission.
>
> **Q2**: We attempt to address this question within the **Limited motivation** comments and edits above.
>
> **Q3**: We address in the **Limited Empirical Validation & Q3** section above.

---

### Official Review · Reviewer_sJrV · 2025-11-01

**Soundness:** 3
**Presentation:** 3
**Contribution:** 3
**Rating:** 6
**Confidence:** 3

**Summary:**

The paper studies contextual assortment optimization under a contextual MNL model. It proposes a kernel-weighted local likelihood estimator for context-varying log-preferences, introduces a debiasing step, and develops a Gaussian multiplier bootstrap to build simultaneous confidence bands for the index of the optimal assortment. Theoretical results include uniform convergence rates for the estimator, an error expansion for the debiased estimator, and validity of simultaneous confidence bands and tests.

**Strengths:**

This study addresses one of the central problems in deploying AI in the real world. The authors propose a cutting-edge approach grounded in classical methods whose theoretical performance is well understood by statisticians. The problem setting is well motivated, and the theoretical results appear sound. The work is somewhat preliminary in a few respects—for example, (i) the inferential target
$S^*(x)$ relies on known utilities $r(x)$, and (ii) the contextual MNL inherits IIA—but these issues are secondary to the main contribution. As a pioneering effort in this area, I do not view these weaknesses as major concerns.

**Weaknesses:**

See above,

**Questions:**

See above,

---

> ### Author Response · Authors · 2025-11-29
> **Reply to Reviewer sJrV**
>
> We thank the reviewer for their review and feedback of our submission!
>
> We agree with the reviewer that the IIA assumption is possibly a limiting assumption, but we also note that this is a common assumption in the optimal assortment literature (both classic and recent), with appealing properties. Specifically, the form of the MNL model permits simple identification of the optimal assortment, having estimated the items’ preference parameters (with known/deterministic utility functions).
>
> Similarly, we agree with the reviewer’s notes that the assumption of known/deterministic utility functions is a limitation of these analyses. However we believe this to be a reasonable assumption, which also introduces flexibility in the practical application of this work. We have added small writing in the Discussion to transparently reiterate these points.
>
> We again are thankful for the reviewer’s time and feedback in reading our work.

---

### Official Review · Reviewer_kZoM · 2025-11-01

**Soundness:** 3
**Presentation:** 2
**Contribution:** 3
**Rating:** 4
**Confidence:** 2

**Summary:**

This paper studies contextual evaluation for preference alignment and assortment optimization within a multinomial logit (MNL) discrete choice model. The problem setting considers a set of items with utility values and context-dependent preference values, and assortment optimization aims to identify the subset of items maximizing the expected utility. Inference tasks can be defined for the optimal assortment to test its properties. Using data on observed selections from assortments under specific contexts, the paper proposes methods for estimation and inference. The estimation task estimates contextual preference parameters with uniform convergence guarantees. The inference task tests whether an item is included in the contextually optimal assortment, and theoretical results support the validity of the proposed testing procedure.

**Strengths:**

The paper recognizes the importance of contextual influence in assortment optimization. Under a realistic contextual MNL model, the paper develops theoretically sound procedures for estimating preference parameters and making inference on optimal assortments. The inference method is especially notable, as it supports in-depth analysis of the optimal assortment, which may be useful for guiding practical selection of decision-support agents.

**Weaknesses:**

1. Contributions should be clarified: In Section 1.2, the paper describes its contributions as (1) “providing uniform bounds on preference parameter and utility gap estimation” and (2) “first inferential procedures proposed in the contextual multinomial logit choice model literature”. I think both aspects require additional discussion to be fully convincing.

On the first aspect related to estimation, what are some known estimation methods in contextual MNL choice model? And what type of performance guarantees are available? To obtain the uniform bound result in this paper, what is novel or different about this paper’s proposed estimation?

On the second aspect related to inference, the paper mentions Wang et al. (2025b) as providing motivation, is the presented inference procedure a direct application of Wang et al., or is any non-trivial extension required? While there is no inferential method for contextual MNL model, is there any inferential method for MNL model without contextual consideration? If so, are there any meaningful comparisons with the paper’s proposed inferential procedure?


2. Lack illustrative examples for practical connection: The paper begins with an interesting practical challenge of identifying if new tools should be included in assortment of decision aids for clinicians, but the rest of the paper does not connect back to this motivating task. While the paper is largely methodological, I think illustrative examples are needed to demonstrate the proposed method in practical setting. I see two possible ways to go about this: (1) a case study with synthetic or real-world data generated with the process described in Section 2, then apply the proposed methods on this data to reveal insights about preference and assortment (2) without using actual data, describe a few application scenarios where observable data is available for estimation and discuss examples of relevant hypotheses to test in inference.

**Questions:**

Please refer to my questions listed in the weaknesses. In addition, I have some clarification questions.

1. How realistic is it to consider the utility values (r) are known on all items? I would expect these values to often be similarly unknown as the preference parameters.

2. How to think about the observed decisions (y) in practice? Are these decisions representing a clinician choosing to use a tool from a set of tools? In that case, how about cases where a clinician uses multiple tools together from a set of tools?

---

> ### Author Response · Authors · 2025-11-29
> **Reply to Reviewer kZoM**
>
> We thank the reviewer for their review of our work. These comments have raised important points in the motivation and presentation of our work. We have edited submission where appropriate based upon this thoughtful feedback. We hope that these edits have improved the clarify of our work and more clearly motivate our study of this contextual assortment problem.
>
> We address Weakness (W#) and Questions (Q#) raised by the reviewer below. Where appropriate, we have separated some comments into sub-sections, to address specific concerns or suggestions from the reviewer.
>
> **W1**: We have included additional writing on the Contributions section explicitly referencing literature from the previous section (and additional references) for comparisons to existing works in the contextual setting. While we do not discuss results in the online learning in-depth, we do cite works to outline the distinction between our offline work and these regret bound analyses, which we believe our work complements well. We also include a larger focus on the inferential contributions, which we do believe to be the focus of our work. This mirrors edits to our manuscript that attempt to more heavily discuss our motivating application and link this to the optimal assortment problem studied, predominantly in Section 4.
>
> **W1a:** We attempt to address the reviewer’s concerns for estimation methods in contextual MNL models in addressing their previous comment. The overall proof strategy mirrors previous works but with important novelties (e.g. new objects from local-approximation of the likelihood, technical results for the marginal utility gap, contextual assortment testing). The inferential proofs apply a technical result from Wang et al. 2025b, but once again for a distinct setup and thus study distinct objects. New technical results are necessary in order to prove valid inference  This is discussed further in the comment below.
>
>
> **W1b:** There are importantly two considerations in relating our work to Wang et al. (2025b). We have included a summary of these points in Sections 1.1, 1.2.
>
> Firstly, our primary focus of the optimal assortment, is a distinct motivation and estimand from Wang et al. (2025b). We introduce utility parameters that allow us to study two additional functionals of interest: the assortment-level expected utility and the optimal assortment estimand.
>
> Secondly, we use an extension of Gaussian multiplier bootstrap theory from Wang et al. (2025b). Applying this result requires a series of non-trivial technical lemmas in order to avail the desired confidence band construction. Importantly, we also apply this technical result to new object (the marginal utility gap)
>
> In short, we feel this work stands apart from Wang et al. (2025b) on two fronts. Firstly we introduce a distinct motivating problem and resulting estimands, both of which derive from a different data-generating model. Secondly, while we avail ourselves of technical results from Wang et al. (2025b), we do so on new objects of interest (related to our optimal assortment problem), which does require non-trivial technical work to demonstrate validity in applying this result.
>
> **W2:**
> In response to this comment and others, we attempt to weave our motivating example throughout our submission. This includes more thorough discussion in the modeling setup (Section 2) and relating our theoretical results to this setup more explicitly in Sections 3 and 4.
>
> Specifically, in Section 2.3, we previously presented two examples of the construction of tests. We have added writing that connects these constructions to the motivation of the Introduction in further detail. In Section 4.3, where we outline the testing procedure, we add writing that references these updates to Section 2.3, once more connecting the methodology to our motivating examples of our Introduction.
>
> **Q1:** We acknowledge that the assumption of known utility functions is a possible limitation but believe this to be reasonable. Classically, these utility parameters correspond to cost (in studies of consumer purchasing behavior). This similarly extends to our motivating examples, with the assumption corresponding to the costs of tools at the institutional (or at least, analyst) level being known. Practically, these may not differ contextually, but our method allows this flexibility if these functions of $\mathbf{x}$ are known.
>
> **Q2**: Within our motivating example, one can consider these decisions being generated as the reviewer has stated: a clinician choosing their favorite tool from a set. The identification of the optimal assortment is not limited to any specific tool but may identify a subset of any size among the multiple, offered tools. A benefit of this form of data collection (“top-1 ranking”) is in minimizing the burden of the clinical (or general decision-agent) in responding with their selection, compared to requiring fuller rankings of items or other, more granular information.

---

### Official Review · Reviewer_K2VV · 2025-11-04

**Soundness:** 2
**Presentation:** 1
**Contribution:** 1
**Rating:** 2
**Confidence:** 4

**Summary:**

Given a set of decision support tools, this paper aims to determine a subset of these tools that should be included in the optimal subset of decision aids given contextual information and varying objectives.

**Strengths:**

The problem of identifying which decision support tools are appropriate for a given context / utility function is an important problem.

**Weaknesses:**

- The clarity of the paper is lacking in terms of the problem statement and mathematical setup, making it difficult for a reader to understand and appreciate the main results. I believe the paper would clarity improvements in Section 1.4, Section 2 prior to publication.
- For example, the paper could be improved by more clearly stating the problem that it aims to solve. (I needed multiple passes through the introduction/abstract and the mathematical setup). For example, in the introduction Line 50, the authors write that "The assortment optimization problem posits $n$
items, each with an associated preference and utility value, and attempts to identify the subset of
items that maximizes an expected total return utility." It is not clear to me (1) how preferences affect this optimization problem (it appears that the choice of items to select would simply depend on utility?) and (2) are there any limitations on the number of items that we can select (e.g. how large is the subset). These are relatively simple aspects of the setting that would be helpful to spell out precisely -- as the audience may not be familiar with assortment optimization.
- The statement of the mathematical setup could be made more clear by introducing the important mathematical notation, e.g. the decsion model, the set of items, an assortment $\mathcal{S}$, the set of assortment $\mathbf{\mathcal{S}}$ prior to stating the mathematical assumptions because it is difficult to interpret the assumptions without understanding how they fit into the decision model.

**Questions:**

N/A

---

> ### Author Response · Authors · 2025-11-29
> **Reply to Reviewer K2VV**
>
> We thank the reviewer for their time and suggestions towards motivating our problem of interest as well as improving the organization and presentation of this work. Below we address the reviewer’s three points:
>
> **W1:** We hope that our comments below address more detailed comments regarding presentation of material in Section 1.4 (our W3 comment below) and Section 2 (our W2 comment below). We have added clarifying language in the first section of our Introduction, in particular new third and fourth paragraphs that introduce the main problem statement in plain language.
>
> **W2:** As noted above, we have added the additional writing in Section 1 with a goal of addressing the reviewer’s concerns regarding clarity of the problem statement in plain language. We have also added writing that clarifies the role of both the preference and utility parameters in identifying the optimal assortment (as defined both within this paragraph and formally in Section 2). Similarly, within this new paragraph we articulate explicitly the estimands of interest (with these equations remaining in Section 2 but with explicit references, to direct the interested reader to the mathematical expressions without overloading the Introduction with yet unnecessary notation). In the definition of the optimal assortment, we do define that this assortment may contain any subset of the included $n$ items. In our new paragraph in the Introduction, we also define this feature within the problem statement (again in plain language, with a reference to the formal mathematical definition).
>
> **W3:** We agree with the reviewer’s suggestion, and have re-organized the Introductory material to delay discussion of modelling assumptions until after description of the decision model (Section 2). These are now located at the end of Section 2, removed from the setup and motivation of the work. All relevant objects are now defined and discussed prior to the description of more technical assumptions necessary for our theoretical work.

---

### Author Response · Authors · 2025-11-29
**Reply to Reviewers**

We thank all of our reviewers for their time and thoughtful review of our work! We have attempted to improve our work based on their feedback. Edits to our submission based on the reviewers' insights are marked in blue (with exceptions for Algorithms 2, 3, which are new but included in black text in Section 4.3).

---

### Meta-Review · Area_Chair_uXHC · 2026-01-11

**Summary:**

The paper introduces a framework for contextual assortment optimization and inference. Two of the reviewers are negative, one is mildly negative and one is mildly positive and highlight a number of concerns, most prominently regarding the limited motivation and presentation, strong assumptions, and weak empirical evaluation. As a consequence, I am unable to recommend acceptance.

**Reviewer Concerns:**

I think the main concerns highlighted by the reviewers are partially addressed.

**Reviewer Scores:**

I do not think the reviewers would have changed their score.

---

### Decision · Program_Chairs · 2026-01-26

Reject